# Manifold Generalization Provably Proceeds Memorization in Diffusion Models

**Zebang Shen**[*]    **Ya-Ping Hsieh**[*]   **Niao He**
ETH Zurich
{zebang.shen, yaping.hsieh, niao.he}@inf.ethz.ch

## Abstract

Diffusion models often generate novel samples even when the learned score is only *coarse*—a phenomenon not accounted for by the standard view of diffusion training as density estimation. In this paper, we show that, under the *manifold hypothesis*, this behavior can instead be explained by coarse scores capturing the *geometry* of the data while discarding the fine-scale distributional structure of the population measure $\mu_{\mathrm{data}}$. Concretely, whereas estimating the full data distribution $\mu_{\mathrm{data}}$ supported on a $k$-dimensional manifold is known to require the classical minimax rate $\tilde{\mathcal{O}}(N^{-1/k})$, we prove that diffusion models trained with coarse scores can exploit the *regularity of the manifold support* and attain a near-parametric rate toward a *different* target distribution. This target distribution has density uniformly comparable to that of $\mu_{\mathrm{data}}$ throughout any $\tilde{\mathcal{O}}\big(N^{-\beta/(4k)}\big)$-neighborhood of the manifold, where $\beta$ denotes the manifold regularity. Our guarantees therefore depend only on the smoothness of the underlying support, and are especially favorable when the data density itself is irregular, for instance non-differentiable. In particular, when the manifold is sufficiently smooth, we obtain that *generalization*—formalized as the ability to generate novel, high-fidelity samples—occurs at a statistical rate strictly faster than that required to estimate the full population distribution $\mu_{\mathrm{data}}$.

## 1 Introduction

Diffusion and score-based generative models deliver striking sample quality in high-dimensional domains (Ho et al., 2020; Song et al., 2021; Dhariwal & Nichol, 2021; Rombach et al., 2022; Karras et al., 2022). Yet a persistent empirical pattern is that genuinely *novel* samples—outputs that are not mere near-duplicates of the training set—often emerge only when the learned score is *coarse*, for instance under early stopping or limited model capacity (Gu et al., 2023; Somepalli et al., 2023; Bonnaire et al., 2025; Achilli et al., 2025b). This seems at odds with the dominant theoretical paradigm, which treats diffusion training as a *density estimation* problem and establishes sampling or convergence guarantees under sufficiently accurate score/denoiser estimation, typically in large-sample regimes (Tang & Yang, 2023; Lee et al., 2023; De Bortoli, 2022; Oko et al., 2023; Azangulov et al., 2024; Chen et al., 2023). In that view, improving score accuracy should monotonically improve approximation to the population distribution. We therefore ask:

> *How can an **inaccurate** score still yield **non-memorized**, high-quality samples?*

We study this question under the *manifold hypothesis* (Fefferman et al., 2016): data concentrate on a $k$-dimensional $C^\beta$ submanifold $\mathcal{M}^\star \subset \mathbb{R}^D$ with $k \ll D$. Our thesis is that the relevant objective behind "generalization" is often not minimax recovery of the full density $\mu_{\mathrm{data}}$, but rather *coverage of $\mathcal{M}^\star$* at a nontrivial spatial resolution.

**A coverage criterion.** Fix $\delta > 0$. Informally, we say that a distribution $\mu$ has *$\delta$-coverage* of $\mu_{\mathrm{data}}$ if there exists a constant $c > 0$, independent of the sample size, such that for every $y \in \mathcal{M}^\star := \mathrm{supp}(\mu_{\mathrm{data}})$,

$$\mu\big(B_\delta^{\mathcal{M}}(y)\big) \ \geq \ c\,\mu_{\mathrm{data}}\big(B_\delta^{\mathcal{M}}(y)\big),$$

---

[*]Equal contributions.

where $B_\delta^{\mathcal{M}}(y)$ is the geodesic ball of radius $\delta$ on $\mathcal{M}^\star$. This formalizes the requirement that $\mu$ does not "miss" any region of $\mathcal{M}^\star$ that is non-negligible under $\mu_{\text{data}}$ at resolution $\delta$. In this light, the empirical distribution $\mu_{\text{emp}}$ faces a fundamental obstruction: the smallest $\delta$ for which $\mu_{\text{emp}}(B_\delta^{\mathcal{M}}(y)) > 0$ for all $y$ scales as $\tilde{\mathcal{O}}(N^{-1/k})$.

In contrast, our main finding is that diffusion sampling with a *coarsely* learned score can nonetheless yield distributions with *much finer* on-manifold coverage.

**Theorem 1.1 (Main; informal)** *Assume $\mu_{\text{data}}$ is supported on a $k$-dimensional $C^\beta$ submanifold $\mathcal{M}^\star \subset \mathbb{R}^D$ and satisfies mild regularity conditions. Given $N$ i.i.d. samples from $\mu_{\text{data}}$, consider a diffusion model trained only to **coarse** score accuracy. Then, with high probability, the induced sampling dynamics is $\tilde{\mathcal{O}}(N^{-1})$-close in squared Hellinger distance to a distribution that achieves $\delta$-coverage at the scale*

$$\delta \;=\; \tilde{\mathcal{O}}\big(N^{-\beta/4k}\big).$$

In particular, when the smoothness parameter $\beta > 4$, diffusion sampling achieves *strictly finer* on-manifold coverage while learning only a *covered surrogate* at a near-parametric rate $\tilde{\mathcal{O}}(N^{-1/2})$. Operationally, this means that the resulting samples lie (approximately) on the underlying data manifold while remaining far from any individual empirical datapoint. In this sense, diffusion models achieve *generalization*: they produce novel, high-quality samples without memorizing the training set.

To put our result in perspective, it is natural to compare it with the classical minimax rate for estimating the full data distribution $\mu_{\text{data}}$. For an $\alpha$-smooth density supported on a $k$-dimensional domain, the optimal rate scales as $\tilde{\mathcal{O}}\big(N^{-\alpha/k}\big)$ (Divol, 2022; Achilli et al., 2025a; Tang & Yang, 2024). This benchmark assumes that $\mu_{\text{data}}$ itself admits an $\alpha$-smooth density, whereas our guarantees instead rely on the *geometric regularity* of the underlying manifold. In particular, the density smoothness $\alpha$ is typically smaller than the manifold regularity $\beta$, and in the regimes of interest one may even have $\alpha \ll \beta$. Consequently, even relative to smooth-density benchmarks, our rate is significantly sharper. The high-level message is thus that *generalization does not require density estimation*.

**Intuition and technical highlights.** Let $\mu_t := \mu_{\text{data}} * \mathcal{N}(0, tI_D)$ denote the Gaussian-smooth data measure and let $\text{Proj}_{\mathcal{M}}$ be the nearest-point projection onto $\mathcal{M}^\star$ (well-defined on a tubular neighborhood of $\mathcal{M}^\star$). A central object in our analysis is the *smooth–then–project* distribution

$$\mu_{\text{proj}} \;:=\; \text{Proj}_{\mathcal{M}\#}\, \mu_t \;=\; \text{Proj}_{\mathcal{M}\#}\Big(\mu_{\text{data}} * \mathcal{N}(0, tI_D)\Big). \qquad (\mu_{\text{proj}})$$

Intuitively (and will be made precise in Theorem 2.4), when $t$ lies in a moderate regime, $\mu_{\text{proj}}$ serves as a canonical *covered surrogate* for $\mu_{\text{data}}$. Moreover, the two operations defining $\mu_{\text{proj}}$—Gaussian smoothing and geometric projection—each enjoy favorable statistical properties:

1. **Smoothing is statistically cheap.** Although $\mu_{\text{emp}}$ is a poor proxy for $\mu_{\text{data}}$ at fine scales, *Gaussian smoothing makes the estimation problem essentially parametric*: for any fixed $t > 0$,

$$\text{KL}(\mu_t \,\|\, \mu_{\text{emp}} * \mathcal{N}(0, tI_D)) \;=\; \tilde{\mathcal{O}}(N^{-1}),$$

where $\tilde{\mathcal{O}}(\cdot)$ hides constants depending on $t$ and $\mathcal{M}^\star$; see Theorem I.1. This estimate in turn implies that the diffusion model can be learned quickly, in Hellinger distance, toward a distribution that approximates $\mu_{\text{proj}}$ defined in $(\mu_{\text{proj}})$; see Theorem 2.1.

2. **Geometry is easier than full density estimation.** Approximating $\mu_{\text{proj}}$ primarily requires recovering the projection map $\text{Proj}_{\mathcal{M}}$, a geometric object that can be estimated at rates significantly faster than recovering $\mu_{\text{data}}$.

Together, these suggest that learning $\mu_{\text{proj}}$ can be substantially easier than learning $\mu_{\text{data}}$ in the minimax sense, while still being sufficient for producing non-memorized, high-quality samples.

**Our approach.** Motivated by these observations, we decompose the analysis into two noise regimes. In the moderate-to-large noise regime ($t \geq t_0$ for a manifold-dependent threshold $t_0$), we assume sufficiently accurate score learning. In this range, training effectively targets the Gaussian-smoothed

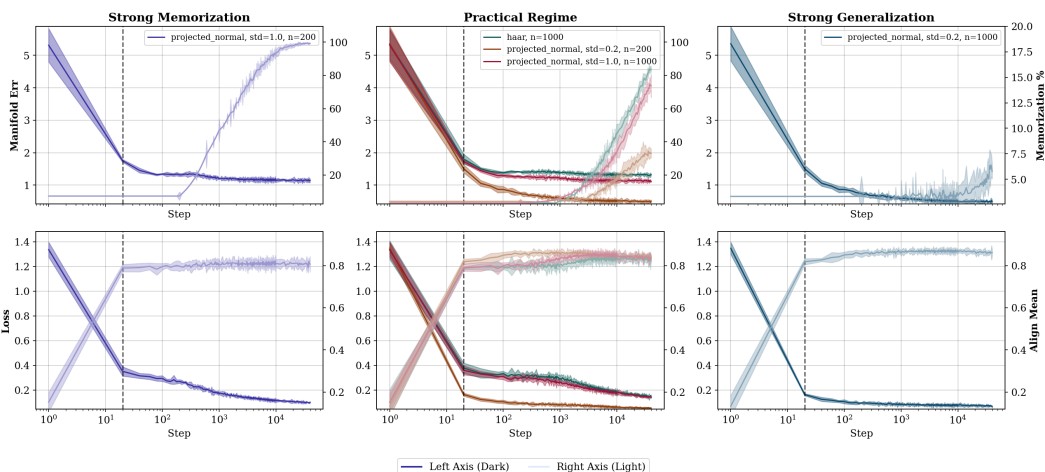

Figure 1: **Geometry precedes memorization in diffusion training.** *Top row:* training dynamics across three regimes. The manifold error (dark, left axis) decreases rapidly, while the memorization rate (light, right axis) stays low for coarsely optimized scores. The "generalization" window is the regime where both manifold error and memorization are small. *Bottom row:* our diagnostic for **manifold learning**. Alongside the training loss (dark, left axis), we report the mean alignment (light, right axis) between the learned score $s^\theta$ and the projection direction, $\langle \mathrm{Proj}_\mathcal{M}, s^\theta \rangle / (\| \mathrm{Proj}_\mathcal{M} \| \| s^\theta \|)$. Across regimes, alignment rises quickly and saturates early, suggesting that the coarse score network first recovers manifold geometry, while memorization is a later-stage effect.

empirical law $\mu_{\mathrm{emp}} * \mathcal{N}(0, t_0 I_D)$ and thus yields a near-parametric approximation of $\mu_{t_0}$ by the preceding discussion; this is the "easy" regime.

Our main technical contribution is in the small-noise regime, where the objective is *geometric recovery* rather than distributional learning (see Figure 1 for empirical evidence and Section I.1 for experimental details): For a function class chosen to reflect both theory and this empirical behavior, we show that a *coarsely* learned score—when *coupled with the ODE integrator most commonly used in practice* (rather than the elementary reverse-time SDEs)—implicitly realizes an approximate projection map $\mathrm{Proj}_{\widehat{\mathcal{M}}}$. Quantitatively, this yields a manifold estimator $\widehat{\mathcal{M}}$ with Hausdorff and projection accuracy (Theorem 2.2 and Theorem J.1)

$$ \mathrm{d}_\mathcal{H}(\widehat{\mathcal{M}}, \mathcal{M}^\star) = \tilde{\mathcal{O}}(N^{-\beta/k}), \qquad \| \mathrm{Proj}_\mathcal{M} - \mathrm{Proj}_{\widehat{\mathcal{M}}} \|_\infty = \tilde{\mathcal{O}}(N^{-\beta/(2k)}). $$

A geometric transfer step then converts projection accuracy into $\delta$-coverage at intrinsic scale $\delta = \tilde{\mathcal{O}}(N^{-\beta/(4k)})$; see Theorem 2.4.

**Literature Review and Preliminaries.** To our knowledge, this approach is original and has few direct precursors in the existing literature. We provide a detailed review in Section A. In addition, due to space constraints, we defer the necessary background material to Section B.

## 2 FAST COVERAGE VIA MANIFOLD GENERALIZATION WITH COARSE SCORES

After a simple reduction to the small-noise regime via Theorem 2.1, our key technical ingredient is Theorem 2.2, which shows that, in the small-noise regime, a *coarsely* learned score implicitly yields a minimax-optimal estimator of the data manifold—equivalently, an estimator of the projection map. We prove our main coverage guarantee in Theorem 2.4.

### 2.1 DIFFUSION SETUP: DENOISING SCORE MATCHING

**Gaussian corruption and marginals.** Let $\mu_{\mathrm{data}}$ be the data-generating distribution supported on $\mathcal{M}^\star \subset \mathbb{R}^D$. For $t > 0$, define the Gaussian corruption kernel

$$ q_t(x \mid x_0) := \mathcal{N}(x; x_0, t I_D), $$

and the corresponding corrupted marginal

$$\mu_t \; \coloneqq \; \int q_t(\cdot \mid x_0) \, \mathrm{d}\mu_{\mathrm{data}}(x_0) \; = \; \mu_{\mathrm{data}} * \mathcal{N}(0, t I_D). \tag{1}$$

For the simplicity of notation, we identify $\mu_t$ with its density w.r.t. the Lebesgue measure on $\mathbb{R}^D$. Note that such density exists for all $t > 0$. The corresponding true score is

$$s^\star(x, t) \; \coloneqq \; \nabla_x \log \mu_t(x), \qquad x \in \mathbb{R}^D. \tag{2}$$

For the Gaussian kernel, the conditional score has the closed form

$$\nabla_x \log q_t(x \mid x_0) \; = \; -\frac{x - x_0}{t}. \tag{3}$$

**Denoising score matching (DSM).** Let $\mathcal{S}$ be a class of time-indexed vector fields, and let $\mu_{\mathrm{emp}}$ denote the empirical measure of $N$ i.i.d. samples from $\mu_{\mathrm{data}}$. For any $s \in \mathcal{S}$, define

$$\mathrm{DSM}_t(s; x_0) \coloneqq \mathbb{E}_{x \sim q_t(\cdot \mid x_0)} \Big[ \big\| s(x, t) - \nabla_x \log q_t(x \mid x_0) \big\|^2 \Big]. \tag{4}$$

At each noise level $t$, diffusion models are commonly trained by denoising score matching, i.e. by regressing onto the average conditional score:

$$\mathrm{DSM}_t(s) \; \coloneqq \; \mathbb{E}_{x_0 \sim \mu_{\mathrm{emp}}} \big[ \mathrm{DSM}_t(s; x_0) \big]. \tag{5}$$

At the population level (replacing $\mu_{\mathrm{emp}}$ by $\mu_{\mathrm{data}}$), the minimizer over all measurable $s(\cdot, t)$ is the marginal score $s^\star(\cdot, t)$ in (2), and the excess risk admits the standard identity

$$\mathrm{DSM}_t(s) - \mathrm{DSM}_t(s^\star) \; = \; \| s(\cdot, t) - s^\star(\cdot, t) \|_{L^2(\mu_t)}^2 \coloneqq \mathbb{E}_{x \sim \mu_t} \big[ \| s(x, t) - s^\star(x, t) \|^2 \big]. \tag{6}$$

**Hybrid sampling dynamics.** Let $\hat{s}(\cdot, t)$ be a learned score. We analyze a two-stage sampler that mirrors common implementations: a reverse-time SDE is run from large noise down to a terminal level $t_0$, and the final segment is integrated via the probability-flow ODE. Concretely, for an arbitrary cutoff time $\tau > 0$, consider

$$\textbf{(SDE stage)} \qquad \mathrm{d}X_t = -\hat{s}(X_t, t)\,\mathrm{d}t \; + \; \mathrm{d}\bar{W}_t, \qquad t : T \searrow t_0, \tag{7}$$

$$\textbf{(ODE stage)} \qquad \mathrm{d}X_t = -\tfrac{1}{2}\,\hat{s}(X_t, t)\,\mathrm{d}t, \qquad t : t_0 \searrow \tau, \tag{8}$$

where $\bar{W}_t$ is a standard Brownian motion run backward in time, so (7) is a reverse-time SDE.[1] This SDE–then–ODE strategy is widely used in practice and is empirically more numerically stable than naive reverse-SDE discretizations, especially at small noise (Ho et al., 2020; Song et al., 2020; Karras et al., 2022).

**Flow map and induced projection surrogate.** Let $\Phi_{\tau \leftarrow t_0} : \mathbb{R}^D \to \mathbb{R}^D$ denote the flow map of the ODE stage (8): for any $x \in \mathbb{R}^D$, $\Phi_{\tau \leftarrow t_0}(x)$ is the solution at time $\tau$ with initial condition $X_{t_0} = x$. We define the induced projection surrogate

$$\mathrm{Proj}_{\widehat{\mathcal{M}}} \; \coloneqq \; \Phi_{\tau \leftarrow t_0}. \tag{9}$$

## 2.2 Large-noise reduction to a smoothed empirical law

Fix a terminal noise level $t_0 > 0$, which is to be specified later as a constant depending only on the manifold. A guiding object throughout our analysis is the *smooth–then–project* surrogate $\mu_{\mathrm{proj}}$ in ($\mu_{\mathrm{proj}}$). In line with the hybrid sampler of Section 2.1, we first isolate the *large-noise* regime $t \geq t_0$, where score estimation is statistically and algorithmically easier under a standard condition on the training error for DSM.

**Assumption 1 (Large-noise DSM; $\varepsilon_{\mathrm{LN}}$-accurate training)** *Fix $t_0 > 0$ and a large terminal time $T > t_0$ for the SDE stage. Assume the learned score $\hat{s}(\cdot, t)$ satisfies the integrated excess DSM bound*

$$\int_{t_0}^{T} \Big( \mathrm{DSM}_t(\hat{s}) - \inf_{s(\cdot, t)} \mathrm{DSM}_t(s) \Big) \, \mathrm{d}t \; \leq \; \varepsilon_{\mathrm{LN}}. \tag{10}$$

---

[1] For simplicity we present the VE-style form above (Song et al., 2021); the same decomposition (reverse-time SDE and probability-flow ODE) holds for standard VP/VE schedules with drift/diffusion coefficients, and our arguments extend to those settings with notational changes.

Our main result in this section, whose proof is deferred to Section C, shows that in this regime, accurate score learning ensures that the reverse-time dynamics at time $t_0$ approximately recovers the smoothed distribution $\mu_{\mathrm{data}} * \mathcal{N}(0, t_0 I_D)$. The problem is therefore reduced to understanding the terminal ODE map $\mathrm{Proj}_{\widehat{\mathcal{M}}}$.

**Theorem 2.1 (Large-noise reduction)** *Let $\mu_{\mathrm{DM}}$ be the output distribution of the hybrid sampler (7)–(8), and recall $\mathrm{Proj}_{\widehat{\mathcal{M}}} = \Phi_{\tau \leftarrow t_0}$ from (9). Then, under Assumption 1, for any $a > 0$, with probability at least $1 - N^{-a}$ over the $N$ samples and any algorithmic randomness,*

$$H^2\Big( \mathrm{Proj}_{\widehat{\mathcal{M}}\#} \big( \mu_{\mathrm{data}} * \mathcal{N}(0, t_0 I_D) \big) , \mu_{\mathrm{DM}} \Big) = \mathcal{O}\left( \frac{a \log N}{N} \right) + \mathcal{O}(\varepsilon_{\mathrm{LN}}), \qquad (11)$$

*where $H^2(P, Q) = \int (\sqrt{p} - \sqrt{q})^2$ denotes squared Hellinger distance (for densities $p, q$).*

### 2.3    SMALL-NOISE COARSE SCORES AND (9) AS (NEAR-)MINIMAX PROJECTION MAPS

We now turn to the most delicate regime, namely the *small-noise* interval $t \in [\tau, t_0]$. Our goal in this section is to show that the estimator (9) is *minimax-optimal* for recovering the projection map onto the data manifold $\mathcal{M}^\star$. Following the standard setup of (Aamari & Levrard, 2019; Divol, 2022), we work over the class of manifolds whose reach is uniformly lower bounded by $\zeta_{\min}$ (defined in Equation (B.3)). *For the remainder of the paper, we fix $t_0 := \zeta_{\min}/4$.*

**Key intuition: geometry dominates density at small noise.**    Our guiding intuition is provided by the following small-noise expansion of the population score, recently derived by Li et al. (2025); Liu et al. (2025):[2]

$$\forall x \in \mathcal{M}^\star, \quad s_t^\star(x, t) = -\frac{1}{t}\big(x - \mathrm{Proj}_{\mathcal{M}}(x)\big) + \nabla_{\mathcal{M}^\star} \log p\big(\mathrm{Proj}_{\mathcal{M}}(x)\big) + \frac{1}{2}\mathbb{H}(x) + r_t(x), \quad (12)$$

where $p$ is the density of $\mu_{\mathrm{data}}$ on $\mathcal{M}^\star$ (w.r.t. volume), $\nabla_{\mathcal{M}^\star}$ denotes the Riemannian gradient on $\mathcal{M}^\star$, $\mathbb{H}$ is the mean curvature of $\mathcal{M}^\star$, and $r_t(x) = o(1)$ as $t \downarrow 0$ (uniformly on a fixed tube around $\mathcal{M}^\star$ under the regularity assumptions of Li et al. (2025); Liu et al. (2025)).

The expansion highlights a sharp scale separation: the *normal* "projection" term $-(x - \mathrm{Proj}_{\mathcal{M}}(x))/t$ has magnitude $\Theta(t^{-1})$, while the *tangential* density term $\nabla_{\mathcal{M}^\star} \log p(\mathrm{Proj}_{\mathcal{M}}(x))$ remains $O(1)$. Consequently, recovering only the leading $t^{-1}$ term is enough to capture the geometry of $\mathcal{M}^\star$: even if the score error diverges as $t^{-\gamma}$ for some $\gamma \in (0, 1)$, the leading-order component can still faithfully encode the projection direction $\mathrm{Proj}_{\mathcal{M}}$.

**Technical challenges and contributions over prior work.**    To our knowledge, the only existing minimax-optimal manifold estimators are the *local polynomial* procedures of (Aamari & Levrard, 2019) (and subsequent refinements such as (Azangulov et al., 2024)). While our analysis draws substantial inspiration from these works, translating minimax manifold estimation into the diffusion/score-learning setting requires overcoming two obstacles:

**(i) From nonparametric geometry estimation to score learning with coarse accuracy.**  The estimators in (Aamari & Levrard, 2019) are not tied to diffusion models and do not arise from (or naturally interact with) score learning. In particular, they are nonparametric and therefore do not suggest a direct route to implementations compatible with standard neural architectures or to analyses driven by *coarse* score accuracy.

**(ii) Smoothness is essential for downstream coverage.** As noted in (Aamari & Levrard, 2019), the estimator is constructed as a collection of local polynomial patches, and in general there is no guarantee that the resulting set forms a globally smooth submanifold. While such nonsmoothness is acceptable for certain geometric risk criteria, it is incompatible with the *coverage* guarantees proved in Section 2.4, where smooth projection-like dynamics play a central role.

**Our approach addresses these challenges on two fronts.**

---

[2]This expansion is included only as heuristic motivation for the discussion and for the choice of function class below. Its derivation in Li et al. (2025); Liu et al. (2025) requires additional regularity on the density $p$ (in particular, $p \in \mathcal{C}^1$). Our analysis does *not* rely on (12) and imposes no such regularity assumption on $p$.

*Front 1: a PDE-based function class for smooth manifold recovery.* Motivated by the small-noise expansion (12), we capture the leading geometric term $-\frac{1}{t}\big(x - \mathrm{Proj}_{\mathcal{M}}(x)\big)$ through a *distance potential* $\eta$. A key ingredient is the *Eikonal equation* satisfied by the squared distance-to-manifold potential (recall (B.2) for notation): $\eta^{\star}(x) := \frac{1}{2}\,\mathrm{dist}(x, \mathcal{M}^{\star})^2$. On any tubular neighborhood where $\mathrm{Proj}_{\mathcal{M}}$ is well-defined, $\eta^{\star}$ verifies the key relation[3]:

$$\|\nabla\eta(x)\|^2 \;=\; 2\eta(x). \tag{Eik}$$

This viewpoint has two advantages. First, as a differential constraint, (Eik) admits principled *parametric* approximations—for instance via physics-informed architectures that enforce PDE structure during training (Raissi et al., 2019). Second, and more importantly for our theory, we show that under the boundary and regularity conditions specified in (D.4), the eikonal constraint is (in a precise sense) *necessary and sufficient* for $\eta$ to be locally the squared distance to *some* smooth submanifold. Consequently, unlike (Aamari & Levrard, 2019), our estimator targets a *smooth* manifold surrogate and hence induces a smooth projection map. We develop this correspondence in Sections F to G.

*Front 2: from coarse DSM control to minimax projection estimation.* Once the function class is fixed and shown to be well-defined, the remaining task is to connect *coarse* score learning to accurate projection estimation. Our central observation is that a *uniform* control of the DSM objective—formalized in Assumption 2—implies accuracy for a nonlinear analogue of PCA that we term *Principal Manifold Estimation (PME)*; see (H.4) for the loss definition. We then show that any sufficiently accurate PME estimator yields a projection estimator that achieves the same minimax rate as the local polynomial estimators of (Aamari & Levrard, 2019).

We defer the details to Section D

**Local denoising score matching.** Having specified the score class, we now formalize what it means to optimize *coarsely* in the small-noise regime. The key point is that we impose control *uniformly over local neighborhoods* of the empirical support—rather than only in expectation under $\mu_{\mathrm{emp}}$ as in the classical DSM objective in (5)–while allowing this control to deteriorate (and possibly blow up) as $t \to 0$. To this end, we introduce a localized variant of DSM as follows. Recall the per-sample loss $\mathrm{DSM}_t(s; x_0)$ from Equation (4). Fix a bandwidth $h > 0$ (to be specified in Theorem 2.2). For each reference point $x_{\mathrm{ref}} \in \mathrm{supp}(\mu_{\mathrm{emp}})$, define the localized empirical measure

$$\mu_{\mathrm{emp}}^{x_{\mathrm{ref}},h} \;:=\; \mathbb{1}_{B_D^{\mathrm{Euc}}(x_{\mathrm{ref}},h)}\,\mu_{\mathrm{emp}},$$

i.e., the restriction of $\mu_{\mathrm{emp}}$ to the Euclidean ball $B_D^{\mathrm{Euc}}(x_{\mathrm{ref}}, h)$. We then define the *local* DSM objective at noise level $t$ by

$$\mathrm{LDSM}_t(s; x_{\mathrm{ref}}) \;:=\; \mathbb{E}_{x_0 \sim \mu_{\mathrm{emp}}^{x_{\mathrm{ref}},h}}\big[\,\mathrm{DSM}_t(s; x_0)\,\big]. \tag{13}$$

The following assumption formalizes our coarse optimization requirement on the score error.

**Assumption 2 (Local-DSM coarse optimality in the small-noise regime)** *Fix $t_0 = \zeta_{\min}/4$ and a bandwidth $h > 0$. For each $t \in (0, t_0]$, let $\mathcal{S}$ denote the candidate class in (D.5), and let $\hat{s}(\cdot, t) \in \mathcal{S}$ be the learned score at time $t$. Assume that there exist constants $C > 0$ such that, for all $t \in (\tau, t_0]$[4],*

$$\sup_{x_{\mathrm{ref}} \in \mathrm{supp}(\mu_{\mathrm{emp}})} \left\{ \mathrm{LDSM}_t\big(\hat{s}(\cdot, t); x_{\mathrm{ref}}\big) \;-\; \inf_{s \in \mathcal{S}} \mathrm{LDSM}_t(s; x_{\mathrm{ref}}) \right\} \;\leq\; C\,t^{-1}. \tag{14}$$

**Remark.** Assumption 2 is intentionally *coarse*: it only asks $\hat{s}(\cdot, t)$ to capture the leading *projection* component of the small-noise score,

$$s^{\star}(x, t) \;\approx\; -\frac{x - \mathrm{Proj}_{\mathcal{M}}(x)}{t},$$

and places essentially no constraint on the lower-order, *data-dependent* contribution (e.g., tangential density information along $\mathcal{M}^{\star}$). As a result, the assumption is calibrated for learning *geometry* (a projection-like drift), but is too weak to imply full recovery of the data distribution in the small-noise regime—which, as we shall see in Section 2.4, is not required to explain the kind of "generalization" empirically observed in diffusion models.

We are finally ready to state our main result, whose proof is deferred to Section E.

---

[3]While the Eikonal equation is necessary condition for $\eta^{\star}$ to be a squared distance function to $\mathcal{M}^{\star}$, it alone is insufficient, e.g. a constant 0 function also satisfies Equation (Eik).

[4]Here, the factor $1/t$ can be replaced by $1/t^{\gamma}$ for any $\gamma \in (0, 2)$; we take $\gamma = 1$ for notational simplicity.

**Theorem 2.2 (Hausdorff recovery and projection accuracy)** *Assume that $\mu_{\mathrm{data}}$ is supported on a compact, connected, boundaryless, $k$-dimensional $C^\beta$ submanifold $\mathcal{M}^\star \subset \mathbb{R}^D$ with $\beta \geq 2$, and that $\mathrm{reach}(\mathcal{M}^\star) \geq \zeta_{\min} > 0$. Suppose that the parameter $\mathbf{L}$ in $\mathcal{D}_L^k$ is chosen sufficiently large such that $\eta^\star \in \mathcal{D}_L^k$, where $\eta^\star$ is defined in Equation (B.1). Pick $h = \Theta((\log N/N)^{1/k})$. Let $\hat{s}$ be a score estimate learned from $N$ i.i.d. samples satisfying Assumption 2. For a sufficiently large $N$, the estimator $\widehat{\mathcal{M}} := \{x \in \mathbb{U} : \hat{s}(x,t) = 0\}$ satisfies with probability $1 - \mathcal{O}\left(\left(\frac{1}{N}\right)^{\frac{\beta}{k}}\right)$: for all $t \in (\tau, t_0]$,*

$$\mathrm{d}_{\mathcal{H}}(\widehat{\mathcal{M}}, \mathcal{M}^\star) = \tilde{\mathcal{O}}(N^{-\beta/k}), \tag{15}$$

$$\sup_{x \in \mathcal{T}_r(\mathcal{M}^\star)} \left\| t\,\hat{s}(x,t) - \mathrm{Proj}_{\mathcal{M}}(x) \right\| = \tilde{\mathcal{O}}(N^{-\beta/(2k)}), \qquad r = \zeta_{\min}/4, \tag{16}$$

*where $\tilde{\mathcal{O}}(\cdot)$ hides polylogarithmic factors in $N$ and constants depending only on $(k, D, \beta, \zeta_{\min})$.*

As alluded to above, the main contribution of Theorem 2.2 is to show that—in contrast to the nonsmooth, piecewise-polynomial estimators of Aamari & Levrard (2019), which are fully nonparametric and not tied to diffusion models—a score that is only *coarsely* optimized under the *local* DSM objective already suffices for near-optimal projection estimation, provided we restrict attention to the geometry-motivated class (D.5). In particular, the resulting estimator matches the rate of Aamari & Levrard (2019) up to at most a polylogarithmic factor, and is therefore (nearly) minimax-optimal.

## 2.4 FROM PROJECTION DYNAMICS TO COVERAGE

Having established that a coarse score implicitly learns the manifold, we now show that this geometric recovery already suffices for strong *coverage* guarantees. Specifically, we prove (in the sense formalized in Theorem 2.3) that the diffusion output distribution $\mu_{\mathrm{DM}}$ produced from coarsely learned scores achieves an on-manifold coverage resolution that is strictly finer than what an empirical measure supported on $N$ atoms can provide. This formalizes the message that "generalization"—in the operational sense of producing a novel point *on the manifold*—is statistically much easier than full density estimation. Deferred proofs are collected in Sections J to K.

**Key intuition: restricted tangential shifts imply good coverage.** Recall from Theorem 2.1 that $\mu_{\mathrm{DM}}$ converges at a fast rate to the population surrogate

$$\widehat{\mu_{\mathrm{proj}}} := \mathrm{Proj}_{\widehat{\mathcal{M}} \#}\left(\mu_{\mathrm{data}} * \mathcal{N}(0, t_0 I_D)\right), \tag{17}$$

where $\mathrm{Proj}_{\widehat{\mathcal{M}}}$ is the flow map associated with the ODE (8); see (9). Thus, it suffices to prove coverage for $\widehat{\mu_{\mathrm{proj}}}$.

Our theory in Section 2.3 suggests modeling the learned score in the terminal regime $t \in [\tau, t_0]$ as the sum of a leading-order projection term and a (coarse) remainder error:

$$\hat{s}(x,t) = -\frac{x - \mathrm{Proj}_{\mathcal{M}}(x)}{t} + \frac{e(x,t)}{t}, \qquad t \in [\tau, t_0]. \tag{18}$$

We will use the shorthand

$$\varepsilon := \sup_{t \in [\tau, t_0]} \sup_{x \in \mathcal{T}_r(\mathcal{M})} \|e(x,t)\|, \tag{19}$$

for some tubular radius $r \leq \zeta_{\min}/4$ (so that $\mathrm{Proj}_{\mathcal{M}}$ is single-valued on $\mathcal{T}_r(\mathcal{M}^\star)$).

The high-level intuition is that running (8) with a score of the form (18)–(19) (for appropriately chosen $t_0$ and $\tau$) produces samples for which:

- **Normal contraction.** The output lies $\tilde{\mathcal{O}}(\varepsilon)$-close to $\mathcal{M}^\star$ (in ambient distance), by a direct contraction estimate for $\mathrm{dist}(\cdot, \mathcal{M}^\star)$ along the terminal-time ODE (Theorem J.1).

- **Restricted tangential drift.** More importantly, the induced displacement *along* the manifold is also small: the "tangential shift"—i.e. the geodesic deviation of $\mathrm{Proj}_{\mathcal{M}}(\mathrm{Proj}_{\widehat{\mathcal{M}}}(x))$ from $\mathrm{Proj}_{\mathcal{M}}(x)$— scales like $\tilde{\mathcal{O}}(\sqrt{\varepsilon})$ via an ambient-to-geodesic transfer bound (Theorem I.4).

Therefore, it is natural to separate the argument into a "baseline" and a "stability" step. As a baseline, we first analyze the idealized distribution obtained by projecting the smoothed population measure with the *true* projection,

$$\mu_{\mathrm{proj}} := \mathrm{Proj}_{\mathcal{M}\#}\big(\mu_{\mathrm{data}} * \mathcal{N}(0, t_0 I_D)\big), \tag{20}$$

and show that it has good coverage of $\mu_{\mathrm{data}}$ (via the local-trivialization lower bound in Theorem K.2). The remaining step—replacing $\mathrm{Proj}_{\mathcal{M}}$ by $\mathrm{Proj}_{\widehat{\mathcal{M}}}$ in (20)—is then purely geometric: given a map on $\mathcal{M}$ that moves each point geodesically by at most $\tilde{\mathcal{O}}(\sqrt{\varepsilon})$, how large a "hole" (a region of vanishing mass, hence failed coverage) can it create? Intuitively, such a map can only deform sets at the $\sqrt{\varepsilon}$ scale, so the worst-case loss of coverage is controlled at a comparable resolution. Finally, plugging in the minimax estimate $\sqrt{\varepsilon} = \tilde{\mathcal{O}}\big(N^{-\beta/(4k)}\big)$ from Theorem 2.2 completes the picture. See Section J for details.

**Restricted normal and tangential shifts lead to good coverage.** We are now ready to show that diffusion models equipped with a *coarse* score, when sampled via (7)–(8), achieve substantially better coverage of the data manifold than the empirical measure. For parameters $(\delta, \alpha) > 0$ and $y \in \mathcal{M}^\star$, define the $\alpha$-*thickened geodesic ball*

$$B_{\delta,\alpha}^{\mathcal{M}}(y) := \Big\{x \in \mathcal{T}_{\zeta_{\min}}(\mathcal{M}^\star): \ \mathrm{dist}(x, \mathcal{M}^\star) \leq \alpha, \ \mathrm{Proj}_{\mathcal{M}}(x) \in B_\delta^{\mathcal{M}}(y)\Big\}, \tag{21}$$

where $B_\delta^{\mathcal{M}}(y) \subset \mathcal{M}^\star$ denotes the intrinsic geodesic ball of radius $\delta$ centered at $y$. Our notion of coverage is as follows.

**Definition 2.3 (Covering)** *Let $c > 0$. We say that a probability measure $\mu$ $(\alpha, \delta, c)$-**covers** $\mu_{\mathrm{data}}$ if, for every $y \in \mathcal{M}^\star$,*

$$\mu\big(B_{\delta,\alpha}^{\mathcal{M}}(y)\big) \geq c\,\mu_{\mathrm{data}}\big(B_{\delta,\alpha}^{\mathcal{M}}(y)\big)$$

**Remark.** Since $\mu_{\mathrm{data}}$ is supported on $\mathcal{M}^\star$, thickening does not change its mass:

$$\mu_{\mathrm{data}}\big(B_{\delta,\alpha}^{\mathcal{M}}(y)\big) = \mu_{\mathrm{data}}\big(B_\delta^{\mathcal{M}}(y)\big), \qquad \text{for all } \alpha > 0.$$

Intuitively, $\mu$ $(\alpha, \delta, c)$-covers $\mu_{\mathrm{data}}$ if it places mass comparable to $\mu_{\mathrm{data}}$ on every geodesic ball of radius $\delta$, after robustifying that neighborhood by an $\alpha$-thickening in the normal direction, uniformly over all centers $y \in \mathcal{M}^\star$.

This notion highlights a fundamental limitation of empirical measures: If $\mu_{\mathrm{emp}}$ is supported on $N$ samples on a $k$-dimensional manifold, then its support can form at best an $\tilde{\mathcal{O}}(N^{-1/k})$-net. Consequently, for any $\delta = o(N^{-1/k})$ there exists $y \in \mathcal{M}^\star$ such that $\mu_{\mathrm{emp}}(B_{\delta,\alpha}^{\mathcal{M}}(y)) = 0$ while $\mu_{\mathrm{data}}(B_\delta^{\mathcal{M}}(y)) > 0$, so $\mu_{\mathrm{emp}}$ cannot $(\alpha, \delta, c)$-cover $\mu_{\mathrm{data}}$ for any $c > 0$ at that resolution. In contrast, the following theorem shows that $\widehat{\mu_{\mathrm{proj}}}$ *does* $(\alpha, \delta, c)$-cover $\mu_{\mathrm{data}}$ at an intrinsic resolution far finer than what $\mu_{\mathrm{emp}}$ can achieve, provided the manifold is sufficiently smooth (e.g. $\beta \gg 1$ under our regularity assumptions).

**Theorem 2.4 (Coverage of the population surrogate)** *Let $t_0 = \zeta_{\min}/4$, and let $\widehat{\mu_{\mathrm{proj}}}$ be the surrogate measure defined in* (17). *Assume the coarse-score conditions of Assumption 2 and the function class specification in Section 2.3. Then there exist constants $c_{\min}$ (explicitly given in Equations (K.15) to (K.16)) and $N_0 \in \mathbb{N}$, depending only on $p_{\min}, p_{\max}$ and geometric parameters of $\mathcal{M}^\star$, such that for all $N \geq N_0$, the measure $\widehat{\mu_{\mathrm{proj}}}$ $(\alpha, \delta, c_{\min})$-covers $\mu_{\mathrm{data}}$ with*

$$\alpha = \tilde{\mathcal{O}}\big(N^{-\beta/(2k)}\big), \qquad \delta = \tilde{\mathcal{O}}\big(N^{-\beta/(4k)}\big). \tag{22}$$

As discussed at the beginning of this subsection, the result is essentially an immediate consequence of Theorems J.1 to J.2; the remaining steps are largely tedius calculations; see Section K.

ACKNOWLEDGMENT

The work is supported by Swiss National Science Foundation (SNSF) Project Funding No. 200021-207343 and SNSF Starting Grant. YPH thanks Parnian Kassraie for thoughtful discussions on the experiments and for generously sharing her expertise, which awakened a long-lost flamboyance in the author.

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

# Appendix

## A    LITERATURE REVIEW

**Minimax manifold estimation vs. diffusion theory.**    A classical line of work develops minimax-optimal rates for estimating (i) an embedded manifold $\mathcal{M}^\star$ and its local geometry (Aamari & Levrard, 2019) and (ii) measures supported on $\mathcal{M}^\star$, under reach and $C^\beta$ regularity assumptions; see, e.g., (Divol, 2022). More recent diffusion theory adapts parts of this minimax toolkit to obtain sharp *distributional* recovery guarantees for diffusion models (Oko et al., 2023; Azangulov et al., 2024; Tang & Yang, 2024). However, this literature does not address our motivating puzzle—why only *coarse* scores can still yield *novel*, high-quality samples—and it does not provide finite-sample guarantees phrased in terms of *on-manifold coverage*. In particular, to the best of our knowledge, no existing work establishes minimax-style rates for *manifold (or projection) estimation via diffusion models*. Finally, we emphasize that our coarse-score requirement (Assumption 2) alone *cannot* guarantee distributional recovery, since it may hold for two very different distributions $\mu_{\text{data}}$ and $\mu'_{\text{data}}$ as long as they share the same support.

**Geometry, memorization, and interventions in diffusion models.**    A growing empirical and conceptual literature suggests that diffusion models encode salient geometric information, especially at small noise: score geometry has been used to estimate intrinsic or local dimension (Stanczuk et al., 2022; Kamkari et al., 2024), and memorization has been analyzed through the geometry of learned manifolds or selective loss of tangent directions (Ross et al., 2024b; Achilli et al., 2024). Numerous algorithmic interventions aim to mitigate memorization (often motivated by privacy) without explicit geometric modeling (Somepalli et al., 2023; Gu et al., 2023; Daras et al., 2023; Wen et al., 2024; Daras et al., 2024; Kazdan et al., 2024; Chen et al., 2024; Ren et al., 2024; Wu et al., 2024; Liu et al., 2024; Ross et al., 2024a; Wang et al., 2024; Zhang et al., 2024; Jain et al., 2024; Hintersdorf et al., 2025; Shah et al., 2025). Recent theory further sharpens the memorization/generalization picture, e.g. by proving separations between empirical and population objectives and corresponding approximation barriers (Ye et al., 2025) or by linking model collapse under synthetic-data training to a generalization-to-memorization transition driven by entropy decay (Shi et al., 2025). Complementary stylized analyses study phase transitions under latent-manifold models (Achilli et al., 2025b;a) or explain novelty via implicit score smoothing and interpolation (Chen, 2025; Farghly et al., 2025).

Despite this progress, we are not aware of results that quantify *finite-sample statistical rates* separating the difficulty of learning *geometry* from that of learning the *distribution*—a key step in our analysis. Closest in spirit are Li et al. (2025); Liu et al. (2025), which identify a geometry–distribution separation at the *population* level: in the small-noise limit, geometric information encoded by the score is substantially more robust than distributional information. This observation provides key theoretical motivation for our choice of function class in Section 2.3. However, Li et al. (2025); Liu et al. (2025) do not provide a statistical analysis, whereas our results are explicitly finite-sample and tailored to *coverage*, whose proof requires substantially different techniques.

## B    PRELIMINARIES AND PROBLEM SETUP

We recall standard definitions from statistical estimation of manifolds; see, e.g., (Aamari & Levrard, 2019; Divol, 2022).

**Embedded manifolds.**    Throughout the paper, we assume that every manifold $\mathcal{M} \subset \mathbb{R}^D$ is a compact, connected, boundaryless, embedded $k$-dimensional submanifold, where $1 \leq k \leq D - 1$. We reserve the notation $\mathcal{M}^\star$ for the support of $\mu_{\text{data}}$, that is,

$$\mathcal{M}^\star := \text{supp}(\mu_{\text{data}}).$$

Let $T_y\mathcal{M}$ and $N_y\mathcal{M}$ denote the tangent and normal spaces at a point $y \in \mathcal{M}$. The embedding induces a Riemannian metric on $\mathcal{M}$; we write $d_\mathcal{M}$ for the corresponding geodesic distance and

$$B_\delta^\mathcal{M}(y) := \{y' \in \mathcal{M} : d_\mathcal{M}(y', y) \leq \delta\}$$

for the geodesic ball of radius $\delta$ centered at $y$. Let $\mathrm{vol}_{\mathcal{M}}$ denote the Riemannian volume measure.

**$\beta$-smoothness.** Let $\beta \geq 2 \in \mathbb{N}$. We say that $\mathcal{M}$ is $\beta$-*smooth* (i.e. of class $C^\beta$) if for every $y \in \mathcal{M}$ there exist neighborhoods $U \subset \mathbb{R}^D$ of $y$ and $V \subset \mathbb{R}^k$ of $0$, and a $C^\beta$ immersion $\phi : V \to \mathbb{R}^D$ such that $\phi(V) = U \cap \mathcal{M}$.

**Reach, tubular neighborhood, and projection.** For $x \in \mathbb{R}^D$ define

$$\mathrm{dist}(x, \mathcal{M}) := \inf_{y \in \mathcal{M}} \|x - y\| \quad \text{and} \quad \eta^\star(x) := \frac{1}{2} \mathrm{dist}^2(x, \mathcal{M}^\star), \tag{B.1}$$

and the tubular neighborhood $\mathcal{T}_r(\mathcal{M}) := \{x \in \mathbb{R}^D : \mathrm{dist}(x, \mathcal{M}) < r\}$. The *reach* $\mathrm{reach}(\mathcal{M}) \in (0, \infty]$ is the largest $r$ such that every point in $\mathcal{T}_r(\mathcal{M})$ has a *unique* nearest point on $\mathcal{M}$ (Federer, 1959). Equivalently, for any $r < \mathrm{reach}(\mathcal{M})$ the nearest-point projection $\mathrm{Proj}_{\mathcal{M}} : \mathcal{T}_r(\mathcal{M}) \to \mathcal{M}$ is well-defined by

$$\mathrm{Proj}_{\mathcal{M}}(x) = \arg \min_{y \in \mathcal{M}} \|x - y\|.$$

It is well known that every compact $\mathcal{C}^2$ submanifold has strictly positive reach (Thäle, 2008, Proposition 14). A basic identity linking the squared-distance and the projection, which we will use repeatedly, is

$$\forall x \in \mathcal{T}_{\mathrm{reach}(\mathcal{M}^\star)}(\mathcal{M}^\star), \qquad \nabla \eta^\star(x) = x - \mathrm{Proj}_{\mathcal{M}}(x). \tag{B.2}$$

We note that positive reach is a minimal regularity condition ensuring stability of projection and local geometric control; see, e.g., (Federer, 1969; Thäle, 2008). From a statstical perspective, (Aamari & Levrard, 2019, Theorem 1) shows that if the model class allows the reach to degenerate to $0$, then statistical estimation becomes ill-posed. Therefore, throughout this work, we assume some non-zero lower bound on the reach of $\mathcal{M}^\star$ is known, i.e.

$$\mathrm{reach}(\mathcal{M}^\star) \geq \zeta_{\min} > 0. \tag{B.3}$$

The estimator of $\zeta_{\min}$ can be found, for example, in (Aamari et al., 2019).

**Set-distance and local geometry metrics.** For closed sets $A, B \subset \mathbb{R}^D$, the Hausdorff distance is

$$\mathrm{d}_{\mathcal{H}}(A, B) := \max \left\{ \sup_{a \in A} \mathrm{dist}(a, B), \sup_{b \in B} \mathrm{dist}(b, A) \right\}.$$

For two $k$-dimensional subspaces $U, V \subset \mathbb{R}^D$, let $P_U, P_V$ be the orthogonal projections; a common distance is $\|P_U - P_V\|_{\mathrm{op}}$, which equals $\sin(\theta_{\max})$ where $\theta_{\max}$ is the largest principal angle between $U$ and $V$.

**Distributions on $\mathcal{M}^\star$.** We model the data distribution as a probability measure $\mu_{\mathrm{data}}$ supported on $\mathcal{M}^\star$ and absolutely continuous with respect to $\mathrm{vol}_{\mathcal{M}^\star}$:

$$\mu_{\mathrm{data}}(\mathrm{d}y) = p(y) \, \mathrm{vol}_{\mathcal{M}^\star}(\mathrm{d}y).$$

We assume the on-manifold density is bounded: there exist constants $0 < p_{\min} \leq p_{\max} < \infty$ such that $p_{\min} \leq p(y) \leq p_{\max}$ for all $y \in \mathcal{M}^\star$. Importantly, we impose no additional regularity (such as smoothness) on $p$.

## C  PROOF OF THEOREM 2.1

In this section, we prove the main result in Section 2.2.

**Proof.** [proof of Theorem 2.1]

**Step 1: identify the empirical DSM minimizer and excess-risk identity.** Fix $t \in (t_0, T]$ and set $\sigma = \sqrt{t}$. Define the empirical corrupted marginal $\mu_{\mathrm{emp}}^\sigma := \mu_{\mathrm{emp}} * \mathcal{N}(0, tI_D)$. Consider the DSM objective (5):

$$\mathrm{DSM}_t(s) := \mathbb{E}_{x_0 \sim \mu_{\mathrm{emp}}} \mathbb{E}_{x \sim \mathcal{N}(x_0, tI_D)} \big\| s(x, t) - \nabla_x \log \mathcal{N}(x; x_0, tI_D) \big\|^2, \tag{C.1}$$

and let $s^{\mathrm{emp}}(\cdot, t) \in \arg\min_{s(\cdot,t)} \mathrm{DSM}_t(s)$ be its minimizer (over all measurable vector fields). Conditioning on $x$ shows the pointwise minimizer is the regression function

$$s^{\mathrm{emp}}(x, t) = \mathbb{E}[\nabla_x \log \mathcal{N}(x; x_0, tI_D) \mid x].$$

Using Bayes' rule and differentiating under the integral,

$$\begin{aligned}
\nabla_x \log \mu_{\mathrm{emp}}^\sigma(x) &= \frac{\nabla_x \int \mathcal{N}(x; x_0, tI_D)\, \mathrm{d}\mu_{\mathrm{emp}}(x_0)}{\int \mathcal{N}(x; x_0, tI_D)\, \mathrm{d}\mu_{\mathrm{emp}}(x_0)} \\
&= \frac{\int \mathcal{N}(x; x_0, tI_D)\, \nabla_x \log \mathcal{N}(x; x_0, tI_D)\, \mathrm{d}\mu_{\mathrm{emp}}(x_0)}{\int \mathcal{N}(x; x_0, tI_D)\, \mathrm{d}\mu_{\mathrm{emp}}(x_0)} \\
&= \mathbb{E}[\nabla_x \log \mathcal{N}(x; x_0, tI_D) \mid x],
\end{aligned}$$

hence $s^{\mathrm{emp}}(x, t) = \nabla_x \log \mu_{\mathrm{emp}}^\sigma(x)$ for $\mu_{\mathrm{emp}}^\sigma$-a.e. $x$. Moreover, the usual regression Pythagorean identity yields the excess-risk decomposition

$$\mathrm{DSM}_t(\hat{s}) - \mathrm{DSM}_t(s^{\mathrm{emp}}) = \|\hat{s}(\cdot, t) - s^{\mathrm{emp}}(\cdot, t)\|_{L^2(\mu_{\mathrm{emp}}^\sigma)}^2. \tag{C.2}$$

**Step 2: from excess DSM to a KL bound on the SDE-stage marginal.** Let $\mathbb{P}^{\mathrm{emp}}$ be the path law of the reverse-time SDE stage on $[t_0, T]$ driven by drift $-s^{\mathrm{emp}}(\cdot, t)$, and let $\mathbb{P}^{\hat{s}}$ be the corresponding path law driven by $-\hat{s}(\cdot, t)$, using the same diffusion coefficient and the same initialization at time $T$. By Girsanov's theorem,

$$\mathrm{KL}(\mathbb{P}^{\mathrm{emp}} \,\|\, \mathbb{P}^{\hat{s}}) = \frac{1}{2}\, \mathbb{E}_{\mathbb{P}^{\mathrm{emp}}} \int_{t_0}^T \left\| \hat{s}(X_t, t) - s^{\mathrm{emp}}(X_t, t) \right\|^2 \mathrm{d}t.$$

Under $\mathbb{P}^{\mathrm{emp}}$, the time-$t$ marginal equals $\mu_{\mathrm{emp}}^{\sqrt{t}}$ by construction, hence

$$\mathrm{KL}(\mathbb{P}^{\mathrm{emp}} \,\|\, \mathbb{P}^{\hat{s}}) = \frac{1}{2} \int_{t_0}^T \|\hat{s}(\cdot, t) - s^{\mathrm{emp}}(\cdot, t)\|_{L^2(\mu_{\mathrm{emp}}^{\sqrt{t}})}^2 \, \mathrm{d}t = \frac{1}{2} \int_{t_0}^T \left( \mathrm{DSM}_t(\hat{s}) - \mathrm{DSM}_t(s^{\mathrm{emp}}) \right) \mathrm{d}t,$$

where we used (C.2). Since $\mathrm{DSM}_t(s^{\mathrm{emp}}) = \inf_s \mathrm{DSM}_t(s)$, Assumption 1 gives

$$\mathrm{KL}(\mathbb{P}^{\mathrm{emp}} \,\|\, \mathbb{P}^{\hat{s}}) \le \tfrac{1}{2}\, \varepsilon_{\mathrm{LN}}.$$

Let $\nu_{t_0}$ denote the time-$t_0$ marginal under $\mathbb{P}^{\hat{s}}$, while the time-$t_0$ marginal under $\mathbb{P}^{\mathrm{emp}}$ is $\mu_{\mathrm{emp}}^{\sigma_0}$ (since $\sigma_0 = \sqrt{t_0}$). Marginalization is a Markov kernel, so KL data processing yields

$$\mathrm{KL}\left(\mu_{\mathrm{emp}}^{\sigma_0} \,\|\, \nu_{t_0}\right) \le \mathrm{KL}(\mathbb{P}^{\mathrm{emp}} \,\|\, \mathbb{P}^{\hat{s}}) \le \tfrac{1}{2}\, \varepsilon_{\mathrm{LN}}. \tag{C.3}$$

**Step 3: conclude via Hellinger composition and the high-probability smoothing bound.** Because the ODE stage (8) is deterministic, $\mu_{\mathrm{DM}} = \mathrm{Proj}_{\widehat{\mathcal{M}}\#} \nu_{t_0}$. Since Hellinger distance contracts under measurable maps, we have

$$H\left( \mathrm{Proj}_{\widehat{\mathcal{M}}\#} \mu_{\mathrm{data}}^{\sigma_0}, \mu_{\mathrm{DM}} \right) = H\left( \mathrm{Proj}_{\widehat{\mathcal{M}}\#} \mu_{\mathrm{data}}^{\sigma_0}, \mathrm{Proj}_{\widehat{\mathcal{M}}\#} \nu_{t_0} \right) \le H\left( \mu_{\mathrm{data}}^{\sigma_0}, \nu_{t_0} \right).$$

By the triangle inequality for $H$,

$$H(\mu_{\mathrm{data}}^{\sigma_0}, \nu_{t_0}) \le H(\mu_{\mathrm{data}}^{\sigma_0}, \mu_{\mathrm{emp}}^{\sigma_0}) + H(\mu_{\mathrm{emp}}^{\sigma_0}, \nu_{t_0}).$$

Squaring and using $(a + b)^2 \le 2a^2 + 2b^2$ gives

$$H^2(\mu_{\mathrm{data}}^{\sigma_0}, \nu_{t_0}) \le 2H^2(\mu_{\mathrm{data}}^{\sigma_0}, \mu_{\mathrm{emp}}^{\sigma_0}) + 2H^2(\mu_{\mathrm{emp}}^{\sigma_0}, \nu_{t_0}).$$

Finally use $H^2(P, Q) \le \mathrm{KL}(P\|Q)$:

$$H^2(\mu_{\mathrm{data}}^{\sigma_0}, \mu_{\mathrm{emp}}^{\sigma_0}) \le \mathrm{KL}(\mu_{\mathrm{data}}^{\sigma_0} \,\|\, \mu_{\mathrm{emp}}^{\sigma_0}), \qquad H^2(\mu_{\mathrm{emp}}^{\sigma_0}, \nu_{t_0}) \le \mathrm{KL}(\mu_{\mathrm{emp}}^{\sigma_0} \,\|\, \nu_{t_0}).$$

Therefore,

$$H^2\left( \mathrm{Proj}_{\widehat{\mathcal{M}}\#} \mu_{\mathrm{data}}^{\sigma_0}, \mu_{\mathrm{DM}} \right) \le 2\, \mathrm{KL}(\mu_{\mathrm{data}}^{\sigma_0} \,\|\, \mu_{\mathrm{emp}}^{\sigma_0}) + 2\, \mathrm{KL}(\mu_{\mathrm{emp}}^{\sigma_0} \,\|\, \nu_{t_0}). \tag{C.4}$$

By Theorem I.1 applied at $\sigma_0^2 = t_0$, for any $a > 0$, with probability at least $1 - N^{-a}$ over the $N$ samples,

$$\mathrm{KL}(\mu_{\mathrm{data}}^{\sigma_0} \,\|\, \mu_{\mathrm{emp}}^{\sigma_0}) = \mathcal{O}\left( \frac{a \log N}{N} \right).$$

On the other hand, (C.3) holds deterministically under Assumption 1:

$$\mathrm{KL}(\mu_{\mathrm{emp}}^{\sigma_0} \,\|\, \nu_{t_0}) \leq \tfrac{1}{2}\, \varepsilon_{\mathrm{LN}}.$$

Substituting these two bounds into (C.4), we obtain that, with probability at least $1 - N^{-a}$ over the $N$ samples and any algorithmic randomness,

$$H^2\Big( \mathrm{Proj}_{\widehat{\mathcal{M}}\#} \big(\mu_{\mathrm{data}} * \mathcal{N}(0, t_0 I_D)\big) , \, \mu_{\mathrm{DM}} \Big) \;=\; \mathcal{O}\bigg( \frac{a \log N}{N} \bigg) \;+\; \mathcal{O}(\varepsilon_{\mathrm{LN}}),$$

which is exactly the claimed bound. ■

**Notations in Sections E to H** We introduce some notation that will be used extensively in the following sections.

For a function $f : \mathbb{R}^m \to \mathbb{R}$, we use $D^j[f](x)$ to denote the $j^{th}$ derivative of $f$ at $x$, provided its existence. For a function $g : \mathbb{R}^m \to \mathbb{R}^n$, $D^j[g](x)$ denotes the concatenation of the entry-wise derivatives,

$$D^j[g](x) = \big[ D^j[g_1](x), \dots, D^j[g_n](x) \big].$$

Note that $D^j[g](x)$ is a $j$-linear operator, and we denote its operator norm by $\|\cdot\|_{op}$. We abbreviate $D^1[\cdot]$ by $D[\cdot]$.

## D  FUNCTION CLASS

Let $\mathrm{supp}(\mu_{\mathrm{emp}}) = Y_N := \{y_1, \dots, y_N\}$ and recall that $\zeta_{\min}$ denotes the minimal reach over the manifold class under consideration. As in prior work, we assume that the intrinsic dimension $k$ of $\mathcal{M}^\star$ is known. For each $y_i \in Y_N$, let $W_i \in \mathbb{R}^{D \times k}$ have orthonormal columns, and suppose that $\mathrm{span}(W_i)$ approximates the tangent space $T_{y_i}\mathcal{M}^\star$ up to a *constant* angle:

$$\theta_{\max}\big(\mathrm{span}(W_i),\, T_{y_i}\mathcal{M}^\star\big) \;\leq\; 0.1\pi, \tag{D.1}$$

where $\theta_{\max}$ denotes the largest principal angle between subspaces (see Section B). Such constant-accuracy tangent estimates are standard and can be obtained, for instance, by local PCA (Aamari & Levrard, 2018). In the regime we consider, achieving this accuracy requires only a constant number of samples per anchor point.

Define the localized domain ($B_D^{\mathrm{Euc}}$ denotes the Euclidean ball)

$$\mathbb{U} \;:=\; \bigcup_{i=1}^{N} B_D^{\mathrm{Euc}}\Big(y_i; \frac{\zeta_{\min}}{2}\Big). \tag{D.2}$$

It is easy to show that $\mathbb{U}$ is connected with high probability; see Theorem G.1. For boundary points $x \in \partial\mathbb{U}$, we define the set of outward unit normals by

$$\vec{n}(x) \;:=\; \Big\{ n \in \mathbb{R}^D \,\Big|\, \exists y \in Y_N \text{ s.t. } \|x - y\| = \tfrac{\zeta_{\min}}{2} \text{ and } n = \frac{x - y}{\|x - y\|} \Big\}. \tag{D.3}$$

Fix smoothness parameters $\mathbf{L} := (L_1, \dots, L_\beta)$, and define *the distance-potential class*

$$\mathcal{D}_{\mathbf{L}}^k := \Big\{ \eta \in C^\beta(\overline{\mathbb{U}}) \,:\, \begin{aligned} &\text{(Eikonal)} &&\forall x \in \mathbb{U}, \quad \|\nabla\eta(x)\|^2 = 2\eta(x); \\ &\text{(Non-escape)} &&\exists \delta > 0, \forall x \in \partial\mathbb{U}, \forall n \in \vec{n}(x), \quad \langle \nabla\eta(x), n \rangle \geq \delta; \\ &\text{(Anchoring)} &&\forall i \in [N], \quad \eta(y_i) = 0; \\ &\text{(Rank)} &&\forall i \in [N], \quad \mathrm{rank}\big(\nabla^2\eta(y_i)\big) = D - k; \\ &\text{(Angle)} &&\forall i \in [N], \quad \theta_{\max}\big(\mathrm{span}(W_i), \ker(\nabla^2\eta(y_i))\big) \leq 0.1\pi; \\ &\text{(Smoothness)} &&\forall j \in [\beta], \quad \|\nabla^j\eta\|_{\mathrm{op}} \leq L_j. \end{aligned} \Big\}. \tag{D.4}$$

Finally, we specify the terminal-time score class as:

$$\mathcal{S} \;:=\; \Big\{ s : (0, t_0] \times \mathbb{R}^D \to \mathbb{R}^D \,\Big|\, s(x, t) = -\tfrac{1}{t}\nabla\eta(x) \text{ for } x \in \mathbb{U} \text{ with } \eta \in \mathcal{D}_{\mathbf{L}}^k,$$

$$\text{and } s(x, t) = 0 \text{ for } x \notin \mathbb{U} \Big\}. \tag{D.5}$$

**Remark.** The defining feature of $\mathcal{D}_{\mathbf{L}}^k$ is the *eikonal* constraint, which captures the geometry of the squared distance potential $\eta^\star = \frac{1}{2}\operatorname{dist}(\cdot, \mathcal{M}^\star)^2$ and hence the leading $t^{-1}$ projection term in (12). Another key ingredient is the *(Non-escape)* condition in (D.4), whose verification for $\eta^\star$ is nontrivial and is proved in Section G.5. The remaining requirements are natural: the anchoring constraints act as boundary conditions; the rank constraint enforces the intended codimension $D - k$; and the principal-angle condition holds with high probability when $W_i$ is obtained via local PCA (Aamari & Levrard, 2018). Finally, for any $k$-dimensional closed $\mathcal{C}^\beta$ embedded submanifold $\mathcal{M}^\star \subset \mathbb{R}^D$, there exists a sufficiently large constant $\mathbf{L}$ such that $\eta^\star \in \mathcal{D}_{\mathbf{L}}^k$ (see, e.g., Aamari & Levrard (2019)); we therefore fix such an $\mathbf{L}$ throughout. See Section F for details.

# E   PROOF SKETCH OF THEOREM 2.2

Theorem 2.2 is proved in a "bootstrap" fashion: We first show that the estimator $\widehat{\mathcal{M}}$ is $\mathcal{C}^{\beta-1}$, which allows us derive a similar result to Theorem 2.2 with a slightly weaker approximation guarantee (see below); This first step allows us to further show $\widehat{\mathcal{M}}$ is $\mathcal{C}^\beta$ and the approximation error is further reduced to $\mathcal{O}(\frac{1}{N^{\beta/k}})$ as in Theorem 2.2. Conditioned on that Theorem E.1 is correct, the proof of Theorem 2.2 is stated in Section H.5.

The key ingredient behind this improvement is the following nontrivial fact: if $\widehat{\mathcal{M}}$ is close to the ground-truth manifold $\mathcal{M}^\star$ in Hausdorff distance (as guaranteed by Theorem E.1), then the associated function $\hat{\eta}$ (such that $\hat{s}$ and $\hat{\eta}$ satisfy Equation (D.5)) coincides with the squared distance function to $\widehat{\mathcal{M}}$. In contrast, for a general $\eta \in \mathcal{D}_{\mathbf{L}}^k$, it is *not* true that $\eta$ is the squared distance function to its zero set, $\mathcal{M}_\eta = \{x \in \mathbb{U} : \eta(x) = 0\}$.

With this ingredient, we can then use the Poly-Raby Theorem (see for example (Denkowski, 2019, Theorem 2.14) or (Salas & Thibault, 2019, Theorem 5.1)) to show that $\widehat{\mathcal{M}}$ is $\mathcal{C}^\beta$. Once we have this enhancement, we can reuse the proof of the $\mathcal{C}^{\beta-1}$ again to obtain the improved result.

**Theorem E.1 (Weaker version of Theorem 2.2)** *Assume that $\mu_{\mathrm{data}}$ is supported on a compact, connected, boundaryless, $k$-dimensional $C^\beta$ submanifold $\mathcal{M}^\star \subset \mathbb{R}^D$ with $\beta \geq 2$, and that* $\operatorname{reach}(\mathcal{M}^\star) \geq \zeta_{\min} > 0$. *Suppose that the parameter $\mathbf{L}$ in $\mathcal{D}_{\mathbf{L}}^k$ is chosen sufficiently large such that $\eta^\star \in \mathcal{D}_{\mathbf{L}}^k$, where $\eta^\star$ is defined in Equation* (B.1). *Pick $h = \Theta((\log N/N)^{1/k})$. Let $s_{\hat{\eta}}$ be a score estimate learned from $N$ i.i.d. samples satisfying Assumption 2. For a sufficiently large $N$, the estimator $\widehat{\mathcal{M}} := \{x \in \mathbb{U} : s_{\hat{\eta}}(x, t) = 0\}$ satisfies with probability $1 - \mathcal{O}\left(\left(\frac{1}{N}\right)^{\frac{\beta}{k}}\right)$: for all $t \in (\tau, t_0]$,*

$$\mathrm{d}_{\mathcal{H}}(\widehat{\mathcal{M}}, \mathcal{M}^\star) = \tilde{\mathcal{O}}(N^{-(\beta-1)/k}), \tag{E.1}$$

*where $\tilde{\mathcal{O}}(\cdot)$ hides polylogarithmic factors in $N$ and constants depending only on $(k, D, \beta, \zeta_{\min})$.*

**Remark E.2** *The only difference (highlighted in red and bold face) of Theorem E.1 and Theorem 2.2 is that the exponent in Equation* (E.1) *is $(\beta - 1)$ instead of $\beta$ as in Equation* (15).

We now provide a more detailed proof sketch for Theorem E.1.

**Proof.** The proof consists of three main steps:

- Characterize the topological, geometrical, and analytical regularity of the set $\mathcal{M}_\eta = \{x \in \mathbb{U} : \eta(x) = 0\}$ for the functions $\eta$ in the set $\mathcal{D}_{\mathbf{L}}^k$ (D.4).

  **Topology of $\mathcal{M}_\eta$.** We first show $\mathcal{M}_\eta$ is connected with a deformation retract argument. Moreover, since every $\eta \in \mathcal{D}_{\mathbf{L}}^k$ is locally a Morse-Bott function, we can further conclude that $\mathcal{M}_\eta$ is a $\mathcal{C}^{\beta-1}$ smooth embedded submanifold of $\mathbb{R}^D$ without boundary. This result is summarized in Theorem G.3.

> We highlight that in general, for a $\mathcal{C}^\beta$ Morse-Bott function $\eta$, we can only show that its critical set, $\mathcal{M}_\eta$, is $\mathcal{C}^{\beta-1}$ submanifold. In contrast, if $\eta$ happens to be the squared distance function to $\mathcal{M}_\eta$, this can be further improved to $\mathcal{C}^\beta$, e.g. by (Denkowski, 2019, Theorem 2.14). However, at this stage, we *cannot* show that $\eta$ is a squared distance function to $\mathcal{M}_\eta$ and this is the fundamental reason why we can only have a weaker result (in the sense of regularity) in Theorem E.1.

**Geometrical Property of $\mathcal{M}_\eta$.** Our next goal is to derive a local geometric description of $\mathcal{M}_\eta$. Specifically, we show that for every point $x \in \mathcal{M}_\eta$, there exists a $D$-dimensional Euclidean open ball centered at $x$ in which $\mathcal{M}_\eta$ can be represented as the graph of a function over an open ball in $\mathbb{R}^k$. This is highly nontrivial because it requires a uniform positive lower bound on the reach of $\mathcal{M}_\eta$. The reach depends on both the curvature of the manifold and the possibility of near self-intersections. The smoothness assumption in Equation (D.4) (last line) controls the curvature, but it does not directly control near self-intersections.

To overcome this difficulty, we prove two facts. First, in a neighborhood of fixed radius around every point $x \in Y_n \subseteq \mathcal{M}^\star$, the function $\eta$ is exactly the squared distance function to $\mathcal{M}_\eta$; see Theorem G.5. Second, this implies that the same neighborhood contains no points from the medial axis of $\mathcal{M}_\eta$; see Theorem G.6. Together, these two facts yield the desired local graph representation of $\mathcal{M}_\eta$.

**Regularity of the local graph representation of $\mathcal{M}_\eta$.** Our next step is to convert the regularity and the smoothness of $\mathcal{M}_\eta$ to its local graph representation. This step is mainly built on the implicit function theorem.

- Show that, for a candidate solution $s_{\hat\eta}$ that fulfills Assumption 2, the corresponding function $\hat\eta \in \mathcal{D}_{\mathbf{L}}^k$ also minimizes a Principal Manifold Estimation (PME) loss (H.4).

- Show that when the PME loss is small for $\hat\eta$, a polynomial estimation loss (H.19) is also small.

The third statement can be converted into a bound on the Hausdorff distance between the estimated manifold $\mathcal{M}_{\hat\eta}$ and the ground-truth manifold $\mathcal{M}^\star$. Finally, since both $\mathcal{M}^\star$ and $\mathcal{M}_{\hat\eta}$ have reach bounded away from zero, this Hausdorff control can in turn be translated into closeness of the corresponding projection maps on the intersection of their tubular neighborhoods. ∎

## F   A GRAPH-OF-FUNCTION REPRESENTATION OF A SMOOTH SUBMANIFOLD

We collect here the preparatory material needed to specify the function class in Section G. For any closed (compact without boundary) $k$-dimensional $\mathcal{C}^\beta$ ($\beta \geq 2$) submanifold $\mathcal{M}$ embedded in $\mathbb{R}^D$, it admits the following representation; see Figure F.1: Let $x_{\mathrm{ref}} \in \mathcal{M}$ be any given reference point. There exist open sets $V \subseteq \mathbb{R}^D$ centered at $x_{\mathrm{ref}}$ and $U \subseteq \mathbb{R}^k$ centered at 0, such that every point $x \in V \cap \mathcal{M}$ can be represented as

$$x = \Psi(v) := x_{\mathrm{ref}} + W_{\mathrm{ref}} v + W_{\mathrm{ref}}^\perp N_{\mathrm{ref}}(v) \text{ with some } v \in U. \tag{F.1}$$

Here $W_{\mathrm{ref}} \in \mathbb{R}^{d \times k}$ is a column orthogonal matrix that spans the tangent space $T_{x_{\mathrm{ref}}}\mathcal{M}$, i.e. $T_{x_{\mathrm{ref}}}\mathcal{M}_\eta = \mathrm{span}(W_{\mathrm{ref}})$, and $W_{\mathrm{ref}}^\perp \in \mathbb{R}^{(d-k) \times k}$ is its orthogonal complement; $N_{\mathrm{ref}} : \mathbb{R}^k \to \mathbb{R}^{d-k}$ is locally a $\mathcal{C}^\beta$ function and $(v, N_{\mathrm{ref}}(v)) \in \mathbb{R}^k \times \mathbb{R}^{d-k}$ is the coordinate of $x$ under the basis $(W_{\mathrm{ref}}, W_{\mathrm{ref}}^\perp)$. Moreover, $N_{\mathrm{ref}}$ admits the following conditions

$$N_{\mathrm{ref}}(0) = 0 \qquad \text{and} \qquad D[N_{\mathrm{ref}}](0) = 0, \tag{F.2}$$

where $v = 0$ corresponds to the point $x_{\mathrm{ref}}$ in the chosen chart. The first condition ensures that $\mathcal{M}$ passes through $x_{\mathrm{ref}}$, and the second ensures that the tangent space of $\mathcal{M}$ at $x_{\mathrm{ref}}$ is exactly $\mathrm{span}(W_{\mathrm{ref}})$.

Further, by compactness, for a fixed $\mathcal{C}^2$ submanifold, we have (1) its reach is bounded from below and (2) for any $x_{\mathrm{ref}} \in \mathcal{M}^\star$, the operators of $D^j[N_{\mathrm{ref}}]$ are bounded from above within the open domain $V$. To derive concrete statistical complexity bounds for submanifold recovery, we follow the previous work (Aamari & Levrard, 2019) and specify these bounds as follow.

**Definition F.1** *For $\beta \geq 3$, $\zeta_{\min} > 0$, and $\mathbf{L} := (L_2, L_3, \ldots, L_\beta)$, let $\mathcal{C}_{\zeta_{\min}, \mathbf{L}}^\beta$ be the class of $k$-dimensional closed submanifolds $\mathcal{M} \subset \mathbb{R}^D$ such that:*

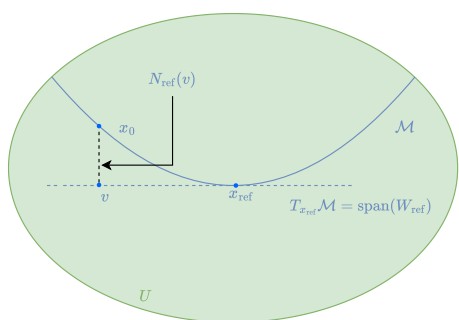

Figure F.1: A local representation of a submanifold $\mathcal{M} \in \mathcal{C}^\beta$.

- *Reach condition:* $\operatorname{reach}(\mathcal{M}) \geq \zeta_{\min}$.

- *Local graph representation: For every $x_{\mathrm{ref}} \in \mathcal{M}$, there exists a radius $r \geq \frac{1}{4L_2}$, an open set $V \subseteq \mathbb{R}^D$, and a $\mathcal{C}^\beta$ map $N_{\mathrm{ref}} : B_k(0, r) \to \mathbb{R}^{D-k}$ such that $\mathcal{M} \cap V$ admits a one-to-one parametrization*

$$\Psi : B_k(0, r) \to \mathcal{M} \cap V, \qquad \Psi \text{ as in Equation (F.1) with } N_{\mathrm{ref}}.$$

- *Derivative bounds: For every $v \in \mathbb{R}^k$ with $|v| \leq \frac{1}{4L_2}$ and every $2 \leq j \leq \beta$, $\|D^j N_{\mathrm{ref}}(v)\|_{\mathrm{op}} \leq L_j$.*

*Here $D^j \phi(v)$ denotes the $j$th derivative of a map $\phi : \mathbb{R}^k \to \mathbb{R}^{D-k}$ at $v$, viewed as a $j$-linear form, and $\|\cdot\|_{\mathrm{op}}$ is the associated operator norm.*

We note that the submanifold class is exactly the same as the one considered in (Aamari & Levrard, 2019, Definition 1) and hence the lower bounds in (Aamari & Levrard, 2019, Theorems 3, 5, 7) also apply here.

## G   AUXILIARY DETAILS FOR THE FUNCTION CLASS CONSTRUCTION

This section records additional details underlying the construction of the function class used in Section 2.3.

**Connectedness of $\mathbb{U}$.**   Recall that in Section 2.3 we let

$$\operatorname{supp}(\mu_{\mathrm{emp}}) = Y_N = \{y_1, \ldots, y_N\} \subseteq \mathcal{M}^\star.$$

Set

$$h = \tilde{\mathcal{O}}\left(\left(\frac{\log N}{N}\right)^{1/k}\right).$$

A standard covering argument implies that, for $N$ sufficiently large, $Y_N$ is an $\epsilon$-net of the target manifold $\mathcal{M}^\star$ with $\epsilon = h/2$, with probability at least $1 - N^{-\beta/k}$; see, e.g., (Aamari & Levrard, 2019, Lemma 4).

Recall the definition of $\mathbb{U}$ from (D.2):

$$\mathbb{U} := \bigcup_{i=1}^{N} B_D^{\mathrm{Euc}}\left(y_i; \frac{\zeta_{\min}}{2}\right), \tag{G.1}$$

where $\zeta_{\min}$ denotes the minimal reach over the manifold class under consideration. By construction, $\mathbb{U} \subseteq \mathbb{R}^D$ is a neighborhood of $\mathcal{M}^\star$. The next lemma records the basic topological and geometric properties of $\mathbb{U}$.

**Lemma G.1 (Connectivity and minimum width of $\mathbb{U}$)** *Suppose that $Y_N$ is an $\epsilon$-net of $\mathcal{M}^\star$ (in the ambient Euclidean metric) for some $\epsilon < \zeta_{\min}/2$. Then $\mathbb{U}$ is connected. Moreover, $\mathbb{U}$ contains the tubular neighborhood of $\mathcal{M}^\star$ of radius $\zeta_{\min}/2 - \epsilon$, i.e.,*

$$\mathcal{T}_{\zeta_{\min}/2-\epsilon}(\mathcal{M}^\star) = \{x \in \mathbb{R}^D : \text{dist}(x, \mathcal{M}^\star) \leq \zeta_{\min}/2 - \epsilon\} \subseteq \mathbb{U}.$$

Please find the proof in Section G.4.1. In the rest of this section, we will justify the construction of the function class $\mathcal{D}_\mathbf{L}^k$ (D.4) by showing every member function $\eta \in \mathcal{D}_\mathbf{L}^k$ is "distance-like"; Moreover, on a subset of $\mathbb{U}$, $\eta$ is exactly a distance function to some embedded submanifold.

- We first consider a superset of $\mathcal{D}_\mathbf{L}^k$: With only the Eikonal equation and the non-escape boundary condition (first and second lines in Equation (D.4)), define

$$\mathcal{D} := \{\eta \in \mathcal{C}^\beta(\bar{\mathbb{U}}) \mid \quad \forall x \in \mathbb{U}, \|\nabla\eta(x)\|^2 = 2\eta(x); \quad \text{(Eikonal equation)}$$
$$\exists\delta > 0, \forall x \in \partial\mathbb{U}, \forall n \in \vec{n}(x), \nabla\eta(x) \cdot n > \delta\},$$
$$\text{(Boundary barrier)}$$

  where we recall the definition of the outward normal set $\vec{n}(x)$ in Equation (D.3). We show in Section G.1 that all members of the above function class are distance-like functions: For every $\eta \in \mathcal{D}$, define $\mathcal{M}_\eta = \{x \in \mathbb{U} \mid \eta(x) = 0\}$.

  - $\mathcal{M}_\eta$ is a connected closed smooth embedded submanifold of $\mathbb{R}^D$;
  - $\eta(x) = \frac{1}{2}d_\mathbb{U}^2(x, \mathcal{M}_\eta)$, where

$$d_\mathbb{U}(x, \mathcal{M}_\eta) := \inf\Big\{\text{Length}(\alpha) \mid \alpha : [0,1] \to \mathbb{U} \text{ absolutely continuous,}$$
$$\alpha(0) = x, \ \alpha(1) \in \mathcal{M}_\eta\Big\}. \tag{G.2}$$

  Further, consider the following open set (half-size to $\mathbb{U}$)

$$\mathbb{U}_2 := \bigcup_{i=1}^N B_D^{\text{Euc}}\Big(y_i; \frac{\zeta_{\min}}{4}\Big). \tag{G.3}$$

  - We show that for $x \in \mathbb{U}_2$, $d_\mathbb{U}(x, \mathcal{M}_\eta) \equiv \text{dist}(x, \mathcal{M}_\eta)$.
  - Built on this result, and together with the feature ball lemma (Dey, 2006, Lemma 1.1), we show that for any $D$-dimensional ball $U \subseteq \mathbb{U}_2$, $U \cap \mathcal{M}_\eta$ has at most only one connected component.

We then show that $\mathcal{M}_\eta$ can be locally represented as the graph of a function, as discussed in Section F. This is useful for our later derivations.

- With the further anchoring constraint, rank constraint, and subspace angle constraint (third, fourth, fifth lines in Equation (D.4)), we show in Section G.2 that for every $\eta \in \mathcal{D}_\mathbf{L}^k$, the dimension of its zero set $\mathcal{M}_\eta := \{x \in \mathbb{U} \mid \eta(x) = 0\}$ is $k$ and locally, it admits a representation as discussed in Section F, i.e. the graph of a function over a ball in $\mathbb{R}^k$.

- With the smoothness constraint (last line in Equation (D.4)), we show in Section G.3 that the graph-of-function representation of $\mathcal{M}_\eta$ has nice regularity properties; i.e., the derivatives of the corresponding local function are bounded in terms of operator norm up to order $\beta - 1$.

## G.1 Distance-like function class restricted on $\mathbb{U}$

**Lemma G.2 (Global-in-time existence of gradient flow under (Boundary barrier))** *Recall the definition of $\mathbb{U}$ in Equation (D.2). For $\eta \in \mathcal{D}$, consider the negative gradient flow*

$$\dot{x}(t) = -\nabla\eta(x(t)), \qquad x(0) = x_0 \in \mathbb{U}. \tag{G.4}$$

*Then a unique global solution exists and $x(t) \in \mathbb{U}$ for all $t \geq 0$.*

Please find the proof in Section G.4.2. Built on the above result, we can identify the manifold structure of the zero set of any $\eta \in \mathcal{D}$.

**Lemma G.3** *For any function $\eta \in \mathcal{D}$, define $\mathcal{M}_\eta := \{x \in \mathbb{U} \mid \eta(x) = 0\}$. We have that $\mathcal{M}_\eta \neq \emptyset$ and it is a closed connected $\mathcal{C}^{\beta-1}$ smooth embedded submanifold of $\mathbb{R}^D$.*

Please find the proof in Section G.4.3. Note that we cannot determine the dimension of $\mathcal{M}_\eta$ with only the requirements in $\mathcal{D}$, and further assumptions like the rank constraint (fourth line in Equation (D.4)) are needed for that purpose.

**Theorem G.4 (Classical eikonal solution equals the distance to $\mathcal{M}_\eta$)** *Recall the definition of $\mathbb{U}$ in Equation (D.2). For any $\eta \in \mathcal{D}$, define $\mathcal{M}_\eta := \{x \in \mathbb{U} \mid \eta(x) = 0\}$, which from Theorem G.3 we know is an embedded smooth submanifold. Recall the definition of $d_\mathbb{U}(\cdot, \mathcal{M}_\eta)$ in Equation (G.2). We have*

$$\eta(x) = \frac{1}{2} d_\mathbb{U}(x, \mathcal{M}_\eta)^2 \qquad \forall x \in \mathbb{U}.$$

Please find the proof in Section G.4.4. Moreover, we show that on $\mathbb{U}_2$, a smaller neighborhood of $Y_n$, $d_\mathbb{U}(\cdot, \mathcal{M}_\eta)$ identifies with $\mathrm{dist}(\cdot, \mathcal{M}_\eta)$.

**Lemma G.5** *On $\mathbb{U}_2$, we have $d_\mathbb{U}(\cdot, \mathcal{M}_\eta) = \mathrm{dist}(\cdot, \mathcal{M}_\eta)$.*

**Proof.** For any point $x \in \mathbb{U}_2$, by definition, there exists $y \in Y_n$ such that $\|x - y\| \leq \zeta_{\min}/4$. We clearly have $\mathrm{dist}(x, \mathcal{M}_\eta) \leq \|x - y\| \leq \zeta_{\min}/4$, since we also have $y \in \mathcal{M}_\eta$ (anchoring constraint). Consequently, we have

$$\pi_\eta(x) \in B_D^{\mathrm{Euc}}(y; \frac{\zeta_{\min}}{2}) \subseteq \mathbb{U}. \tag{G.5}$$

Now both $x$ and $\pi_\eta(x)$ are in $B_D^{\mathrm{Euc}}(y; \frac{\zeta_{\min}}{2}) \subseteq \mathbb{U}$, and note that $B_D^{\mathrm{Euc}}(y; \frac{\zeta_{\min}}{2})$ is a convex set. So the whole line segment between $x$ and $\pi_\eta(x)$ is in $B_D^{\mathrm{Euc}}(y; \frac{\zeta_{\min}}{2}) \subseteq \mathbb{U}$. Consequently, $d_\mathbb{U}(x, \mathcal{M}_\eta) = d(x, \mathcal{M}_\eta)$ for any point $x \in \mathbb{U}_2$. ∎

**Lemma G.6** *For any open ball $U \subseteq \mathbb{U}_2$ and any $\eta \in \mathcal{D}$, we have that $U \cap \mathcal{M}_\eta$ has at most one connected component.*

**Proof.** We prove by contradiction. Suppose that there exists an open ball $U \subseteq \mathbb{U}_2$ such that $U \cap \mathcal{M}_\eta$ has at least two connected components. Use $k$ to denote the dimension of $\mathcal{M}_\eta$. Clearly, $U$ intersects with $\mathcal{M}_\eta$ at least two points. Moreover, since $U \cap \mathcal{M}_\eta$ is not connected, it is not homeomorphic to a ball in $\mathbb{R}^k$. By the feature ball lemma (Dey, 2006, Lemma 1.1), there exists a medial axis point in $U$. However, since $U \subseteq \mathbb{U}_2$, by Theorem G.5, $\eta = \frac{1}{2} \mathrm{dist}^2(\cdot, \mathcal{M}_\eta)$ is non-differentiable at this medial axis point (since the projection onto $\mathcal{M}_\eta$ is not unique). However, since for any $\eta \in \mathcal{D}$, $\eta \in \mathcal{C}^\beta(\mathbb{U})$, we have a contradiction. ∎

## G.2 Graph-of-function Representation of $\mathcal{M}_\eta$ on a Local Patch

Theorem G.3 shows that, for any $\eta \in \mathcal{D}$ (a superset of $\mathcal{D}_\mathbf{L}^k$), $\mathcal{M}_\eta := \{x \in \mathbb{U} \mid \eta(x) = 0\}$ is a smooth submanifold. The rank constraint (fourth lines in Equation (D.4)) specifies the dimension of the zero set for $\eta \in \mathcal{D}_\mathbf{L}^k$.

**Lemma G.7** *For every $\eta \in \mathcal{D}_L^k$, its zero set $\mathcal{M}_\eta := \{x \in \mathbb{U} \mid \eta(x) = 0\}$ is a $k$-dimensional connected closed $\mathcal{C}^{\beta-1}$ smooth embedded submanifold.*

Recall the extra anchoring constraint (third line in Equation (D.4)) in $\mathcal{D}_\mathbf{L}^k$. According to Section F, in the neighborhood of every anchoring point $x_{\mathrm{ref}} \in Y_n$, we can represent $\mathcal{M}_\eta$ locally as the graph of a function over a ball in the tangent space $T_{x_{\mathrm{ref}}} \mathcal{M}_\eta \subseteq \mathbb{R}^k$. Please see Figure G.1 for an example.

**Remark G.8** *We highlight that this is a non-trivial result since we do not make assumptions on the reach of $\mathcal{M}_\eta$: To establish the graph-of-function representation of $\mathcal{M}_\eta$ in Theorem F.1, we need to rule out the case where, for some $\eta \in \mathcal{D}$, $\mathcal{M}_\eta$ is almost self-intersecting. Since otherwise, it*

*is possible that for any $d$-dimensional ball $U$ with a fixed radius, there exists some $\eta \in \mathcal{D}$ where $U \cap \mathcal{M}_\eta$ could have two disconnected components. While this worst case scenario can be naturally avoided by a global reach lower bound, we manage to exclude it in Theorem G.6 even without making such a strong reach assumption.*

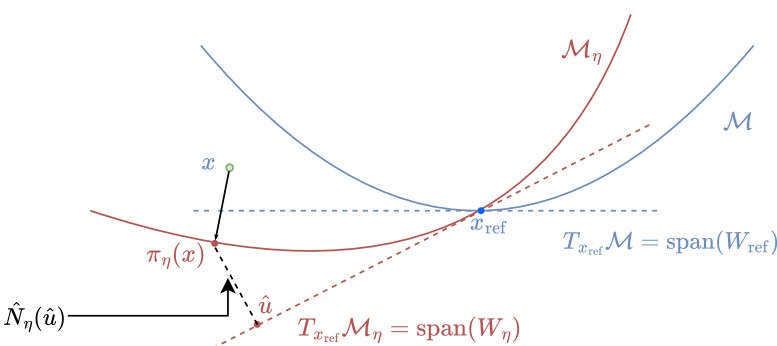

Figure G.1: Understanding the hypothesis score function class $\{s_\eta\}$ in the local coordinate: i) pick a reference point $x_{\mathrm{ref}} \in \mathcal{M}^\star$; ii) any $k$-dimensional $\mathcal{C}^{\beta-1}$ submanifold $\mathcal{M}_\eta$ passing $x_{\mathrm{ref}}$ can be parameterized by $[W_\eta, \hat{N}_\eta]$ in the sense that for all $\hat{x} \in \mathcal{M}_\eta \cap B_D^{\mathrm{Euc}}(x_{\mathrm{ref}}, h)$, there exists a unique coordinate $(\hat{u}, \hat{N}_\eta(\hat{u}))$ under the basis $(W_\eta, W_\eta^\perp)$; iii) for any $x \in B_D^{\mathrm{Euc}}(x_{\mathrm{ref}}, h)$, the projection onto $\mathcal{M}_\eta$ is unique, denoted by $\pi_\eta(x)$; iv) the score function indexed by $\eta$ can be written as $s_\eta(t, x) := -\frac{x - \pi_\eta(x)}{t}$ for $x \in B_D^{\mathrm{Euc}}(x_{\mathrm{ref}}, h)$.

For each $\eta$, let $W_\eta \in \mathbb{R}^{d \times k}$ be a column-orthonormal matrix whose columns span the tangent space $T_{x_{\mathrm{ref}}}\mathcal{M}_\eta$. Let $W_\eta^\perp \in \mathbb{R}^{d \times (d-k)}$ denote an orthonormal complement, and let $\hat{N}_\eta : \mathbb{R}^k \to \mathbb{R}^{d-k}$ be a polynomial map of total degree at most $\beta - 1$. As discussed in Section F, for a sufficiently small chart neighborhood $V$ around $x_{\mathrm{ref}}$, every point $\hat{x} \in \mathcal{M}_\eta \cap V$ admits the representation[5]

$$\hat{x} = x_{\mathrm{ref}} + W_\eta \hat{u} + W_\eta^\perp \hat{N}_\eta(\hat{u}), \qquad \hat{u} \in \mathbb{R}^k. \tag{G.6}$$

The anchoring and tangency at $x_{\mathrm{ref}}$ impose the normalization conditions (see Equation (F.2))

$$\hat{N}_\eta(0) = 0 \qquad \text{and} \qquad D[\hat{N}_\eta](0) = 0, \tag{G.7}$$

where $\hat{u} = 0$ corresponds to the point $x_{\mathrm{ref}}$ in the chosen chart.

### G.2.1 CHANGE OF BASIS

For the subsequent analysis, it is more convenient to re-express the same local patch around $x_{\mathrm{ref}}$ in the *unknown* ground-truth basis $(W_{\mathrm{ref}}, W_{\mathrm{ref}}^\perp)$, where $T_{x_{\mathrm{ref}}}\mathcal{M}^\star = \mathrm{span}(W_{\mathrm{ref}})$.

Accordingly, given a point $\hat{x} \in \mathcal{M}_\eta$ around $x_{\mathrm{ref}}$, we derive its coordinates $(u, N_\eta(u))$ under the ground-truth basis $(W_{\mathrm{ref}}, W_{\mathrm{ref}}^\perp)$ from its coordinates $(\hat{u}, \hat{N}_\eta(\hat{u}))$ under the hypothesis basis $(W_\eta, W_\eta^\perp)$. This change of coordinates implicitly defines a new function $N_\eta : \mathbb{R}^k \to \mathbb{R}^{d-k}$, which will be the object used in our analysis. Concretely, for any $\hat{x} \in \mathcal{M}_\eta \cap B_D^{\mathrm{Euc}}(x_{\mathrm{ref}}, h)$, we can represent it under both bases

$$\hat{x} = x_{\mathrm{ref}} + W_\eta \hat{u} + W_\eta^\perp \hat{N}_\eta(\hat{u}) = x_{\mathrm{ref}} + W_{\mathrm{ref}} u + W_{\mathrm{ref}}^\perp N_\eta(u). \tag{G.8}$$

Define two functions $F : \mathbb{R}^k \to \mathbb{R}^k$ and $G : \mathbb{R}^k \to \mathbb{R}^{d-k}$

$$F(\hat{u}) = W_{\mathrm{ref}}^\top W_\eta \hat{u} + W_{\mathrm{ref}}^\top W_\eta^\perp \hat{N}_\eta(\hat{u}) \text{ and } G(\hat{u}) = W_{\mathrm{ref}}^{\perp \top} W_\eta \hat{u} + W_{\mathrm{ref}}^{\perp \top} W_\eta^\perp \hat{N}_\eta(\hat{u}). \tag{G.9}$$

---

[5]Since we assume that, for every $\eta \in \mathcal{D}_{\mathbf{L}}^k$, the operator norms of its derivatives are uniformly bounded, it follows that the size of $V$ is bounded below by a positive constant. We provide the details below.

Multiplying both sides of Equation (G.8) by $W_{\text{ref}}^\top$ and $(W_{\text{ref}}^\perp)^\top$ yields two equations

$$F(\hat{u}) = u \text{ and } G(\hat{u}) = N_\eta(u). \tag{G.10}$$

We highlight that the subspace constraint (fifth line in Equation (D.4)) ensures that $F$ is invertible locally around $\hat{u} = 0$ ($\hat{u} = 0$ corresponds to the point $x_{\text{ref}}$), and hence one can locally write

$$N_\eta(u) = [G \circ F^{-1}](u). \tag{G.11}$$

We make the above derivation rigorous using the inverse function theorem. We highlight that $N_\eta$ is only used in the analysis. It is *not* practically available as it involves $W_{\text{ref}}$, which is unknown.

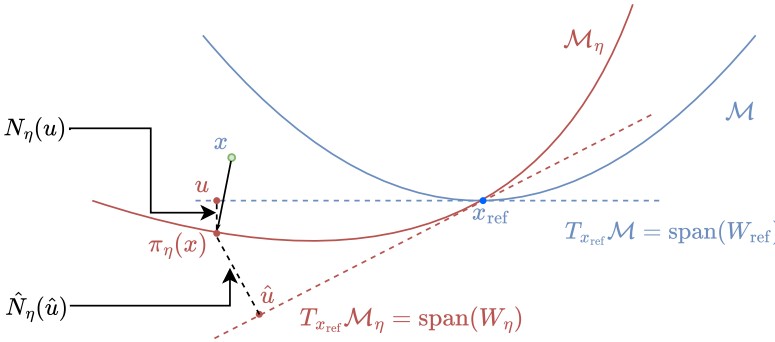

Figure G.2: Change of basis. For any point $\hat{x} \in \mathcal{M}_\eta \cap B_D^{\text{Euc}}(x_{\text{ref}}, h)$, use $(\hat{u}, \hat{N}_\eta(\hat{u}))$ and $(u, N_\eta(u))$ to denote its coordinates under the bases $(W_\eta, W_\eta^\perp)$ and $(W_{\text{ref}}, W_{\text{ref}}^\perp)$ respectively. When $\sigma_{\min}(W_\eta^\top W_{\text{ref}}) > 0$, for a sufficiently small $h > 0$, one can identify $N_\eta$ with $\hat{N}_\eta$ up to a diffeomorphism.

**Theorem G.9 (Change of basis)** *Let $\eta \in \mathcal{D}_L^k$. Recall the expressions of $F$ and $G$ in Equation* (G.10). *Under this condition, for a sufficiently small $h$ and for all $u \in B_k(0, h)$, the function $F$ defined in Equation* (G.10) *is invertible and the local coordinate function $N_\eta$ under the basis $[W, W^\perp]$ writes*

$$N_\eta(u) = [G \circ F^{-1}](u). \tag{G.12}$$

*Further, one has that*

- *The minimum eigenvalue of $D[F](0)$ is lower bounded by $\sqrt{1 - \sin^2(0.2\pi)} > 0$.*

- *The Jacobian of $N_\eta$ is given by*

$$D[N_\eta](u) = D[G](F^{-1}(u))D[F^{-1}](u) = D[G](F^{-1}(u))(D[F](F^{-1}(u)))^{-1}, \tag{G.13}$$

$$D[G](\hat{u}) = W_{\text{ref}}^{\perp \top} W_\eta + W_{\text{ref}}^{\perp \top} W_\eta^\perp D[\hat{N}_\eta](\hat{u}) \tag{G.14}$$

$$D[F](\hat{u}) = W_{\text{ref}}^\top W_\eta + W_{\text{ref}}^\top W_\eta^\perp D[\hat{N}_\eta](\hat{u}). \tag{G.15}$$

*Moreover, the first-order Taylor expansion of $N_\eta$ around $0$ is*

$$N_\eta(v) = N_\eta(0) + D[N_\eta](0)v + \mathcal{O}(\|v\|^2),$$

*where we have*

$$N_\eta(0) = 0 \text{ and } D[N_\eta](0) = W_{\text{ref}}^{\perp \top} W_\eta \left(W_{\text{ref}}^\top W_\eta\right)^{-1}. \tag{G.16}$$

- *$N_\eta \in \mathcal{C}^{\beta-1}$ and the operator norms of the derivatives of $N_\eta$ up to order $\beta - 1$ is bounded in $B_k(0, h)$.*

**Proof.** We prove this result using the implicit function theorem (Krantz & Parks, 2002, Theorem 3.3.1). We highlight that the subspace angle constraint (fifth line in Equation (D.4)) plays a key role in establishing the invertibility of $F$.

Clearly, to show the existence of $N_\eta$ as defined in Equation (G.12), we only need to show the existence of $F^{-1}$. Following the notation of (Krantz & Parks, 2002, Theorem 3.3.1), set

$$\Phi(u, \hat{u}) = u - F(\hat{u}). \tag{G.17}$$

If we can verify that $D_{\hat{u}}\Phi$ is invertible around 0, we have the existence of $F^{-1}$ around 0 and moreover we can explicitly write down its Jacobian by the implicit function theorem. To this end, recall that $D[\hat{N}_\eta](0) = 0$ by construction; see Equation (G.7). We can hence calculate

$$D_{\hat{u}}[\Phi](0) = -W_{\mathrm{ref}}^\top W_\eta.$$

Use $\sigma_{\min}$ and $\sigma_{\max}$ to denote the minimum and maximum singular value of a matrix. Note that by (Ji-Guang, 1987, Theorem 2.1)

$$\sigma_{\min}(W_{\mathrm{ref}}^\top W_\eta) = \cos\theta_{\max}(\mathrm{span}(W_{\mathrm{ref}}), \mathrm{span}(W_\eta)) = \sqrt{1 - \sin^2\theta_{\max}(\mathrm{span}(W_{\mathrm{ref}}), \mathrm{span}(W_\eta))}$$

where we recall that $\theta_{\max}$ denotes the largest principal angle between two subspaces. Hence by the subspace angle constraint (fifth line in Equation (D.4)), we have that singluar values of $D_{\hat{u}}[\Phi](0)$ are lower bounded by $\sqrt{1 - \sin^2 0.1\pi} > 0$ and hence $D_{\hat{u}}[\Phi](0)$ is invertible. ∎

### G.3 REGULARITY OF THE GRAPH-OF-FUNCTION REPRESENTATION

To derive a concrete statistical complexity bound, we need to ensure that the operator norms of the derivatives of $N_\eta$ are bounded by some constant. In the following, we show in Theorem G.10 that the smoothness constraint (last line in Equation (D.4)) can be used to bound the operator norms of $D^j \hat{N}_\eta$ (defined in Equation (G.6)), which in turn bounds the operator norms of $D^j N_\eta$ (defined in Equation (G.12)), as shown in Theorem G.11.

**Lemma G.10** *Let $\eta \in \mathcal{D}_L^k$. Denote its zero set $\mathcal{M}_\eta = \{x \in \mathbb{U} \mid \eta(x) = 0\}$. Let $x_{\mathrm{ref}} \in \mathcal{M}^\star$ be any fixed reference point and recall the definition of $\hat{N}_\eta$ in Equation (G.6). By Theorem G.5, we have that $\nabla\eta(x) = x - \pi_\eta(x)$ on $\mathbb{U}_2$, where $\pi_\eta$ denotes the projection onto $\mathcal{M}_\eta$. Then, we have that $\hat{N}_\eta \in \mathcal{C}^{\beta-1}$, and for each $j \in \{2, \ldots, \beta - 1\}$ there exists constants $\hat{\mathbf{L}} = (\hat{L}_2, \ldots, \hat{L}_j, \ldots, \hat{L}_{\beta-1})$ that only depends on $(k, D, j, \mathbf{L})$ such that, for all $h$ below a constant threshold*

$$\forall u \in B_k(0, h), \ \|D[\hat{N}_\eta](u)\|_{\mathrm{op}} = \hat{L}_1 h \ \text{and} \ \|D^j[\hat{N}_\eta](u)\|_{\mathrm{op}} \leq \hat{L}_j. \tag{G.18}$$

Please find the proof in Section G.4.5.

**Lemma G.11** *Let $\eta \in \mathcal{D}_L^k$. Denote its zero set $\mathcal{M}_\eta = \{x \in \mathbb{U} \mid \eta(x) = 0\}$. For any $x_{\mathrm{ref}} \in Y_n$, recall the graph-of-function representation of $\mathcal{M}^\star \cap B_D^{\mathrm{Euc}}(x_{\mathrm{ref}}, h)$ under the unknown ground truth basis in Equation (G.8) and the definition of $N_\eta$ in Equation (G.12). For $h$ below some constant threshold, we have the following results:*

- *For all $\hat{u} \in B_k(0, h)$, we have $\sigma_{\min}(DF(\hat{u})) \geq m$ for some universal constant $m > 0$ and hende $DF(\hat{u})$ is invertible on $B_k(0, h)$.*

- *There exists some $\mathbf{L}' = (L_1', L_2', \ldots, L_\beta') = \mathbf{L}'(\mathbf{L}, j, d, D)$ such that*

$$\|D^j N_\eta(0)\|_{\mathrm{op}} \leq L_j'. \tag{G.19}$$

**Proof.** Recall the definition of $N_\eta$ in Equation (G.12). First, we show that for all $\hat{u} \in B_k(0, h)$, the matrix $D[F](\hat{u})$ is invertible.

**Invertibility of $D[F](\hat{u})$.** To see this, calculate that

$$D[F](\hat{u}) = W_{\text{ref}}^\top W_\eta + W_{\text{ref}}^\top W_\eta^\perp D[\hat{N}_\eta](\hat{u}),$$

and hence we can bound

$$\|D[F](\hat{u})\|_{op} \geq \|W_{\text{ref}}^\top W_\eta\|_{op} - \|W_{\text{ref}}^\top W_\eta^\perp D[\hat{N}_\eta](\hat{u})\|_{op}.$$

From Theorem G.9, we know that $\|W_{\text{ref}}^\top W_\eta\|_{op} = \|D[F](0)\|_{op} \geq \sqrt{1 - \sin^2 0.2\pi}$. Moreover, from Theorem G.10, we know that $\|W_{\text{ref}}^\top W_\eta^\perp D[\hat{N}_\eta](\hat{u})\|_{op} = \mathcal{O}(h)$. All together, for all $u \in B_k(0, h)$, we have that $\|D[F](\hat{u})\|_{op}$ is bounded from below by some universal constant $m$.

By the inverse function theorem, we can ensure that the operator norms of the derivative of $F^{-1}$ can be bounded by the operator norms of the derivative of $F$. Consequently, by the chain rule of composition, the operator norms of $D^j N_\eta$ can be bounded by that of $D^j \hat{N}_\eta$. Together with Theorem G.10 and the smoothness constraint (last line in Equation (D.4)), we have the result. ∎

### G.4  Proofs of Section G

#### G.4.1  Proof of Theorem G.1

**Proof.    Connectedness of $\mathbb{U}$.** We first show that $\mathcal{M}^\star \subseteq \mathbb{U}$: Since $Y_n$ is an $\epsilon$-net of $\mathcal{M}^\star$, for any $x \in \mathcal{M}^\star$, there exists $y \in Y_n$ such that $\|x - y\| \leq \epsilon < \zeta_{\min}/2$. Hence $x \in B_D^{\text{Euc}}(y, \zeta_{\min}/2) \subseteq U$. Consequently, a connected path between any two points $x_1 \in B_D^{\text{Euc}}(y_1, \zeta_{\min}/2)$ and $x_2 \in B_D^{\text{Euc}}(y_2, \zeta_{\min}/2)$ in $\mathbb{U}$ can be constructed as first connect $x_i$ with $y_i$, $i = 1, 2$, and connect $y_1$ and $y_2$ through $\mathcal{M}^\star$.

**Inclusion of a tubular neighborhood of $\mathcal{M}^\star$.** Consider any point $x$ in the tubular neighborhood of $\mathcal{M}^\star$ with radius $(\zeta_{\min}/2 - \epsilon)$, the projection of $x$ onto $\mathcal{M}^\star$ is unique. We denote this point by $\pi(x)$. Since $Y_n$ is an $\epsilon$-net of $\mathcal{M}^\star$, there exists some $y \in Y_n$ such that $\|\pi(x) - y\| \leq \epsilon$. By triangle inequality, one has

$$\|x - y\| \leq \|\pi(x) - x\| + \|\pi(x) - y\| \leq \zeta_{\min}/2 \Rightarrow x \in B_D^{\text{Euc}}(y, \zeta_{\min}/2) \subseteq \mathbb{U}.$$

∎

#### G.4.2  Proof of Theorem G.2

**Proof.** Recall the definition of $U$ in Equation (D.2). Define

$$g_i(x) := \frac{\zeta_{\min}}{2} - |x - y_i|, \qquad b(x) := \max_{1 \leq i \leq N} g_i(x), \tag{G.20}$$

where $y_i \in Y_n$.

Assume for contraction the gradient flow (G.4) hits the boundary of $\partial U$ in finite time, i.e.

$$T = \inf\{t > 0 \mid x(t) \notin U\} < \infty. \tag{G.21}$$

Denote $x_* = x(T)$ and use $I$ to denote the active set at $x_*$, i.e. the indices that $\|x_* - y_i\| = \frac{\zeta_{\min}}{2}$. The corresponding outward normal vector at $x \neq y_i$ is denoted by

$$n_i(x) = \frac{x - y_i}{\|x - y_i\|} \tag{G.22}$$

Pick any $i \in I$, and define along the trajectory

$$h_i(t) := g_i(x(t)) = r_i - |x(t) - y_i|.$$

Each $h_i$ is $\mathcal{C}^1$ on ([0,T]), and

$$\dot{h}_i(t) = \langle \nabla g_i(x(t)), \dot{x}(t)\rangle = \left\langle -\frac{x(t) - y_i}{|x(t) - y_i|}, -\nabla\eta(x(t))\right\rangle = \langle n_i(x(t)), \nabla\eta(x(t))\rangle. \tag{G.23}$$

By the continuity of $n_i$ and $\nabla \eta$ (w.r.t. $x$), there exists a radius $\rho$ such that for all $x \in \overline{U} \cap B(x_*, \rho)$,

$$\langle n_i(x), \nabla \eta(x) \rangle \geq \frac{\delta}{2}.$$

Since $x(t) \to x_*$ as $t \uparrow T$, there exists $\tau \in (0, T)$ such that

$$x(t) \in B(x_*, \rho) \quad \text{for all } t \in [T - \tau, , T].$$

Consequently, for all $t \in [T - \tau, T]$,

$$\dot{h}_i(t) \geq \frac{\delta}{2}.$$

Since $i \in I$, we have $h_i(T) = g_i(x_*) = 0$. Integrating (G.23) from $t$ to $T$ gives

$$0 - h_i(t) = h_i(T) - h_i(t) = \int_t^T \dot{h}_i(s) ds \geq \int_t^T \frac{\delta}{2} ds = \frac{\delta}{2}(T - t),$$

hence

$$h_i(t) \leq -\frac{\delta}{2}(T - t) < 0 \qquad \forall t \in [T - \tau, T).$$

So $T$ is not the first hitting time of $x(t)$ on $\partial U$, which contradicts with the definition of $T$. ∎

### G.4.3 PROOF OF THEOREM G.3

**Proof.** Since $\eta$ is a continuous function and $\mathbb{U}$ is compact, $\mathcal{M}_\eta$ is compact. Moreover, we have from (Boundary barrier)

$$\mathcal{M}_\eta \cap \partial \mathbb{U} = \emptyset.$$

Moreover, by the (Eikonal equation), $\eta \geq 0$. Hence (Eikonal equation) also implies that $\eta$ satisfies the Polyak-Łojasiewicz (PL) inequality on $\mathbb{U}$. Consider the negative gradient flow (G.4). Theorem G.2 shows that it exists globally in time. Moreover, note that

$$\frac{\mathrm{d}}{\mathrm{d}t} \eta(x(t)) = -\|\nabla \eta(x(t))\|^2 = -2\eta(x(t)) \Rightarrow \eta(x(t)) \to 0 \text{ as } t \to 0. \tag{G.24}$$

By the PL inequality, $x(t)$ is convergent. Moreover, we have $x(\infty) \notin \partial \mathbb{U}$ since otherwise $\nabla \eta(x(\infty)) = 0$ which contradicts with (Boundary barrier). Consequently, $x(\infty) \in \mathcal{M}_\eta \neq \emptyset$.

**Manifold structure of $\mathcal{M}_\eta$.** Let $\mathcal{M}_1$ be an arbitrary connected component in $\mathcal{M}_\eta$. For any point $x \in \mathcal{M}_1$, since $\eta$ satisfies the PŁ inequality on $\mathbb{U}$ (and hence around $x$), from (Rebjock & Boumal, 2024), we know that $\mathcal{M}_1$ is locally a $\mathcal{C}^{\beta-1}$ embedded submanifold without boundary.

**Connectedness of $\mathcal{M}_\eta$.** Our strategy is to show that there exists a deformation retract $F : \mathbb{U} \times [0, 1] \to \mathbb{U}$ of $\mathbb{U}$ onto the topological subspace $\mathcal{M}_\eta \subset \mathbb{U}$. If this is true, using the standard result in topology, e.g. (Hatcher, 2002), $\mathcal{M}_\eta$ shares the same connectivity with $\mathbb{U}$. Since we have shown that $\mathbb{U}$ is connected, so is $\mathcal{M}_\eta$.

The negative gradient flow (up to a change of time) induced by the potential $\eta$ gives a natural construction of the deformation retract, please see (Criscitiello et al., 2025). The only thing we need to change in their proof is that their domain is $\mathbb{R}^D$. But since under our boundary condition, the gradient flow never leaves $\mathbb{U}$, the proof remains the same.

We have proved the statement. ∎

### G.4.4 PROOF OF THEOREM G.4

**Proof.** We denote in the following $\rho = \sqrt{2\eta}$. Since $\eta \geq 0$, $\rho \in \mathcal{C}^{\beta-1}(\mathbb{U} \setminus \mathcal{M}_\eta)$. One can calculate that

$$\|\nabla \rho\| = \|\frac{2\nabla \eta}{2\sqrt{2\eta}}\| = 1. \tag{G.25}$$

**Step 1:** $\rho(x) \le d_{\mathbb{U}}(x, \mathcal{M}_\eta)$**.** Fix $x \in \mathbb{U}$. Let $\alpha : [0,1] \to \mathbb{U}$ be absolutely continuous with $\alpha(0) = x$ and $\alpha(1) \in \mathcal{M}_\eta$. For $s \in (0,1)$ the map $\rho \circ \alpha$ is absolutely continuous on $[0, s]$ and for a.e. $t \in [0, s]$ we have (using Cauchy–Schwarz and $|\nabla \rho| = 1$ on $\mathbb{U} \setminus \mathcal{M}_\eta$)

$$\frac{d}{dt} \rho(\alpha(t)) = \nabla \rho(\alpha(t)) \cdot \alpha'(t) \ge -|\nabla \rho(\alpha(t))| \, |\alpha'(t)| = -|\alpha'(t)|.$$

Integrating from 0 to $s$ yields

$$\rho(\alpha(s)) - \rho(x) \ge \int_0^s -|\alpha'(t)| \, dt.$$

Letting $s \uparrow 1$ and using continuity of $\rho$ on $U$ plus $\rho(\alpha(1)) = 0$ gives

$$\rho(x) \le \int_0^1 |\alpha'(t)| \, dt = \mathrm{Length}(\alpha).$$

Thus $\rho(x) \le \mathrm{Length}(\alpha)$ for every admissible $\alpha$, hence $\rho(x) \le d_{\mathbb{U}}(x, \mathcal{M}_\eta)$ after taking the infimum in (G.2).

**Step 2:** $d_{\mathbb{U}}(x, \mathcal{M}_\eta) \le \rho(x)$ **for** $x \in \mathbb{U} \setminus \mathcal{M}_\eta$ **(characteristics).** For each $x \in \mathbb{U} \setminus \mathcal{M}_\eta$, let $\gamma_x : [0, T_x) \to \mathbb{U} \setminus \mathcal{M}_\eta$ be the maximal (classical) solution of the characteristic ODE

$$\gamma_x'(t) = -\nabla \rho(\gamma_x(t)), \qquad \gamma_x(0) = x, \tag{G.26}$$

where $T_x \in (0, \infty]$ is the maximal existence time in $\mathbb{U} \setminus \mathcal{M}_\eta$.

Following a similar proof as in Theorem G.2, the solution $\gamma_x$ exists on $[0, \rho(x)]$ and remains in $\mathbb{U}$. For $t \in [0, \rho(x))$, differentiating $\rho(\gamma_x(t))$ and using (G.26) gives

$$\frac{d}{dt} \rho(\gamma_x(t)) = \nabla \rho(\gamma_x(t)) \cdot \gamma_x'(t) = \nabla \rho(\gamma_x(t)) \cdot \left( -\nabla \rho(\gamma_x(t)) \right) = -|\nabla \rho(\gamma_x(t))|^2 = -1.$$

Therefore $\rho(\gamma_x(t)) = \rho(x) - t$ for $t \in [0, \rho(x))$, and by continuity we obtain

$$\rho(\gamma_x(\rho(x))) = \lim_{t \uparrow \rho(x)} \rho(\gamma_x(t)) = 0.$$

Since $\rho = 0$ precisely on $\mathcal{M}_\eta$, it follows that $\gamma_x(\rho(x)) \in \mathcal{M}_\eta$.

Next, since $|\nabla \rho| = 1$ on $\mathbb{U} \setminus \mathcal{M}_\eta$,

$$\mathrm{Length}\left(\gamma_x|_{[0,\rho(x)]}\right) = \int_0^{\rho(x)} |\gamma_x'(t)| \, dt = \int_0^{\rho(x)} |\nabla \rho(\gamma_x(t))| \, dt = \int_0^{\rho(x)} 1 \, dt = \rho(x).$$

Thus $\gamma_x|_{[0,\rho(x)]}$ is an admissible curve from $x$ to $\mathcal{M}_\eta$ of length $\rho(x)$, so $d_{\mathbb{U}}(x, \mathcal{M}_\eta) \le \rho(x)$.

**Step 3: conclude equality.** Combining Steps 1 and 2 yields $\rho(x) \le d_{\mathbb{U}}(x, \mathcal{M}_\eta) \le \rho(x)$ for all $x \in \mathbb{U} \setminus \mathcal{M}_\eta$. For $x \in \mathcal{M}_\eta$, both sides are 0 by definition. Hence $\rho(x) = d_{\mathbb{U}}(x, \mathcal{M}_\eta)$ for all $x \in \mathbb{U}$ and it is unique. ∎

### G.4.5 PROOF OF THEOREM G.10

**Proof.** From Theorem G.5, we know that on $\mathbb{U}_2$, one has $\eta(\cdot) = \frac{1}{2} \mathrm{dist}^2(\cdot, \mathcal{M}_\eta)$, and hence $\nabla \eta(x) = x - \pi_\eta(x)$, where $\pi_\eta$ denotes the projection operation onto the hypothesis manifold $\mathcal{M}_\eta$. Define the function $\Psi : \mathbb{R}^k \to \mathcal{M}_\eta$ as

$$\Psi(u) = x_{\mathrm{ref}} + W_\eta u + W_\eta^\perp \hat{N}_\eta(u) \text{ with some } u \in \mathbb{R}^k.$$

where we recall the definition of $W_\eta$, $W_\eta^\perp$, and $\hat{N}_\eta$ in Equation (G.6). There exists an open neighborhood $U \subseteq \mathbb{R}^k$ around 0, on which one has the identity

$$\forall u \in U, \quad \pi_\eta(\Psi(u)) = \Psi(u),$$

since $\Psi(u) \in \mathcal{M}$, and $\pi_\eta$ is an identity operation on $\mathcal{M}_\eta$. Following the discussion in Section F, we have that $\hat{N}_\eta \in \mathcal{C}^{\beta-1}$ since $\mathcal{M}_\eta$ is $\mathcal{C}^{\beta-1}$. Moreover, we will exploit the following fact:

$$\forall x \in \mathcal{M}_\eta, D[\pi_\eta](x) = P_{T_x \mathcal{M}_\eta},$$

where for some subspace of $\mathbb{R}^D$, $V$, $P_V$ denotes the orthogonal projection matrix onto $V$.

Apply $(W_\eta^\perp)^\top$ on both sides, one has

$$(W_\eta^\perp)^\top \pi_\eta(\Psi(u)) = \hat{N}_\eta(u). \tag{G.27}$$

Take derivative w.r.t. $u$, one has

$$\left(W_\eta^\perp\right)^\top D[\pi_\eta](\Psi(u))D[\Psi](u) = \left(W_\eta^\perp\right)^\top D[\pi_\eta](\Psi(u))\left(W_\eta + W_\eta^\perp D[\hat{N}_\eta](u)\right) = D[\hat{N}_\eta](u).$$

Rearranging terms, we have

$$D[\hat{N}_\eta](u) = \left(\underbrace{\mathbf{I}_{D-k} - \left(W_\eta^\perp\right)^\top D[\pi_\eta](\Psi(u))W_\eta^\perp}_{=:\mathbb{A}}\right)^{-1} \underbrace{\left(W_\eta^\perp\right)^\top D[\pi_\eta](\Psi(u))W_\eta}_{=:\mathbb{B}}.$$

**Invertibility of $\mathbb{A}$.** Use $W_u$ to denote an orthogonal basis of $\mathrm{span}(T_{\Psi(u)}\mathcal{M}_\eta)$. And for compactness, denote $P_u = P_{T_{\Psi(u)}\mathcal{M}_\eta} = W_u W_u^\top$ and $P_\eta^\perp = P_{(T_{x_{\mathrm{ref}}}\mathcal{M}_\eta)^\perp} = W_\eta^\perp (W_\eta^\perp)^\top$. We have

$$\left(W_\eta^\perp\right)^\top D[\pi_\eta](\Psi(u))W_\eta^\perp = \left(W_\eta^\perp\right)^\top P_u W_\eta^\perp = \left(W_\eta^\perp\right)^\top W_u W_u^\top W_\eta^\perp. \tag{G.28}$$

Let $\sigma_{\max}(\cdot)$ denote the largest singular value of a matrix. One has

$$\sigma_{\max}(\left(W_\eta^\perp\right)^\top W_u W_u^\top W_\eta^\perp) = \sigma_{\max}(W_u^\top W_\eta^\perp \left(W_\eta^\perp\right)^\top W_u).$$

Note that

$$W_u^\top W_\eta^\perp \left(W_\eta^\perp\right)^\top W_u + W_u^\top W_\eta W_\eta^\top W_u = \mathbf{I}_k.$$

Note that if $A + B = I$, then $\sigma_{\min}(A) = 1 - \sigma_{\max}(B)$, and hence

$$\sigma_{\max}(W_u^\top W_\eta^\perp \left(W_\eta^\perp\right)^\top W_u) = 1 - \sigma_{\min}^2(W_\eta^\top W_u).$$

Use (Ji-Guang, 1987, Theorem 2.1) again to obtain

$$\sigma_{\max}(\left(W_\eta^\perp\right)^\top W_u W_u^\top W_\eta^\perp) = \sin^2\theta_{\max}(\mathrm{span}(W_u), \mathrm{span}(W_\eta)) = \|P_u - P_\eta\|_{op}^2.$$

Note that

$$P_u = D[\pi_\eta](\Psi(u)) = \mathbf{I}_D - D^2[\eta](\Psi(u)) \text{ and } P_\eta = D[\pi_\eta](x_{\mathrm{ref}}) = \mathbf{I}_D - D^2[\eta](x_{\mathrm{ref}}).$$

Hence by the smoothness constraint (last line in Equation (D.4)), we have

$$\|P_u - P_\eta\|_{op}^2 = \|D^2[\eta](\Psi(u)) - D^2[\eta](x_{\mathrm{ref}})\|_{op}^2 \le L_3\|\Psi(u) - x_{\mathrm{ref}}\|^2 = \mathcal{O}(h^2), \tag{G.29}$$

since $\Psi(u) \in B_D^{\mathrm{Euc}}(x_{\mathrm{ref}}, h)$. Consequently, $\mathbb{A} \succeq (1 - \mathcal{O}(h^2))\mathbf{I}_{D-k} \succ 0$ for $h$ sufficiently small.

**Boundedness of $\mathbb{B}$.** We can simply bound

$$\| \left(W_\eta^\perp\right)^\top D[\pi_\eta](\Psi(u))W_\eta\|_{op} = \| \left(W_\eta^\perp\right)^\top W_u W_u^\top W_\eta\|_{op} \le \|W_u^\top W_\eta\|_{op}.$$

Following the derivation in Equation (G.29) Note that

$$\forall u \in B_k(0, h), \ \|W_u^\top W_\eta\|_{op} = \|P_u - P_\eta\|_{op} = L_3 h.$$

Combining the above derivation, we conclude that with $\hat{L}_1 = 2L_3$

$$\|D[\hat{N}_\eta](u) \le \hat{L}_1 h,$$

for all $h$ below a constant threshold.

For higher derivatives, apply the multivariate Faà di Bruno formula to the composition $(W_\eta^\perp)^\top \pi_\eta \circ \Psi$. At order $j \ge 2$, every term in $D^j\left[(W_\eta^\perp)^\top \pi_\eta \circ \Psi\right](u)$ is a finite sum of tensors built from:

- $D^{\ell}[\pi_{\eta}](\Psi(u))$ for $1 \le \ell \le j$, and
- derivatives $D^q[\Psi](u)$ for $1 \le q \le j$.

Crucially, the only term in the expansion of $D^j\big[(W_{\eta}^{\perp})^{\top}\pi_{\eta} \circ \Psi\big](u)$ that contains $D^j[\hat{N}_{\eta}](u)$ is

$$(W_{\eta}^{\perp})^{\top} D[\pi_{\eta}](\Psi(u)) W_{\eta}^{\perp} D^j[\hat{N}_{\eta}](u), \tag{G.30}$$

Hence the order-$j$ identity obtained by differentiating (G.27) at 0 has the form

$$D^j[\hat{N}_{\eta}](u) = \mathbb{A}^{-1}\mathbf{F}_j\Big(\{D^{\ell}[\pi_{\eta}](\Phi(u))\}_{\ell=1}^{j}, \ \{D^q[\hat{N}_{\eta}](u)\}_{q=1}^{j-1}\Big),$$

where $\mathbf{F}_j$ is a universal multilinear combination (coming from the Faà di Bruno formula) that does *not* involve $D^j[\hat{N}_{\eta}](u)$ on the right-hand side. This yields an induction: assuming bounds for $D^q[\hat{N}_{\eta}](u)$ for $1 \le q \le j-1$, one bounds $D^j[\hat{N}_{\eta}](u)$ by a polynomial in $\|D[\pi_{\eta}](\Phi(u))\|, \dots, \|D^j[\pi_{\eta}](\Phi(u))\|$ with a constant $\hat{L}_j$ depending only on $(k, D, j, \mathbf{L})$. This proves (G.18).  ∎

## G.5  Feasibility of the Ground Truth Distance Function, i.e. $\eta^{\star} \in \mathcal{D}_{\mathbf{L}}^k$

Suppose that the parameters in $\mathbf{L}$ are taken to be sufficiently large. All requirements in Equation (D.4) are straight-forward to verify except for the non-escape boundary condition (second line).

To verify the non-escape condition, for any $x \in \partial\mathbb{U}$, if we can show that for any $x \in \partial\mathbb{U}$, $\mathrm{dist}(x, \mathcal{M}^{\star}) \ge \zeta_{\min}/2 - \epsilon$, conditioned on the high probability event that $Y_n \subseteq \mathcal{M}^{\star}$ is an $\epsilon$-net of $\mathcal{M}^{\star}$. We can then use Theorem I.4 to show that all points $y$ on $\mathcal{M}^{\star}$ such that $\|y - x\| = \zeta_{\min}/2$ are close to $proj(x)$ in the manifold geodesic distance. Since we have that the Euclidean distance between $y$ and $\mathrm{Proj}_{\mathcal{M}}(x)$ is bounded by the corresponding geodesic distance, we can use the law of cosines to ensure the existence of $\delta$ in the non-escape boundary (note that $\nabla\eta^{\star}(x) = x - \mathrm{Proj}_{\mathcal{M}}(x)$).

To establish a lower bound for $\mathrm{dist}(x, \mathcal{M}^{\star})$, notice that since $Y_n$ is an $\epsilon$-net of $\mathcal{M}^{\star}$, there exists $y \in Y_n$ such that $\|y - \mathrm{Proj}_{\mathcal{M}}(x)\| \le \epsilon$. Moreover, since $x \in \partial\mathbb{U}$, $\|y - x\| \ge \zeta_{\min}/2$ (otherwise, $x \in \mathbb{U}$ which is not in $\partial U$). We hence have

$$\|x - \mathrm{Proj}_{\mathcal{M}}(x)\| \ge \|x - y\| - \|y - \mathrm{Proj}_{\mathcal{M}}(x)\| \ge \zeta_{\min}/2 - \epsilon. \tag{G.31}$$

Now use Theorem I.4 with $\mathcal{M} = \mathcal{M}^{\star}$, $d = \mathrm{dist}(x, \mathcal{M}^{\star})$ and note that $\epsilon' \le \epsilon$, we have that $d_{\mathcal{M}^{\star}}^2(y, \mathrm{Proj}_{\mathcal{M}}(x)) = \mathcal{O}(\epsilon)$. Following the above discussion, we ensure that $\nabla\eta^{\star}(x) = x - \mathrm{Proj}_{\mathcal{M}}(x)$ fulfills the non-escape boundary condition (second line of Equation (D.4)).

## H  Proof of Theorem 2.2

We take the following steps to prove Theorem E.1. We then prove Theorem 2.2 based on Theorem E.1 in Section H.5.

- We first show Assumption 2 can be translated to guarantee that a local principal manifold estimation (PME) problem is solved with high accuracy.
- We then show that the small PME loss implies a polynomial estimation problem is solved up to the accuracy of $t$.
- By picking $t = \mathcal{O}(h^{2(\beta-1)})$, we can use (Aamari & Levrard, 2019, Proposition 2) to show that we have estimated the derivatives of the graph-of-function representation for the ground truth manifold $\mathcal{M}^{\star}$ to a high accuracy. We can hence follow the same argument as (Aamari & Levrard, 2019, Theorem 6) to conclude the closeness between $\mathcal{M}^{\star}$ and $\mathcal{M}_{\eta}$ in the Hausdorff sense.

### H.1  From Denoising Score Matching to Principal Manifold Estimation

Recall the definition of $U_2$ in Equation (G.3) and recall Theorem G.5 which proves that for any fixed $\eta \in \mathcal{D}_{\mathbf{L}}^k$,

$$\forall x \in \mathbb{U}_2, \quad \eta(x) = \frac{1}{2}\mathrm{dist}^2(x, \mathcal{M}_{\eta}), \tag{H.1}$$

where $\mathcal{M}_\eta = \{x \in \mathbb{U} \mid \eta(x) = 0\}$ is the corresponding zero set. Consequently, we have for every $s_\eta \in \mathcal{S}$ (we use the subscript to highlight the correspondence between $\eta$ and $s$)

$$\forall x \in \mathbb{U}_2, \quad s_\eta(t, x) = -\frac{x - \pi_\eta(x)}{t}, \tag{H.2}$$

where $\pi_\eta$ denotes the projection onto $\mathcal{M}_\eta$.

Define the truncated Gaussian measure as

$$z \sim \mathcal{N}_{tr}^s(0, tI_d) = \frac{1}{Z_t \cdot (1 - \Pr(\|z\| \geq s))} \exp(-\frac{\|z\|^2}{2t}) \mathbb{1}(\|z\| \leq s). \tag{H.3}$$

where $Z_t$ is the normalizing factor for the standard $d$ dimensional Gaussian with variance $t$, and $s$ is some threshold.

For a given reference point $x_{\mathrm{ref}} \in Y_n$, we define a corresponding Principal Manifold Estimation (PME) loss as follows

$$\mathrm{PME}_t(\eta) := \mathbb{E}_{x_0 \sim \mu_{\mathrm{emp}}^{x_{\mathrm{ref}}, h}, x = x_0 + z, z \sim \mathcal{N}_{tr}^h(0, tI_d)} \left[\mathrm{dist}^2(x, \mathcal{M}_\eta)\right]. \tag{H.4}$$

We justify the naming of PME by noting that the above loss defines the average deviation of the samples $x$ from the corresponding zero set $\mathcal{M}_\eta$. This is a non-linear extension to the classical principal component analysis.

**Lemma H.1** *Let $s_{\hat\eta} \in \mathcal{S}$ be a function that satisfies Assumption 2. Let $\hat\eta$ be the corresponding function in $\mathcal{D}_L^k$. We have*

$$PME_t(\hat\eta) = \mathcal{O}(t). \tag{H.5}$$

**Proof.** Recall the definition of $\mathrm{DSM}_t(s; x_0)$ in Equation (4). Expand the above quadratic, we have

$$\mathrm{DSM}_t(s; x_0) = \mathbb{E}_{x \sim q_t(x|x_0)} \left[\|s(t, x)\|^2 + \|\nabla_x \log q_t(x \mid x_0)\|^2 - 2s(t, x) \cdot \nabla_x \log q_t(x \mid x_0)\right]$$

Using Young's inequality for the last term, one has

$$|2s(t, x) \cdot \nabla_x \log q_t(x \mid x_0)| \leq \frac{1}{2} \|s(t, x)\|^2 + 2\|\nabla_x \log q_t(x \mid x_0)\|^2.$$

Moreover, one can explicitly calculate that

$$\mathbb{E}_{x \sim q_t(x|x_0)} \left[\|\nabla_x \log q_t(x \mid x_0)\|^2\right] = \mathbb{E}_{(x - x_0) \sim \mathcal{N}(0, tI_d)} \left[\frac{\|x - x_0\|^2}{t^2}\right] = \frac{1}{t}. \tag{H.6}$$

We hence have, for any $s \in \mathcal{S}$

$$\frac{1}{2} \mathbb{E}_{x \sim q_t(x|x_0)} \left[\|s(t, x)\|^2\right] - \frac{1}{t} \leq \mathrm{DSM}_t(s; x_0) \leq \frac{3}{2} \mathbb{E}_{x \sim q_t(x|x_0)} \left[\|s(t, x)\|^2\right] + \frac{3}{t} \tag{H.7}$$

Let $\pi^\star$ denote the projection onto the ground truth manifold $\mathcal{M}^\star$. We use $s^\star$ to denote the score function corresponding to the ground truth manifold, i.e.

$$s^\star = -\frac{x - \pi^\star(x)}{t} \text{ for } x \in \mathbb{U} \quad \text{and} \quad s^\star = 0 \text{ otherwise.} \tag{H.8}$$

We have (use $\mathcal{N}(x; x_0, tI_d)$ to denote the density of $\mathcal{N}(x_0, tI_d)$ at $x$)

$$\mathbb{E}_{x \sim q_t(x|x_0)} \left[\|s^\star(x)\|^2\right] = \int_{x \in \mathbb{U}} \frac{1}{t^2} \mathrm{dist}(x, \mathcal{M}^\star)^2 \mathcal{N}(x; x_0, tI_d) \mathrm{d}x$$

$$(\text{since } x_0 \in \mathcal{M}^\star) \quad \leq \int_{x \in \mathbb{U}} \frac{1}{t^2} \|x - x_0\|^2 \mathcal{N}(x; x_0, tI_d) \mathrm{d}x \leq \int \frac{1}{t^2} \|x - x_0\|^2 \mathcal{N}(x; x_0, tI_d) \mathrm{d}x$$

$$= \mathcal{O}(\frac{1}{t}). \tag{H.9}$$

Using the first inequality in Equation (H.7), we have

$$\frac{1}{2} \mathbb{E}_{x_0 \sim \mu_{\mathrm{emp}}^{x_{\mathrm{ref}}, h}, x \sim q_t(x|x_0)} \left[\|s_{\hat\eta}(x)\|^2\right] - \frac{1}{t} \leq \mathbb{E}_{x_0 \sim \mu_{\mathrm{emp}}^{x_{\mathrm{ref}}, h}} [\mathrm{DSM}_t(s_{\hat\eta}; x_0)] \tag{H.10}$$

Using assumption 2 (w.l.o.g., assume the constant $C$ therein is $C = 1$) and

$$\min_{\eta \in \mathcal{D}_{\mathbf{L}}^k} \mathbb{E}_{x_0 \sim \mu_{\text{emp}}^{x_{\text{ref}}, h}}[\text{DSM}_t(s_\eta; x_0)] \leq \mathbb{E}_{x_0 \sim \mu_{\text{emp}}^{x_{\text{ref}}, h}}[\text{DSM}_t(s^\star; x_0)]$$

(since $s^\star$ is feasible), we have

$$\mathbb{E}_{x_0 \sim \mu_{\text{emp}}^{x_{\text{ref}}, h}}[\text{DSM}_t(s_{\hat{\eta}}; x_0)] \leq \mathbb{E}_{x_0 \sim \mu_{\text{emp}}^{x_{\text{ref}}, h}}[\text{DSM}_t(s^\star; x_0)] + \frac{1}{t} \tag{H.11}$$

Using the second inequality in Equation (H.7) and Equation (H.9), we have

$$\mathbb{E}_{x_0 \sim \mu_{\text{emp}}^{x_{\text{ref}}, h}}[\text{DSM}_t(s^\star; x_0)] \leq \frac{3}{2}\mathbb{E}_{x_0 \sim \mu_{\text{emp}}^{x_{\text{ref}}, h}, x \sim q_t(x|x_0)}\left[\|s^\star(x)\|^2\right] + \frac{3}{t} = \mathcal{O}(\frac{1}{t}) \tag{H.12}$$

Combining the above results, we have

$$\mathbb{E}_{x_0 \sim \mu_{\text{emp}}^{x_{\text{ref}}, h}, x \sim q_t(x|x_0)}\left[\|s_{\hat{\eta}}(x)\|^2\right] = \mathcal{O}(\frac{1}{t}) \tag{H.13}$$

By Theorem G.4, one has

$$\mathbb{E}_{x_0 \sim \mu_{\text{emp}}^{x_{\text{ref}}, h}, x \sim q_t(x|x_0) \cdot \mathbb{1}(\mathbb{U})}[d_{\mathbb{U}}^2(x, \mathcal{M}_{\hat{\eta}})] = \mathcal{O}(t). \tag{H.14}$$

Since $d_{\mathbb{U}}^2(x, \mathcal{M}_{\hat{\eta}}) \geq 0$, we clearly have

$$\mathbb{E}_{x_0 \sim \mu_{\text{emp}}^{x_{\text{ref}}, h}, x \sim q_t(x|x_0) \cdot \mathbb{1}(\mathbb{U}_2)}[\text{dist}^2(x, \mathcal{M}_{\hat{\eta}})] = \mathcal{O}(t), \tag{H.15}$$

where we used that $\text{dist}_{\mathbb{U}}^2(x, \mathcal{M}_{\hat{\eta}})$ and $\text{dist}^2(x, \mathcal{M}_{\hat{\eta}})$ agree on $\mathbb{U}_2$.

Next, we show that Equation (H.15) implies $\text{PME}_t(\hat{\eta}) = \mathcal{O}(t)$. Note that for any $\eta \in \mathcal{D}_{\mathbf{L}}^k$

$$\text{PME}_t(\eta) = \mathbb{E}_{x_0 \sim \mu_{\text{emp}}^{x_{\text{ref}}, h}} \int_{\|z\| \leq h} \text{dist}^2(x_0 + z, \mathcal{M}_\eta) \frac{1}{Z_t \cdot (1 - \text{Pr}(\|z\| \geq h))} \exp(-\frac{\|z\|^2}{2t}) dz$$

$$\leq \mathbb{E}_{x_0 \sim \mu_{\text{emp}}^{x_{\text{ref}}, h}} \int_{\|z\| \leq \tau_{\min}/4} \text{dist}^2(x_0 + z, \mathcal{M}_\eta) \frac{1}{Z_t \cdot (1 - \text{Pr}(\|z\| \geq h))} \exp(-\frac{\|z\|^2}{2t}) dz$$

$$= \frac{1}{1 - \text{Pr}(\|z\| \geq h)}\mathbb{E}_{x_0 \sim \mu_{\text{emp}}^{x_{\text{ref}}, h}, x \sim q_t(x|x_0) \cdot \mathbb{1}(\mathbb{U}_2)}[\text{dist}^2(x, \mathcal{M}_\eta)]$$

Hence it suffices to show that for $z \sim \mathcal{N}(0, tI_d)$

$$\text{Pr}(\|z\| \geq h) \leq \frac{1}{2}. \tag{H.16}$$

**The probability that $\|z\| \geq h$**    We note that the probability of the event that $\|z\| \geq h$ is bounded by (up to a constant)

$$I(t) := \frac{1}{t^{D/2}} \int_{|z| \geq h} e^{-|z|^2/t} dz,$$

up to some constant. Moreover, by the standard gamma function asymptotics, we have

$$I(t) = \mathcal{O}(t^{-D/2} \exp(-h/t)). \tag{H.17}$$

Since we choose $t = \mathcal{O}(h^{2(\beta-1)})$, $\text{Pr}(\|z\| \geq h) \leq \frac{1}{2}$ for a sufficiently small $h$.

∎

## H.2   FROM PRINCIPAL MANIFOLD ESTIMATION TO POLYNOMIAL ESTIMATION

Recall the parameterization of the hypothesis submanifold $\mathcal{M}_\eta$ under the ground-truth basis $(W_{\text{ref}}, W_{\text{ref}}^\perp)$ in Section G.2.1. To approximately recover the ground truth manifold $\mathcal{M}^\star$, it is sufficient if one shows the hypothesis coordinate function $N_\eta$ (defined in Equation (G.12)) is close to the ground truth one $N_{\text{ref}}$. In this section, we show that when the PME loss is small, so is the $\mathcal{L}^2$ distance between $N_\eta$ and $N_{\text{ref}}$, under the measure $[\mu_{\text{emp}}^{x_{\text{ref}}, h}]$. This result together with (Aamari & Levrard, 2019, Proposition 2) shows that the coefficients of the Taylor's expansion of $N_\eta$ and $N_{\text{ref}}$ around 0 (this corresponds to $x_{\text{ref}}$) are close up to the $\beta^{th}$ order. Following the same argument as in (Aamari & Levrard, 2019), one can bound the Hausdorrf distance between $\mathcal{M}_\eta$ and $\mathcal{M}^\star$.

**Lemma H.2** *Let $s_{\hat{\eta}} \in \mathcal{S}$ be the hypothesis score function that satisfies Assumption 2 and let $\hat{\eta}$ be the corresponding function in $\mathcal{D}_L^k$. We have (denote $v(x_0) = W_{\mathrm{ref}}^\top(x_0 - x_{\mathrm{ref}})$)*

$$\mathbb{E}_{x_0 \sim \mu_{\mathrm{emp}}^{x_{\mathrm{ref}}, h}} \| N_\eta(v(x_0)) - N_{\mathrm{ref}}(v(x_0)) \|^2 = \mathcal{O}(t) \tag{H.18}$$

*Further, let $T_\beta(v)$ denote the $(\beta - 2)^{th}$ order Taylor expansion of $(N_{\mathrm{ref}} - N_\eta)$ around $0$. By taking $t = \mathcal{O}(h^{2(\beta-1)})$, we have*

$$\mathbb{E}_{x_0 \sim \mu_{\mathrm{emp}}^{x_{\mathrm{ref}}, h}} [\| T_\beta(v(x_0)) \|^2] = \mathcal{O}(h^{2(\beta-1)}). \tag{H.19}$$

**Proof.** First, let us decompose the PME loss into $\mathrm{span}(W_{\mathrm{ref}})$ and $\mathrm{span}(W_{\mathrm{ref}}^\perp)$. To do so, notice that we can rewrite the $x \sim q_t(x \mid x_0)$ equivalently as

$$x \stackrel{\mathrm{d}}{=} x_0 + z, \ x_0 \in \mathcal{M}^\star, \ z \sim \mathcal{N}(0, t\mathbf{I}_d). \tag{H.20}$$

Recall the local representation of a submanifold in Section F, we can write (note that for every $x_0$ there is a corresponding $v = W_{\mathrm{ref}}^\top(x_0 - x_{\mathrm{ref}})$)

$$x_0 = x_{\mathrm{ref}} + W_{\mathrm{ref}}v + W_{\mathrm{ref}}^\perp N_{\mathrm{ref}}(v). \tag{H.21}$$

Decompose the random noise $z$ accordinig to the basis $(W_{\mathrm{ref}}, W_{\mathrm{ref}}^\perp)$

$$z = W_{\mathrm{ref}}z_T + W_{\mathrm{ref}}^\perp z_N, \ \text{where } z_T \sim \mathcal{N}(0, t\mathbf{I}_k) \text{ and } z_N \sim \mathcal{N}(0, t\mathbf{I}_{d-k}). \tag{H.22}$$

In the following discussion, we condition on the event that $\|z\| \leq h$ (since in the $\mathrm{PME}_t$ loss, the expectation is conditioned on this event). Recall the representation of $\mathcal{M}_\eta$ under the basis $(W_{\mathrm{ref}}, W_{\mathrm{ref}}^\perp)$

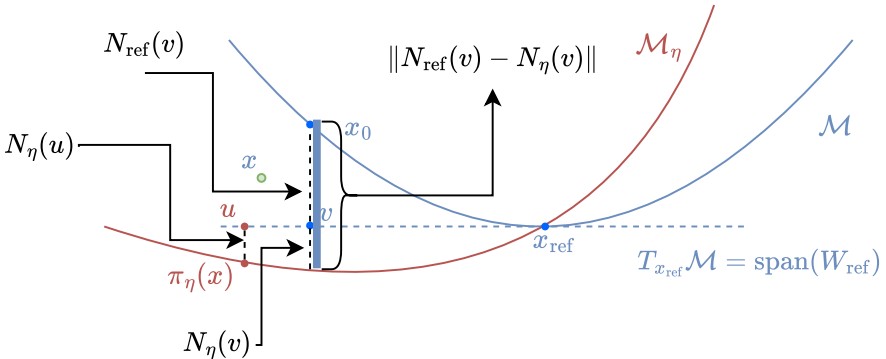

in Theorem G.9: Define the function $\hat{x}_\eta : \mathbb{R}^k \to \mathbb{R}^d$

$$\hat{x}_\eta(u) = x_{\mathrm{ref}} + W_{\mathrm{ref}}u + W_{\mathrm{ref}}^\perp N_\eta(u). \tag{H.23}$$

By definition, we have $\hat{x}_\eta(u) \in \mathcal{M}_\eta$. The projection operator onto $\mathcal{M}_\eta$ can be recovered in the following way: Define

$$u_\eta(x) = \underset{u : \|v - u\| \leq C_h h}{\arg\min} \|x - \hat{x}_\eta(u)\|^2, \tag{H.24}$$

For some sufficiently large constant $C_h \geq 4$ (to be decided later). Note that the constraint $\|v - u\| \leq C_h h$ is inactive, as discussed in Section H.3.1. One has

$$\pi_\eta(x) = W_{\mathrm{ref}}u_\eta(x) + W_{\mathrm{ref}}^\perp N_\eta(u_\eta(x)). \tag{H.25}$$

We can hence decompose $\mathrm{dist}^2(x, \mathcal{M}_\eta)$ under the basis $(W_{\mathrm{ref}}, W_{\mathrm{ref}}^\perp)$

$$\mathrm{dist}^2(x, \mathcal{M}_\eta) = \underset{u : \|v - u\| \leq C_h h}{\min} \left\{ \|x - \hat{x}_\eta(u)\|^2 = \|v + z_T - u\|^2 + \|N_{\mathrm{ref}}(v) + z_N - N_\eta(u)\|^2 \right\}. \tag{H.26}$$

Define $\Delta_v = u - v$ (we focus on $\Delta_v = \mathcal{O}(h)$ due to the constraint in Equation (H.24)) and define $L : \mathbb{R}^k \to \mathbb{R}$

$$\|x - \hat{x}_\eta(u)\|^2 = L(\Delta_v) := \|\Delta_v - z_T\|^2 + \|N_{\mathrm{ref}}(v) + z_N - N_\eta(v + \Delta_v)\|^2 \tag{H.27}$$

Recall the first-order Taylor expansion of $N_\eta$ in Theorem G.9,

$$N_\eta(v + \Delta_v) = N_\eta(v) + D[N_\eta](v)\Delta_v + \mathcal{O}(\|\Delta_v\|^2), \tag{H.28}$$

with $D[N_\eta]$ specified in Equation (G.13). For compactness, we use $\mathbf{J}_v$ to denote $D[N_\eta](v)$.

To exploit this expansion, define

$$L_0(\Delta_v) = \|\Delta_v - z_T\|^2 + \|N_{\mathrm{ref}}(v) + z_N - N_\eta(v) - \mathbf{J}_v\Delta_v\|^2, \tag{H.29}$$

that is, we ignore the higher order term in the second term of Equation (H.27). We note that $L_0(\Delta_v)$ is quadratic in $\Delta_v$.

The difference between $L(\Delta_v)$ and $L_0(\Delta_v)$ writes

$$L(\Delta_v) - L_0(\Delta_v) = 2\langle N_{\mathrm{ref}}(v) + z_N - N_\eta(v) - \mathbf{J}_v^\top\Delta_v, \mathcal{O}(\|\Delta_v\|^2)\rangle + \mathcal{O}(\|\Delta_v\|^4). \tag{H.30}$$

One can hence bound

$$
\begin{aligned}
|L(\Delta_v) - L_0(\Delta_v)| &\leq 2\langle N_{\mathrm{ref}}(v) - N_\eta(v) - \mathbf{J}_v^\top\Delta_v, \mathcal{O}(\|\Delta_v\|^2)\rangle + \mathcal{O}(\|\Delta_v\|^4) + \|z_N\|^2 \\
&= 2\langle (N_{\mathrm{ref}}(v) - N_\eta(v))\|\Delta_v\|^{0.5}, \mathcal{O}(\|\Delta_v\|^{1.5})\rangle + \langle \mathbf{J}_v^\top\Delta_v, \mathcal{O}(\|\Delta_v\|^2)\rangle + \mathcal{O}(\|\Delta_v\|^4) + \|z_N\|^2 \\
&\leq \|N_{\mathrm{ref}}(v) - N_\eta(v)\|^2\|\Delta_v\| + \mathcal{O}(\|\Delta_v\|^3) + \|z_N\|^2 \\
&\leq \mathcal{O}(h)\|N_{\mathrm{ref}}(v) - N_\eta(v)\|^2 + \mathcal{O}(h)\|\Delta_v\|^2 + \|z_N\|^2.
\end{aligned}
$$

where we use the fact that $\Delta_v = \mathcal{O}(h)$. Combining the above results, we have

$$L(\Delta_v) \geq L_0(\Delta_v) - \mathcal{O}(h)\|N_{\mathrm{ref}}(v) - N_\eta(v)\|^2 - \mathcal{O}(h)\|\Delta_v\|^2 - \|z_N\|^2. \tag{H.31}$$

Take minimum on both sides, we have

$$\min_{\Delta_v : \|\Delta_v\| \leq C_h h} L(\Delta_v) \geq \min_{\Delta_v : \|\Delta_v\| \leq C_h h} \left\{ L_0(\Delta_v) - \mathcal{O}(h)\|N_{\mathrm{ref}}(v) - N_\eta(v)\|^2 - \mathcal{O}(h)\|\Delta_v\|^2 - \|z_N\|^2 \right\}. \tag{H.32}$$

This is of interest because

$$\mathrm{dist}^2(x, \mathcal{M}_\eta) = \min_{u : \|v - u\| \leq C_h h} \|x - \hat{x}_\eta(u)\|^2 = \min_{\Delta_v : \|\Delta_v\| \leq C_h h} L(\Delta_v). \tag{H.33}$$

Notice that the R.H.S. of Equation (H.32) is a quadratic w.r.t. $\Delta_h$, i.e. we have $L_0(\Delta_v) - \mathcal{O}(h)\|\Delta_v\|^2 = \|\Delta_v\|_A^2 + 2\langle\Delta_v, b\rangle + c$ with

$$
\begin{aligned}
A &:= \mathbf{I}_k + \mathbf{J}_v^\top\mathbf{J}_v - \mathcal{O}(h), \\
b &:= z_T + \mathbf{J}_v(N_{\mathrm{ref}}(v) + z_N - N_\eta(v)), \\
c &:= \|N_{\mathrm{ref}}(v) + z_N - N_\eta(v)\|^2 + \|z_T\|^2.
\end{aligned}
$$

Denote $\lambda_{\max} = \sigma_{\max}(\mathbf{J}_v^\top\mathbf{J}_v)$. Note that

$$1 - \mathcal{O}(h) \preceq A \preceq 1 + \lambda_{\max} - \mathcal{O}(h).$$

The minimizer of the above quadratic is attained at $\Delta^* = A^{-1}b$. We can show that $\Delta^* = \mathcal{O}(h)$ so it remains feasible when $C_h$ defined in Equation (H.24) is sufficiently large. This is discussed in Section H.3.2.

Hence

$$
\begin{aligned}
\min_{\Delta_v : \|\Delta_v\| \leq C_h h} L_0(\Delta_v) - \mathcal{O}(h)\|\Delta_v\|^2 &= c - \|b\|_{A^{-1}}^2 \\
&= \|N_{\mathrm{ref}}(v) + z_N - N_\eta(v)\|^2 + \|z_T\|^2 - \|z_T + \mathbf{J}_v(N_{\mathrm{ref}}(v) + z_N - N_\eta(v))\|_{A^{-1}}^2 \\
&= \|N_{\mathrm{ref}}(v) - N_\eta(v)\|^2 - \|\mathbf{J}_v(N_{\mathrm{ref}}(v) - N_\eta(v))\|_{A^{-1}}^2 \\
&\quad + 2\langle z_N, N_{\mathrm{ref}}(v) - N_\eta(v)\rangle - 2\langle z_T + \mathbf{J}_v z_N, \mathbf{J}_v(N_{\mathrm{ref}}(v) - N_\eta(v))\rangle_{A^{-1}} \\
&\quad + \|z_N\|^2 + \|z_T + \mathbf{J}_v z_N\|_{A^{-1}}^2 + \|z_T\|^2
\end{aligned}
$$

Bound

$$2|\langle z_N, N_{\text{ref}}(v) - N_\eta(v)\rangle| \leq \underbrace{4\frac{1 - \mathcal{O}(h) + \lambda_{\max}}{1 - \mathcal{O}(h)}\|z_N\|_A^2}_{=\mathcal{O}(\|z\|^2)} + \frac{1}{4}\frac{1 - \mathcal{O}(h)}{1 - \mathcal{O}(h) + \lambda_{\max}}\|N_{\text{ref}}(v) - N_\eta(v)\|^2,$$

and

$$2|\langle z_T + \mathbf{J}_v z_N, \mathbf{J}_v(N_{\text{ref}}(v) - N_\eta(v))\rangle_{A^{-1}}|$$

$$\leq \underbrace{4\frac{1 - \mathcal{O}(h) + \lambda_{\max}}{1 - \mathcal{O}(h)}\|\mathbf{J}_v^\top A^{-1} z_T + \mathbf{J}_v^\top A^{-1}\mathbf{J}_v z_N\|^2}_{=\mathcal{O}(\|z\|^2)} + \frac{1 - \mathcal{O}(h)}{1 - \mathcal{O}(h) + \lambda_{\max}}\frac{1}{4}\|N_{\text{ref}}(v) - N_\eta(v)\|^2$$

We have that

$$\|\mathbf{J}_v(N_{\text{ref}}(v) - N_\eta(v))\|_{A^{-1}}^2 \leq (1 - \frac{1 - \mathcal{O}(h)}{1 - \mathcal{O}(h) + \lambda_{\max}})\|N_{\text{ref}}(v) - N_\eta(v)\|^2,$$

$$\min_{\Delta_v : \|\Delta_v\| \leq C_h h} L_0(\Delta_v) - \mathcal{O}(h)\|\Delta_v\|^2$$

$$\geq \|N_{\text{ref}}(v) - N_\eta(v)\|^2 - \|\mathbf{J}_v(N_{\text{ref}}(v) - N_\eta(v))\|_{A^{-1}}^2 - \frac{1}{2}\frac{1 - \mathcal{O}(h)}{1 - \mathcal{O}(h) + \lambda_{\max}}\|N_{\text{ref}}(v) - N_\eta(v)\|^2 - \mathcal{O}(\|z\|^2)$$

$$\geq \frac{1}{2}\left(\frac{1 - \mathcal{O}(h)}{1 - \mathcal{O}(h) + \lambda_{\max}}\right)\|N_{\text{ref}}(v) - N_\eta(v)\|^2 - \mathcal{O}(\|z\|^2)$$

We can hence bound (for $h$ sufficiently small)

$$\text{dist}^2(x, \mathcal{M}_\eta) + \mathcal{O}(\|z\|^2) \geq \frac{1}{2}\left(\frac{1 - \mathcal{O}(h)}{1 - \mathcal{O}(h) + \lambda_{\max}}\right)\|N_{\text{ref}}(v) - N_\eta(v)\|^2 \geq \frac{1}{4(1 + \lambda_{\max})}\|N_{\text{ref}}(v) - N_\eta(v)\|^2.$$

Take expectation w.r.t. $x = x_0 + z$ with $z \sim \mathcal{N}_{tr}^h(0, tI_d)$ and $x_0 \sim \mu_{\text{emp}}^{x_{\text{ref}}, h}$. The first term on the L.H.S. becomes $\text{PME}_t(\eta)$ and the second term can be bounded by

$$\mathbb{E}_{z \sim \mathcal{N}_{tr}^h(0, tI_d)}[\|z\|^2] \leq \mathbb{E}_{z \sim \mathcal{N}(0, tI_d)}[\|z\|^2] = \mathcal{O}(t). \tag{H.34}$$

and we hence have the first conclusion by taking $\eta = \hat{\eta}$.

**Polynomial Estimation.** Recall that by definition $N_{\text{ref}} \in \mathcal{C}^\beta$ and its derivatives are bounded in operator norm (see Theorem F.1). Moreover, recall that we prove in Theorem G.9 $N_\eta \in \mathcal{C}^{\beta-1}$ and show that its derivatives are bounded in operator norm in Theorem G.11. Consequently, we have that $(N_{\text{ref}} - N_\eta)$ is a $\mathcal{C}^{\beta-1}$ and with its derivatives (up to $(\beta - 1)^{th}$ order) bounded in operator norm. Let $T_\beta(v)$ denote the $(\beta - 2)^{th}$ order Taylor expansion of $(N_{\text{ref}} - N_\eta)$ around 0. We have

$$\mathbb{E}_{x_0 \sim \mu_{\text{emp}}^{x_{\text{ref}}, h}}[\|T_\beta(v) - (N_{\text{ref}}(v) - N_{\hat{\eta}}(v))\|^2] = \mathcal{O}(h^{2(\beta-1)}). \tag{H.35}$$

By taking $t = \mathcal{O}(h^{2(\beta-1)})$, we have (note that $W_{\text{ref}}^\top(x - x_{\text{ref}}) = v$)

$$\mathbb{E}_{x_0 \sim \mu_{\text{emp}}^{x_{\text{ref}}, h}}[\|T_\beta(v)\|^2] = \mathcal{O}(h^{2(\beta-1)}). \tag{H.36}$$

$\blacksquare$

## H.3    FROM POLYNOMIAL ESTIMATION TO HAUSDORFF DISTANCE BOUND

Theorem H.2 together with (Aamari & Levrard, 2019, Proposition 2) ensures that the coefficients of the polynomial $T_\beta$ are all close to 0. We can hence use (Aamari & Levrard, 2019, Theorem 6) to conclude that

$$d_H(\mathcal{M}^\star, \mathcal{M}_{\hat{\eta}}) = \mathcal{O}(h^{\beta-1})$$

with probability at least $1 - \mathcal{O}\left(\left(\frac{1}{N}\right)^{\frac{\beta}{k}}\right)$ if we take $h = \Theta\left(\left(\frac{\log N}{N}\right)^{\frac{1}{k}}\right)$.

### H.3.1 MISSING PROOFS: THE CONSTRAINT IN EQUATION (H.24)

Under the condition that $\|z\| \leq h$, we have

$$\text{dist}^2(x, \mathcal{M}_\eta) = \|x - \pi_\eta(x)\|^2 \leq \|x - x_{\text{ref}}\|^2 = \|x_0 + z - x_{\text{ref}}\|^2 \leq 2\|x_0 - x_{\text{ref}}\|^2 + 2\|z\|^2 = 4h^2.$$

Moreover, considering only the error in the tangent space, we have

$$\|x - \pi_\eta(x)\|^2 \geq \|W_{\text{ref}}^\top(x - x_{\text{ref}}) - W_{\text{ref}}^\top(\pi_\eta(x) - x_{\text{ref}})\|^2 = \|v + z_T - \hat{u}_\eta(x)\|^2 \geq \frac{1}{2}\|v - \hat{u}_\eta(x)\|^2 - 2\|z_T\|^2.$$

Combining these two inequalities together, we have

$$\|v - \hat{u}_\eta(x)\|^2 \leq 12h^2.$$

Hence the constraint in Equation (H.24) is inactive, if we take $C_h > 12$.

### H.3.2 MISSING PROOFS: BOUND ON $\|\Delta_h^*\|$

For a sufficiently small $h$, $A$ is close to identity. All we need to bound is $\|b\|$. Note that both $\|z_T\|$ and $\|z_N\|$ are bounded by $\|z\|$ and $\mathbf{J}_v$ is bounded in operator norm. We hence only need to bound $\|N_{\text{ref}}(v) - N_\eta(v)\|$.

Using $N_{\text{ref}}(0) = N_\eta(0) = 0$ and $DN_{\text{ref}}(0) = DN_\eta(0) = 0$, that we have

$$\|N_{\text{ref}}(v) - N_\eta(v)\| = \|N_{\text{ref}}(v) - N_{\text{ref}}(0) - (N_\eta(v) - N_\eta(0))\|$$
$$\leq \underbrace{\|N_{\text{ref}}(v) - N_{\text{ref}}(0)\|}_{\mathcal{O}(\|v\|^2)} + \underbrace{\|N_\eta(v) - N_\eta(0)\|}_{\mathcal{O}(\|v\|)} = \mathcal{O}(\|v\|) = \mathcal{O}(h),$$

where the first term is of higher order since by definition $D[N_{\text{ref}}](0) = 0$ (see Theorem F.1) and the estimation of the second term is from the Lipschitz continuity of $D[N_\eta]$ in Theorem G.11.

Combining the argument above, we have the conclusion $\|\Delta^*\| = \mathcal{O}(h)$. Hence if we take $C_h > \max 12, L_1'$, the constraint is not active. Here we recall the definition of $L_1'$ in Theorem G.11.

### H.4 HAUSDORFF CLOSENESS IMPLIES PROJECTION CLOSENESS

**Lemma H.3 (Hausdorff closeness implies projection closeness)** *Let* $\mathcal{M}, \widehat{\mathcal{M}} \subset \mathbb{R}^D$ *be closed embedded submanifolds, and assume*

$$\text{reach}(\mathcal{M}) \geq \zeta_{\min}, \qquad \text{reach}(\widehat{\mathcal{M}}) \geq \zeta_{\min},^6 \qquad \text{d}_\mathcal{H}(\mathcal{M}, \widehat{\mathcal{M}}) \leq \varepsilon,$$

*for some* $\zeta_{\min} > 0$ *and* $\varepsilon \in (0, \zeta_{\min}/4)$. *Fix any* $r \in (0, \zeta_{\min} - 2\varepsilon)$ *and let* $\text{Proj}_\mathcal{M} : \mathcal{T}_{\zeta_{\min}}(\mathcal{M}) \to \mathcal{M}$ *and* $\text{Proj}_{\widehat{\mathcal{M}}} : \mathcal{T}_{\zeta_{\min}}(\widehat{\mathcal{M}}) \to \widehat{\mathcal{M}}$ *denote the nearest-point projections. Then, for every* $x \in \mathcal{T}_r(\mathcal{M})$, *both* $\text{Proj}_\mathcal{M}(x)$ *and* $\text{Proj}_{\widehat{\mathcal{M}}}(x)$ *are well-defined and*

$$\|\text{Proj}_\mathcal{M}(x) - \text{Proj}_{\widehat{\mathcal{M}}}(x)\| \leq \varepsilon + 2\sqrt{\frac{\text{dist}(x, \mathcal{M})\,\varepsilon + \varepsilon^2}{1 - (\text{dist}(x, \mathcal{M}) + \varepsilon)/\zeta_{\min}}}. \tag{H.37}$$

*In particular, taking the supremum over* $x \in \mathcal{T}_r(\mathcal{M})$ *yields the uniform bound*

$$\|\text{Proj}_\mathcal{M} - \text{Proj}_{\widehat{\mathcal{M}}}\|_{L^\infty(\mathcal{T}_r(\mathcal{M}))} \leq \varepsilon + 2\sqrt{\frac{r\,\varepsilon + \varepsilon^2}{1 - (r + \varepsilon)/\zeta_{\min}}} \lesssim_{\zeta_{\min}, r} \sqrt{\varepsilon}. \tag{H.38}$$

**Proof.** Fix $x \in \mathcal{T}_r(\mathcal{M})$ and set

$$p := \text{Proj}_\mathcal{M}(x) \in \mathcal{M}, \quad q := \text{Proj}_{\widehat{\mathcal{M}}}(x) \in \widehat{\mathcal{M}}, \quad d := \|x - q\| = \text{dist}(x, \widehat{\mathcal{M}}), \quad d_M := \|x - p\| = \text{dist}(x, \mathcal{M}).$$

---

[6]We apply Theorem H.3 with $\mathcal{M}$ the true manifold and

$$\widehat{\mathcal{M}} := \{x \in \mathbb{U} : \hat{s}(x, t) = 0\},$$

where $\hat{s}$ denotes the estimated score in Theorem 2.2. The manifold $\mathcal{M}$ has positive reach by assumption, and the analogous reach property for $\widehat{\mathcal{M}}$ follows from the proof in Section H.

Since $d_{\mathcal{H}}(\mathcal{M}, \widehat{\mathcal{M}}) \leq \varepsilon$, there exists $y \in \widehat{\mathcal{M}}$ such that

$$\|y - p\| \leq \varepsilon. \tag{H.39}$$

Consequently,

$$\|x - y\| \leq \|x - p\| + \|p - y\| \leq d_M + \varepsilon. \tag{H.40}$$

Also, Hausdorff closeness implies $d = \mathrm{dist}(x, \widehat{\mathcal{M}}) \geq \mathrm{dist}(x, \mathcal{M}) - \varepsilon = d_M - \varepsilon$, hence

$$\|x - y\| \leq d_M + \varepsilon \leq d + 2\varepsilon. \tag{H.41}$$

**Step 1: reach inequality on $\widehat{\mathcal{M}}$.** Since $\mathrm{reach}(\widehat{\mathcal{M}}) \geq \zeta_{\min}$ and $x \in \mathcal{T}_r(\mathcal{M})$ with $r < \zeta_{\min} - 2\varepsilon$, we have

$$d = \mathrm{dist}(x, \widehat{\mathcal{M}}) \leq \mathrm{dist}(x, \mathcal{M}) + d_{\mathcal{H}}(\mathcal{M}, \widehat{\mathcal{M}}) \leq r + \varepsilon < \zeta_{\min},$$

so $x \in \mathcal{T}_{\zeta_{\min}}(\widehat{\mathcal{M}})$ and $q = \mathrm{Proj}_{\widehat{\mathcal{M}}}(x)$ is uniquely defined. A standard consequence of positive reach (see, e.g., Federer's theory of sets with positive reach) is that for every $y \in \widehat{\mathcal{M}}$,

$$\|x - y\|^2 \geq \|x - q\|^2 + \left(1 - \frac{\|x - q\|}{\zeta_{\min}}\right)\|y - q\|^2 = d^2 + \left(1 - \frac{d}{\zeta_{\min}}\right)\|y - q\|^2. \tag{H.42}$$

(One way to derive (H.42) is to use the hypomonotonicity inequality for the normal cone; it is the same inequality used in Theorem I.4 to pass from ambient "near-optimality" to a chord bound.)

Combining (H.42) with (H.41) gives

$$\left(1 - \frac{d}{\zeta_{\min}}\right)\|y - q\|^2 \leq \|x - y\|^2 - d^2 \leq (d + 2\varepsilon)^2 - d^2 = 4d\varepsilon + 4\varepsilon^2,$$

and therefore

$$\|y - q\| \leq 2\sqrt{\frac{d\varepsilon + \varepsilon^2}{1 - d/\zeta_{\min}}}. \tag{H.43}$$

**Step 2: conclude by the triangle inequality.** Using (H.39) and (H.43),

$$\|p - q\| \leq \|p - y\| + \|y - q\| \leq \varepsilon + 2\sqrt{\frac{d\varepsilon + \varepsilon^2}{1 - d/\zeta_{\min}}}.$$

Finally, since $d \leq d_M + \varepsilon = \mathrm{dist}(x, \mathcal{M}) + \varepsilon$, we have $1 - d/\zeta_{\min} \geq 1 - (\mathrm{dist}(x, \mathcal{M}) + \varepsilon)/\zeta_{\min}$, and substituting this bound proves (H.37). Taking the supremum over $x \in \mathcal{T}_r(\mathcal{M})$ yields (H.38). ∎

## H.5    Proof of Theorem 2.2 conditioned on Theorem E.1

We now show that the Hausdorff approximation guarantee can be improved from $\tilde{\mathcal{O}}\big(N^{-(\beta-1)/k}\big)$ in Theorem E.1 to $\tilde{\mathcal{O}}\big(N^{-\beta/k}\big)$, as in Theorem 2.2.

First, given Theorem E.1, and provided the number of samples $N$ is sufficiently large,

$$d_H(\mathcal{M}^\star, \mathcal{M}_{\hat{\eta}}) \leq \zeta_{\min}/8 \Rightarrow \mathcal{M}_{\hat{\eta}} \subset \mathbb{U}_2. \tag{H.44}$$

Here, the set inclusion can easily be shown via contradiction.

Recall that for any $\eta \in \mathcal{D}_{\mathbf{L}}^k$, $\eta$ is the squared distance to $\mathcal{M}_\eta = \{x \in \mathbb{U} : \eta(x) = 0\}$ on $\mathbb{U}_2$. Consequently, $\mathbb{U}_2$ is an open domain that contains the entirety of $\mathcal{M}_{\hat{\eta}}$, on which $\hat{\eta}$ is $\mathcal{C}^\beta$. Using the Poly-Raby Theorem (see, for example, (Denkowski, 2019, Theorem 2.14)), we have that $\widehat{\mathcal{M}}$ is $\mathcal{C}^\beta$.

We can then exactly follow the proof of Theorem E.1, concretely Sections G.3, H.1 and H.2 (replacing $\beta - 1$ with $\beta$ therein), again to derive the improved approximation guarantee.

# I   AUXILIARY RESULTS

## I.1   EXPERIMENTAL SETUP FOR FIGURE 1

We train score-based diffusion models to learn distributions over the rotation group $SO(d)$. Training data consists of rotation matrices sampled from either the Haar measure or a Projected Normal distribution on $SO(d)$, with $d = 5$. The model is trained with a continuous-time Variance-Preserving (VP) noise schedule using the standard denoising score matching (DSM) objective. We vary the training set size $n$, network capacity (width and depth), and regularisation strength across six preset configurations, spanning a spectrum from strong memorisation (small $n$, large models, weak regularisation) to generalisation (large $n$, smaller models, moderate regularisation). During training, we track several diagnostic metrics: alignment of the predicted score with the ideal denoising direction, the tangent-to-normal ratio of gradients on $SO(d)$, and nearest-neighbour distances between generated and training/test samples. We quantify memorisation by comparing the ratio of the first to second nearest-neighbour distances in the training set for each generated sample, and report the fraction of generated samples classified as memorised.

**Architecture.**   The score model is an MLP with residual blocks. Time is embedded via Fourier features of the log-SNR, processed through a two-layer MLP producing a 128-dimensional conditioning vector. Input rotation matrices are flattened to $\mathbb{R}^{d^2}$ and projected to the hidden dimension. The backbone consists of residual blocks each containing two LayerNorm + Linear layers with SiLU activations and additive time conditioning. The output is scaled by $1/\sigma(t)$ to parameterise the score.

**Training.**   The loss is the variance-weighted DSM objective:

$$\mathcal{L} = \mathbb{E}_{t, x_0, \epsilon}\left[\mathrm{Var}(t) \cdot \left\|s_\theta(x_t, t) - \left(-\epsilon/\sigma(t)\right)\right\|^2\right], \tag{I.1}$$

where $x_t = \alpha(t)\, x_0 + \sigma(t)\, \epsilon$ and $\mathrm{Var}(t) = 1 - \bar{\alpha}(t)$. Optimisation uses AdamW with gradient clipping (max norm 1.0).

**Sampling.**   Samples are drawn using annealed Langevin dynamics over 16 noise levels linearly spaced from $t{=}1$ to $t_{\min}$, with 60 Langevin steps per level. Generated samples are projected onto $SO(d)$ via SVD for evaluation.

**Memorisation metric.**   For each generated sample, we compute the Frobenius distances to all training points. The ratio $d_1^2/d_2^2$ of the squared distance to the nearest and second-nearest training point is computed; samples with ratio $< 0.5$ are classified as memorised.

**Configuration presets.**   Table I.1 summarises the six configurations used in our experiments. All experiments use $d{=}5$, batch size 512, and an evaluation set of $2\,000$ fresh samples. Each configuration is swept over data distributions (Haar, Projected Normal with $\sigma \in \{0.2, 1.0\}$) and random seeds.

Table I.1: Hyperparameter presets spanning from memorisation to generalisation.

| Preset | $n_{\mathrm{train}}$ | Hidden | Layers | Weight Decay | Steps | LR | $\beta_{\max}$ | $t_{\min}$ |
|---|---|---|---|---|---|---|---|---|
| deep_memo | 50 | 2048 | 8 | $10^{-8}$ | 20 000 | $10^{-3}$ | 20.0 | $10^{-5}$ |
| fast_memo | 100 | 1024 | 6 | $10^{-8}$ | 20 000 | $10^{-3}$ | 20.0 | $10^{-4}$ |
| std_small | 100 | 512 | 4 | $10^{-6}$ | 10 000 | $5 \times 10^{-4}$ | 10.0 | $10^{-4}$ |
| std_med | 200 | 512 | 4 | $10^{-6}$ | 10 000 | $2 \times 10^{-4}$ | 10.0 | $10^{-3}$ |
| rob_med | 200 | 512 | 3 | $10^{-2}$ | 10 000 | $2 \times 10^{-4}$ | 5.0 | $10^{-3}$ |
| gen | 1000 | 512 | 3 | $10^{-6}$ | 5 000 | $2 \times 10^{-4}$ | 5.0 | $10^{-3}$ |

## I.2   CONVOLUTION SIMPLIFIES ESTIMATION

The key step in the proof of Theorem 2.1 is the following:

**Theorem I.1 (Explicit $1/N$ rate in KL after Gaussian smoothing)** *Let $\mu$ be a probability measure on $\mathbb{R}^d$ supported on the Euclidean ball $B(0, R)$. Let $X_1, \ldots, X_N \overset{iid}{\sim} \mu$ and let $\mu_N :=\frac{1}{N} \sum_{i=1}^N \delta_{X_i}$. Fix $\sigma > 0$ and denote by*

$$\varphi_\sigma(x) := (2\pi\sigma^2)^{-d/2} \exp\left(-\frac{\|x\|^2}{2\sigma^2}\right)$$

*the density of $\mathcal{N}(0, \sigma^2 I_d)$. Define the smoothed densities*

$$p(x) := (\mu * \varphi_\sigma)(x) = \mathbb{E}\big[\varphi_\sigma(x - X)\big], \qquad q_N(x) := (\mu_N * \varphi_\sigma)(x) = \frac{1}{N} \sum_{i=1}^N \varphi_\sigma(x - X_i),$$

*where $X \sim \mu$ is an independent copy.*

*Then, for every $N \geq 1$ and every $\delta \in (0, 1)$, with probability at least $1 - \delta$,*

$$\mathrm{KL}(p\|q_N) \leq \frac{2^{d/2}}{N} \exp\left(\frac{17}{2} \frac{R^2}{\sigma^2}\right) \left(1 + \sqrt{2\log(1/\delta)}\right)^2. \tag{I.2}$$

*In particular, for any $a > 0$, with probability at least $1 - N^{-a}$,*

$$\mathrm{KL}(p\|q_N) \leq \frac{2^{d/2}}{N} \exp\left(\frac{17}{2} \frac{R^2}{\sigma^2}\right) \left(1 + \sqrt{2a\log N}\right)^2.$$

We will need the following standard lemma.

**Lemma I.2 (A basic KL–$\chi^2$ upper bound)** *Let $p, q$ be densities with $q > 0$ almost everywhere. Then*

$$\mathrm{KL}(p\|q) = \int_{\mathbb{R}^d} p(x) \log \frac{p(x)}{q(x)} \, dx \leq \int_{\mathbb{R}^d} \frac{(p(x) - q(x))^2}{q(x)} \, dx.$$

**Proof.** For $u > 0$ we have $\log u \leq u - 1$. With $u(x) = p(x)/q(x)$,

$$\mathrm{KL}(p\|q) = \int p \log u \leq \int p(u - 1) = \int \left(\frac{p^2}{q} - p\right) = \int \frac{p^2}{q} - 1,$$

since $\int p = 1$. Moreover,

$$\int \frac{(p - q)^2}{q} = \int \left(\frac{p^2}{q} - 2p + q\right) = \int \frac{p^2}{q} - 2\int p + \int q = \int \frac{p^2}{q} - 1$$

because $\int q = 1$. Combining the two displays yields the claim. ■

**Proof.** [Proof of Theorem I.1]

**Step 1: Lower bound $q_N$ deterministically.** Since $\mathrm{supp}(\mu) \subseteq B(0, R)$, we have $\|X_i\| \leq R$ almost surely. Fix $x \in \mathbb{R}^d$. Then for each $i$,

$$\|x - X_i\| \leq \|x\| + \|X_i\| \leq \|x\| + R,$$

hence

$$\varphi_\sigma(x - X_i) \geq (2\pi\sigma^2)^{-d/2} \exp\left(-\frac{(\|x\| + R)^2}{2\sigma^2}\right). \tag{I.3}$$

Averaging over $i$ gives the deterministic pointwise bound

$$q_N(x) \geq \underline{q}(x) := (2\pi\sigma^2)^{-d/2} \exp\left(-\frac{(\|x\| + R)^2}{2\sigma^2}\right) \qquad (\forall x \in \mathbb{R}^d). \tag{I.4}$$

**Step 2: Reduce KL to a weighted $L^2$ norm.** By Lemma I.2,

$$\mathrm{KL}(p\|q_N) \leq \int \frac{(p - q_N)^2}{q_N}.$$

Using (I.4) (so $1/q_N \leq 1/\underline{q}$),

$$\mathrm{KL}(p\|q_N) \leq \int_{\mathbb{R}^d} \frac{(p(x) - q_N(x))^2}{\underline{q}(x)} \, dx. \tag{I.5}$$

Introduce the Hilbert space

$$\mathcal{H} := L^2\big(\underline{q}(x)^{-1}dx\big)$$

with norm

$$\|f\|_{\mathcal{H}}^2 := \int_{\mathbb{R}^d} \frac{f(x)^2}{\underline{q}(x)} \, dx.$$

Then (I.5) reads

$$\mathrm{KL}(p\|q_N) \leq \|q_N - p\|_{\mathcal{H}}^2. \tag{I.6}$$

**Step 3: A uniform bound on the kernel in $\mathcal{H}$.** For $y \in B(0, R)$,

$$\|\varphi_\sigma(\cdot - y)\|_{\mathcal{H}}^2 = \int_{\mathbb{R}^d} \frac{\varphi_\sigma(x - y)^2}{\underline{q}(x)} \, dx.$$

Since $\|y\| \leq R$, we have

$$\|x - y\| \geq \big|\|x\| - \|y\|\big| \geq \|x\| - R,$$

hence

$$\varphi_\sigma(x - y)^2 \leq (2\pi\sigma^2)^{-d} \exp\left(-\frac{(\|x\| - R)^2}{\sigma^2}\right).$$

Using the definition of $\underline{q}$,

$$\|\varphi_\sigma(\cdot - y)\|_{\mathcal{H}}^2 \leq (2\pi\sigma^2)^{-d/2} \int_{\mathbb{R}^d} \exp\left(-\frac{(\|x\| - R)^2}{\sigma^2} + \frac{(\|x\| + R)^2}{2\sigma^2}\right) \, dx.$$

As in the original Gaussian integral computation, writing $r = \|x\|$ gives

$$-\frac{(r - R)^2}{\sigma^2} + \frac{(r + R)^2}{2\sigma^2} = \frac{-\frac{1}{2}r^2 + 3Rr - \frac{1}{2}R^2}{\sigma^2},$$

and using

$$3Rr \leq \frac{r^2}{4} + 9R^2$$

yields

$$-\frac{r^2}{2\sigma^2} + \frac{3Rr}{\sigma^2} \leq -\frac{r^2}{4\sigma^2} + \frac{9R^2}{\sigma^2}.$$

Therefore,

$$\|\varphi_\sigma(\cdot - y)\|_{\mathcal{H}}^2 \leq (2\pi\sigma^2)^{-d/2} \exp\left(-\frac{R^2}{2\sigma^2} + \frac{9R^2}{\sigma^2}\right) \int_{\mathbb{R}^d} \exp\left(-\frac{\|x\|^2}{4\sigma^2}\right) \, dx.$$

Since

$$\int_{\mathbb{R}^d} \exp\left(-\frac{\|x\|^2}{4\sigma^2}\right) \, dx = (4\pi\sigma^2)^{d/2},$$

we obtain

$$\sup_{\|y\| \leq R} \|\varphi_\sigma(\cdot - y)\|_{\mathcal{H}}^2 \leq 2^{d/2} \exp\left(\frac{17}{2}\frac{R^2}{\sigma^2}\right) =: A_\sigma. \tag{I.7}$$

**Step 4: Center the empirical process in $\mathcal{H}$.** Define the $\mathcal{H}$-valued random variables

$$Z_i := \varphi_\sigma(\cdot - X_i) - p.$$

Since

$$p = \mathbb{E}\big[\varphi_\sigma(\cdot - X)\big],$$

we have $\mathbb{E}[Z_i] = 0$ in $\mathcal{H}$, and

$$q_N - p = \frac{1}{N} \sum_{i=1}^{N} Z_i.$$

Hence, by (I.6),

$$\mathrm{KL}(p\|q_N) \leq \left\| \frac{1}{N} \sum_{i=1}^{N} Z_i \right\|_{\mathcal{H}}^2. \tag{I.8}$$

Also, by Jensen's inequality and (I.7),

$$\|p\|_{\mathcal{H}} = \big\| \mathbb{E}\big[\varphi_\sigma(\cdot - X)\big] \big\|_{\mathcal{H}} \leq \mathbb{E}\|\varphi_\sigma(\cdot - X)\|_{\mathcal{H}} \leq \sqrt{A_\sigma}.$$

Therefore, for every realization of $X_i$,

$$\|Z_i\|_{\mathcal{H}} \leq \|\varphi_\sigma(\cdot - X_i)\|_{\mathcal{H}} + \|p\|_{\mathcal{H}} \leq 2\sqrt{A_\sigma}.$$

**Step 5: Bound the mean square of the empirical average.** Because the $Z_i$ are independent and mean zero in the Hilbert space $\mathcal{H}$,

$$\mathbb{E} \left\| \frac{1}{N} \sum_{i=1}^{N} Z_i \right\|_{\mathcal{H}}^2 = \frac{1}{N^2} \sum_{i=1}^{N} \mathbb{E}\|Z_i\|_{\mathcal{H}}^2 = \frac{1}{N} \mathbb{E}\|Z_1\|_{\mathcal{H}}^2.$$

Moreover,

$$\mathbb{E}\|Z_1\|_{\mathcal{H}}^2 = \mathbb{E}\|\varphi_\sigma(\cdot - X) - p\|_{\mathcal{H}}^2 = \mathbb{E}\|\varphi_\sigma(\cdot - X)\|_{\mathcal{H}}^2 - \|p\|_{\mathcal{H}}^2 \leq A_\sigma,$$

so

$$\mathbb{E} \left\| \frac{1}{N} \sum_{i=1}^{N} Z_i \right\|_{\mathcal{H}}^2 \leq \frac{A_\sigma}{N}. \tag{I.9}$$

By Cauchy–Schwarz,

$$\mathbb{E} \left\| \frac{1}{N} \sum_{i=1}^{N} Z_i \right\|_{\mathcal{H}} \leq \sqrt{\frac{A_\sigma}{N}}. \tag{I.10}$$

**Step 6: Concentrate via McDiarmid's inequality.** Set

$$f(X_1, \ldots, X_N) := \left\| \frac{1}{N} \sum_{i=1}^{N} Z_i \right\|_{\mathcal{H}}.$$

If only the $i$-th sample is changed from $X_i$ to $X_i'$, then

$$f(X_1, \ldots, X_i, \ldots, X_N) - f(X_1, \ldots, X_i', \ldots, X_N)$$

has absolute value at most

$$\frac{1}{N}\|\varphi_\sigma(\cdot - X_i) - \varphi_\sigma(\cdot - X_i')\|_{\mathcal{H}} \leq \frac{2\sqrt{A_\sigma}}{N}.$$

Thus $f$ satisfies the bounded-differences condition with constants $c_i = 2\sqrt{A_\sigma}/N$. McDiarmid's inequality gives that, for every $\delta \in (0, 1)$, with probability at least $1 - \delta$,

$$f \leq \mathbb{E}f + \sqrt{\frac{1}{2}\Big(\sum_{i=1}^{N} c_i^2\Big) \log(1/\delta)}.$$

Since

$$\sum_{i=1}^{N} c_i^2 = N \cdot \frac{4A_\sigma}{N^2} = \frac{4A_\sigma}{N},$$

we obtain, using (I.10),

$$f \leq \sqrt{\frac{A_\sigma}{N}} + \sqrt{\frac{2A_\sigma \log(1/\delta)}{N}} = \sqrt{\frac{A_\sigma}{N}}\Big(1 + \sqrt{2\log(1/\delta)}\Big).$$

Squaring and using (I.8) yields that with probability at least $1 - \delta$,

$$\mathrm{KL}(p\|q_N) \leq \frac{A_\sigma}{N}\Big(1 + \sqrt{2\log(1/\delta)}\Big)^2.$$

Recalling the definition

$$A_\sigma = 2^{d/2} \exp\left(\frac{17}{2}\frac{R^2}{\sigma^2}\right),$$

this is exactly (I.2).

Finally, substituting $\delta = N^{-a}$ gives the stated $1 - N^{-a}$ bound. ∎

## I.3   AUXILIARY GEOMETRIC LEMMAS

This section collects several geometric lemmas—each provable by standard arguments—that we will invoke in later proofs.

**Lemma I.3 (Euclidean displacement under projection)** *Let $\mathcal{M} \subset \mathbb{R}^D$ be closed and let $\mathrm{Proj}_{\mathcal{M}}$ be the nearest-point projection defined on a set containing $y + v$. If $y \in \mathcal{M}$ and $v \in \mathbb{R}^D$ are such that $\mathrm{Proj}_{\mathcal{M}}(y + v)$ is defined, then*

$$\|\mathrm{Proj}_{\mathcal{M}}(y + v) - y\| \ \leq \ 2\|v\|.$$

**Proof.**   By the triangle inequality,

$$\|\mathrm{Proj}_{\mathcal{M}}(y + v) - y\| \leq \|\mathrm{Proj}_{\mathcal{M}}(y + v) - (y + v)\| + \|v\|.$$

Since $y \in \mathcal{M}$ is a feasible competitor in the minimization defining $\mathrm{Proj}_{\mathcal{M}}(y + v)$,

$$\|\mathrm{Proj}_{\mathcal{M}}(y + v) - (y + v)\| \leq \|y - (y + v)\| = \|v\|.$$

Combining yields $\|\mathrm{Proj}_{\mathcal{M}}(y + v) - y\| \leq 2\|v\|$. ∎

**Lemma I.4 (Ambient near-optimality implies small geodesic shift)** *Let $\mathcal{M} \subset \mathbb{R}^D$ be an embedded submanifold with reach $\zeta_{\min} := \mathrm{reach}(\mathcal{M}) > 0$, and let $\mathrm{Proj}_{\mathcal{M}} : \mathcal{T}_{\zeta_{\min}}(\mathcal{M}) \to \mathcal{M}$ denote the nearest-point projection. Let $d_{\mathcal{M}}$ be the geodesic distance on $\mathcal{M}$. For any $x \in \mathcal{T}_{\zeta_{\min}}(\mathcal{M})$ with $d := \mathrm{dist}(x, \mathcal{M}) < \zeta_{\min}$ and any $y \in \mathcal{M}$ satisfying*

$$\|x - y\| \ \leq \ d + \varepsilon',$$

*one has*

$$d_{\mathcal{M}}^2\big(\mathrm{Proj}_{\mathcal{M}}(x), y\big) \ \leq \ C_{\mathrm{geo}}\frac{2\,d\,\varepsilon' + (\varepsilon')^2}{1 - d/\zeta_{\min}},$$

*where $C_{\mathrm{geo}}$ is an absolute constant (e.g. one may take $C_{\mathrm{geo}} = 4$ whenever $\|y - \mathrm{Proj}_{\mathcal{M}}(x)\| \leq \zeta_{\min}/2$, and $C_{\mathrm{geo}} = \pi^2/4$ whenever $\|y - \mathrm{Proj}_{\mathcal{M}}(x)\| \leq \zeta_{\min}$).*

**Proof.**   Let $p := \mathrm{Proj}_{\mathcal{M}}(x) \in \mathcal{M}$ and $n := x - p$, so that $\|n\| = d$ and $n \perp T_p\mathcal{M}$. We proceed in two steps.

**Step 1: reach inequality $\Rightarrow$ chord control.** A standard consequence of positive reach (often stated as a "hypomonotonicity inequality"; see, e.g., Federer (1969)) is that for every $y \in \mathcal{M}$,

$$\langle n,\ y - p \rangle \ \leq\ \frac{\|n\|}{2\zeta_{\min}}\,\|y-p\|^2 \ =\ \frac{d}{2\zeta_{\min}}\,\|y-p\|^2. \tag{I.11}$$

Expanding $\|x-y\|^2 = \|n-(y-p)\|^2$ and using (I.11) yields

$$\|x-y\|^2 = d^2 + \|y-p\|^2 - 2\langle n, y-p\rangle \ \geq\ d^2 + \left(1 - \frac{d}{\zeta_{\min}}\right)\|y-p\|^2.$$

Rearranging gives

$$\|y-p\|^2 \ \leq\ \frac{\|x-y\|^2 - d^2}{1 - d/\zeta_{\min}}. \tag{I.12}$$

By the near-optimality assumption $\|x-y\| \leq d + \varepsilon'$,

$$\|x-y\|^2 - d^2 \ \leq\ (d+\varepsilon')^2 - d^2 = 2d\,\varepsilon' + (\varepsilon')^2,$$

and hence

$$\|y-p\|^2 \ \leq\ \frac{2d\,\varepsilon' + (\varepsilon')^2}{1 - d/\zeta_{\min}}. \tag{I.13}$$

**Step 2: chord–arc comparability $\Rightarrow$ geodesic control.** Let $\gamma : [0,\ell] \to \mathcal{M}$ be a unit-speed minimizing geodesic from $p$ to $y$, so $\ell = d_{\mathcal{M}}(p,y)$. Since $\mathcal{M}$ has reach $\zeta_{\min}$, its second fundamental form is bounded in operator norm by $1/\zeta_{\min}$, and therefore the ambient curvature of $\gamma$ satisfies $\|\ddot{\gamma}(s)\| \leq 1/\zeta_{\min}$ for all $s$. A standard chord–arc inequality for $C^2$ curves with curvature bounded by $1/\zeta_{\min}$ implies that, whenever $\|y-p\| \leq \zeta_{\min}/2$, one has $\ell \leq 2\|y-p\|$ (and whenever $\|y-p\| \leq \zeta_{\min}$, one has $\ell \leq (\pi/2)\|y-p\|$). Consequently,

$$d_{\mathcal{M}}^2(p,y) = \ell^2 \ \leq\ C_{\mathrm{geo}}\,\|y-p\|^2,$$

with $C_{\mathrm{geo}} = 4$ (resp. $C_{\mathrm{geo}} = \pi^2/4$) under the corresponding local condition. Combining with (I.13) yields

$$d_{\mathcal{M}}^2\big(\mathrm{Proj}_{\mathcal{M}}(x),y\big) \ \leq\ C_{\mathrm{geo}}\,\frac{2d\,\varepsilon' + (\varepsilon')^2}{1 - d/\zeta_{\min}},$$

as claimed. ∎

## J  PROOFS FOR NORMAL AND TANGENTIAL DRIFTS

**Controlling normal and tangential drifts.** Recall the reverse-time probability-flow ODE associated with the learned score (cf. (8)):

$$\mathrm{d}X_t \ =\ -\tfrac{1}{2}\,\hat{s}(X_t, t)\,\mathrm{d}t, \qquad t : t_0 \searrow \tau, \tag{J.1}$$

where $t_0 > 0$ is the terminal-time threshold in Section 2.3 and $\tau \in (0, t_0)$ is a fixed cutoff to be chosen later. It is convenient to reparametrize in forward time by $\bar{X}_t := X_{t_0-t}$ for $t \in [0, t_0 - \tau]$, which yields

$$\mathrm{d}\bar{X}_t \ =\ \tfrac{1}{2}\,\hat{s}(\bar{X}_t, t_0 - t)\,\mathrm{d}t, \qquad t : 0 \nearrow t_0 - \tau. \tag{J.2}$$

Under the terminal score model (18)–(19), the dominant component of the drift is the normal "pull" $-(x - \mathrm{Proj}_{\mathcal{M}}(x))/t$, which contracts trajectories toward $\mathcal{M}^\star$. The next lemma makes this quantitative and shows that the flow drives points into an $\tilde{\mathcal{O}}(\varepsilon)$-tube around $\mathcal{M}^\star$.

**Lemma J.1 (Contraction to an $\varepsilon$-tube)** *Assume $\bar{X}_0 \in \mathcal{T}_{\zeta_{\min}/4}(\mathcal{M}^\star)$. Under (18)–(19), the terminal point $\bar{X}_{t_0-\tau}$ satistifes*

$$\mathrm{dist}(\bar{X}_{t_0-\tau}, \mathcal{M}^\star) \ \leq\ \sqrt{2}\,\varepsilon \ +\ \mathrm{dist}(\bar{X}_0, \mathcal{M}^\star)\,\sqrt{\tau/t_0}.$$

*In particular, taking $\tau/t_0 = \varepsilon^3$ yields $\mathrm{dist}(\bar{X}_{t_0-\tau}, \mathcal{M}^\star) \ \lesssim\ \varepsilon$.*

Theorem J.1 bounds the *normal* error by showing that the terminal-time flow drives points into an $\tilde{\mathcal{O}}(\varepsilon)$-tube around $\mathcal{M}^\star$. This alone does not preclude large motion *along* $\mathcal{M}^\star$: a trajectory may stay close to $\mathcal{M}^\star$ while sliding far in geodesic distance. Thus we must also control the *tangential* displacement induced by the terminal map. Since $\mathrm{Proj}_{\widehat{\mathcal{M}}}(x)$ need not lie on $\mathcal{M}^\star$, we measure tangential motion via the "re-projection" $\mathrm{Proj}_{\mathcal{M}}(\mathrm{Proj}_{\widehat{\mathcal{M}}}(x)) \in \mathcal{M}^\star$.

**Lemma J.2 (Tangential drift bound)** *Assume the terminal score model* (18)–(19) *holds on* $\mathcal{T}_{\zeta_{\min}/4}(\mathcal{M}^\star)$, *and let* $\mathrm{Proj}_{\widehat{\mathcal{M}}}$ *denote the terminal-time map induced by the forward ODE* (J.2) *run on* $[0, t_0 - \tau]$. *Then, for any* $x \in \mathcal{T}_{\zeta_{\min}/4}(\mathcal{M}^\star)$, *the choice* $\tau = t_0 \varepsilon^3$ *yields*

$$d_{\mathcal{M}^\star}\big(\mathrm{Proj}_{\mathcal{M}}(x),\, \mathrm{Proj}_{\mathcal{M}}(\mathrm{Proj}_{\widehat{\mathcal{M}}}(x))\big) \;\leq\; \tilde{\mathcal{O}}\big(\sqrt{\varepsilon}\big).$$

## J.1 PROOF OF THEOREM J.1

**Proof of Theorem J.1.** Recall the forward-time terminal ODE

$$\dot{\bar{X}}_t \;=\; \tfrac{1}{2}\,\hat{s}(\bar{X}_t, t_0 - t), \qquad t \in [0, t_0 - \tau], \tag{J.3}$$

and the terminal score model (18)–(19): for all $t \in [\tau, t_0]$ and $x \in \mathrm{Tub}_r(\mathcal{M}^\star)$,

$$\hat{s}(x, t) \;=\; -\frac{x - \mathrm{Proj}_{\mathcal{M}}(x)}{t} \;+\; \frac{e(x, t)}{t}, \qquad \|e(x, t)\| \leq \varepsilon.$$

**Step 1: differentiate the squared distance.** Define

$$a_t \;:=\; \tfrac{1}{2}\,\mathrm{dist}^2(\bar{X}_t, \mathcal{M}^\star), \qquad t \in [0, t_0 - \tau].$$

On $\mathrm{Tub}_r(\mathcal{M}^\star)$ (with $r < \mathrm{reach}(\mathcal{M}^\star)$), the map $x \mapsto \tfrac{1}{2}\,\mathrm{dist}^2(x, \mathcal{M}^\star)$ is $C^1$ and

$$\nabla\Big(\tfrac{1}{2}\,\mathrm{dist}^2(x, \mathcal{M}^\star)\Big) \;=\; x - \mathrm{Proj}_{\mathcal{M}}(x).$$

Therefore, for a.e. $t \in [0, t_0 - \tau]$,

$$\dot{a}_t \;=\; \big\langle \bar{X}_t - \mathrm{Proj}_{\mathcal{M}}(\bar{X}_t),\, \dot{\bar{X}}_t \big\rangle. \tag{J.4}$$

**Step 2: plug in the terminal drift and bound.** Using (J.3) and the score model at time $t_0 - t$,

$$\dot{\bar{X}}_t = -\frac{1}{2(t_0 - t)}\big(\bar{X}_t - \mathrm{Proj}_{\mathcal{M}}(\bar{X}_t)\big) + \frac{1}{2(t_0 - t)}\,e(\bar{X}_t, t_0 - t).$$

Substituting into (J.4) yields

$$\dot{a}_t = -\frac{1}{2(t_0 - t)}\|\bar{X}_t - \mathrm{Proj}_{\mathcal{M}}(\bar{X}_t)\|^2 + \frac{1}{2(t_0 - t)}\big\langle \bar{X}_t - \mathrm{Proj}_{\mathcal{M}}(\bar{X}_t),\, e(\bar{X}_t, t_0 - t)\big\rangle.$$

Since $\|\bar{X}_t - \mathrm{Proj}_{\mathcal{M}}(\bar{X}_t)\|^2 = 2a_t$ and $\|e(\bar{X}_t, t_0 - t)\| \leq \varepsilon$, Cauchy–Schwarz gives

$$\dot{a}_t \leq -\frac{1}{t_0 - t}\,a_t + \frac{1}{2(t_0 - t)}\|\bar{X}_t - \mathrm{Proj}_{\mathcal{M}}(\bar{X}_t)\|\,\varepsilon = -\frac{1}{t_0 - t}\,a_t + \frac{\varepsilon}{2(t_0 - t)}\sqrt{2a_t} \;\leq\; -\frac{1}{t_0 - t}\,a_t + \frac{\varepsilon}{t_0 - t}\sqrt{a_t}. \tag{J.5}$$

**Step 3: solve the one-dimensional inequality.** Let $b_t := \sqrt{a_t}$. Whenever $b_t > 0$ we have $\dot{b}_t = \dot{a}_t/(2b_t)$, hence from (J.5)

$$\dot{b}_t \leq -\frac{1}{2(t_0 - t)}\,b_t + \frac{1}{2(t_0 - t)}\,\varepsilon.$$

(When $b_t = 0$, the same bound holds for the upper Dini derivative, so the comparison argument below remains valid.) Define $u_t := b_t - \varepsilon$. Then

$$\dot{u}_t \;\leq\; -\frac{1}{2(t_0 - t)}\,u_t,$$

so $t \mapsto u_t(t_0 - t)^{-1/2}$ is nonincreasing. Using $u_0 = b_0 - \varepsilon$ and $t_0 - t = t_0(1 - t/t_0)$, we obtain, for all $t \in [0, t_0 - \tau]$,

$$b_t \le \varepsilon + (b_0 - \varepsilon)\sqrt{1 - t/t_0}. \tag{J.6}$$

**Step 4: evaluate at terminal time.** At $t = t_0 - \tau$, (J.6) gives

$$\sqrt{a_{t_0-\tau}} \le \varepsilon + (\sqrt{a_0} - \varepsilon)\sqrt{\tau/t_0} \le \varepsilon + \sqrt{a_0}\sqrt{\tau/t_0}.$$

Recalling $\text{dist}(\bar{X}_t, \mathcal{M}^\star) = \sqrt{2a_t}$ and $\sqrt{a_0} = \text{dist}(\bar{X}_0, \mathcal{M}^\star)/\sqrt{2}$, we conclude

$$\text{dist}(\bar{X}_{t_0-\tau}, \mathcal{M}^\star) \le \sqrt{2}\,\varepsilon + \text{dist}(\bar{X}_0, \mathcal{M}^\star)\sqrt{\tau/t_0},$$

as claimed. Finally, taking $\tau/t_0 = \varepsilon^3$ yields $\text{dist}(\bar{X}_{t_0-\tau}, \mathcal{M}^\star) \le \sqrt{2}\varepsilon + \text{dist}(\bar{X}_0, \mathcal{M}^\star)\varepsilon^{3/2} \lesssim \varepsilon$ for $\varepsilon$ small (with the implicit constant depending on an a priori bound on $\text{dist}(\bar{X}_0, \mathcal{M}^\star)$, e.g. $\bar{X}_0 \in \text{Tub}_r(\mathcal{M}^\star)$). ∎

## J.2 PROOF OF THEOREM J.2

We first need a simple bound:

**Lemma J.3 (Terminal-time path-length bound)** *Let $\bar{X}_t$ solve the forward-time ODE (J.2) for $t \in [0, t_0 - \tau]$, and assume that the terminal score model* (18)–(19) *holds on $\mathcal{T}_r(\mathcal{M}^\star)$ for some $r < \text{reach}(\mathcal{M}^\star)$. Define $a_0 := \frac{1}{2}\,\text{dist}^2(\bar{X}_0, \mathcal{M}^\star)$ and suppose $\sqrt{a_0} \ge \varepsilon$. Then*

$$\|\bar{X}_{t_0-\tau} - \bar{X}_0\| \le \text{dist}(\bar{X}_0, \mathcal{M}^\star) + \mathcal{O}\Big(\varepsilon \ln \frac{t_0}{\tau}\Big).$$

**Proof.** Write $X_t := \bar{X}_t$ for readability. By the fundamental theorem of calculus,

$$\|X_{t_0-\tau} - X_0\| = \Big\| \int_0^{t_0-\tau} \dot{X}_t \, dt \Big\| \le \int_0^{t_0-\tau} \|\dot{X}_t\| \, dt.$$

We now bound the path length. By the forward-time ODE (J.2) and the terminal score model (18)–(19), for $t \in [0, t_0 - \tau]$ we have

$$\dot{X}_t = \frac{1}{2}\hat{s}(X_t, t_0-t) = -\frac{1}{2(t_0-t)}\big(X_t - \text{Proj}_{\mathcal{M}}(X_t)\big) + \frac{1}{2(t_0-t)}\,e(X_t, t_0-t), \qquad \|e(X_t, t_0-t)\| \le \varepsilon,$$

and hence

$$\|\dot{X}_t\| \le \frac{1}{2(t_0-t)}\,\text{dist}(X_t, \mathcal{M}^\star) + \frac{\varepsilon}{2(t_0-t)}. \tag{J.7}$$

Next, let $a_t := \frac{1}{2}\,\text{dist}^2(X_t, \mathcal{M}^\star)$. The distance estimate (J.6) (proved in Theorem J.1) yields, for all $t \in [0, t_0 - \tau]$,

$$\sqrt{a_t} \le \varepsilon + (\sqrt{a_0} - \varepsilon)\sqrt{1 - t/t_0} \le \varepsilon + \sqrt{a_0}\sqrt{\frac{t_0 - t}{t_0}},$$

where we used $\sqrt{a_0} \ge \varepsilon$. Recalling $\text{dist}(X_t, \mathcal{M}^\star) = \sqrt{2a_t}$ and $\text{dist}(X_0, \mathcal{M}^\star) = \sqrt{2a_0}$, we obtain

$$\text{dist}(X_t, \mathcal{M}^\star) \le \sqrt{2}\,\varepsilon + \text{dist}(X_0, \mathcal{M}^\star)\sqrt{\frac{t_0 - t}{t_0}}. \tag{J.8}$$

Plugging (J.8) into (J.7) gives

$$\|\dot{X}_t\| \le \frac{\text{dist}(X_0, \mathcal{M}^\star)}{2\sqrt{t_0}} \cdot \frac{1}{\sqrt{t_0 - t}} + \frac{1 + \sqrt{2}}{2} \cdot \frac{\varepsilon}{t_0 - t}.$$

Integrating from $t = 0$ to $t = t_0 - \tau$ and using the change of variables $u = t_0 - t$ yields

$$\int_0^{t_0-\tau} \|\dot{X}_t\| \, dt \le \frac{\text{dist}(X_0, \mathcal{M}^\star)}{2\sqrt{t_0}} \int_0^{t_0-\tau} \frac{dt}{\sqrt{t_0 - t}} + \frac{1 + \sqrt{2}}{2}\varepsilon \int_0^{t_0-\tau} \frac{dt}{t_0 - t}$$

$$= \frac{\text{dist}(X_0, \mathcal{M}^\star)}{2\sqrt{t_0}} \int_\tau^{t_0} \frac{du}{\sqrt{u}} + \frac{1 + \sqrt{2}}{2}\varepsilon \int_\tau^{t_0} \frac{du}{u}$$

$$= \frac{\text{dist}(X_0, \mathcal{M}^\star)}{2\sqrt{t_0}} \cdot 2\big(\sqrt{t_0} - \sqrt{\tau}\big) + \frac{1 + \sqrt{2}}{2}\varepsilon \ln \frac{t_0}{\tau}$$

$$\le \text{dist}(X_0, \mathcal{M}^\star) + \mathcal{O}\Big(\varepsilon \ln \frac{t_0}{\tau}\Big),$$

which is the desired bound (with an absolute implied constant).                            ∎

**Proof.** [Proof of Theorem J.2] Fix $x \in \mathcal{T}_r(\mathcal{M}^\star)$ and set $x = \bar{X}_0$, where $\bar{X}_t$ is the solution of (J.2). Define the on-manifold comparison point

$$y := \operatorname{Proj}_{\mathcal{M}}(\operatorname{Proj}_{\widehat{\mathcal{M}}}(x)) \in \mathcal{M}^\star.$$

We will verify the hypothesis of Theorem I.4 with $\varepsilon'$ of order $\tilde{\mathcal{O}}(\varepsilon)$, which readily completes the proof.

**Step 1: an ambient near-optimality bound.**    By the triangle inequality,

$$\|x-y\| \leq \|x-\operatorname{Proj}_{\widehat{\mathcal{M}}}(x)\| + \|\operatorname{Proj}_{\widehat{\mathcal{M}}}(x)-\operatorname{Proj}_{\mathcal{M}}(\operatorname{Proj}_{\widehat{\mathcal{M}}}(x))\| = \|x-\operatorname{Proj}_{\widehat{\mathcal{M}}}(x)\| + \operatorname{dist}(\operatorname{Proj}_{\widehat{\mathcal{M}}}(x), \mathcal{M}^\star). \tag{J.9}$$

The path-length bound Theorem J.3 gives

$$\|x - \operatorname{Proj}_{\widehat{\mathcal{M}}}(x)\| \leq \operatorname{dist}(x, \mathcal{M}^\star) + \mathcal{O}\Big(\varepsilon \ln \frac{t_0}{\tau}\Big). \tag{J.10}$$

Moreover, applying Theorem J.1 with initial condition $\bar{X}_0 = x$ yields

$$\operatorname{dist}(\operatorname{Proj}_{\widehat{\mathcal{M}}}(x), \mathcal{M}^\star) \leq \sqrt{2}\,\varepsilon + \operatorname{dist}(x, \mathcal{M}^\star)\sqrt{\tau/t_0}. \tag{J.11}$$

Combining (J.9)–(J.11), we obtain

$$\|x - y\| \leq \operatorname{dist}(x, \mathcal{M}^\star) + \varepsilon', \qquad \varepsilon' := \mathcal{O}\Big(\varepsilon \ln \frac{t_0}{\tau}\Big) + \sqrt{2}\,\varepsilon + \operatorname{dist}(x, \mathcal{M}^\star)\sqrt{\tau/t_0}. \tag{J.12}$$

Since $x \in \mathcal{T}_r(\mathcal{M}^\star)$, we have $\operatorname{dist}(x, \mathcal{M}^\star) \leq r$, hence for $\tau = t_0\varepsilon^3$,

$$\varepsilon' = \mathcal{O}\Big(\varepsilon \ln \frac{1}{\varepsilon}\Big) + \sqrt{2}\,\varepsilon + r\,\varepsilon^{3/2} = \tilde{\mathcal{O}}(\varepsilon).$$

**Step 2: transfer to geodesic distance.**    We may now apply Theorem I.4 with $x \leftarrow x$, $y \leftarrow y$, and $\varepsilon'$ as in (J.12). This gives

$$d_{\mathcal{M}^\star}^2\big(\operatorname{Proj}_{\mathcal{M}}(x), y\big) \lesssim \frac{2\operatorname{dist}(x, \mathcal{M}^\star)\,\varepsilon' + (\varepsilon')^2}{1 - \operatorname{dist}(x, \mathcal{M}^\star)/\zeta_{\min}}.$$

Using $\operatorname{dist}(x, \mathcal{M}^\star) \leq r$ and $1 - \operatorname{dist}(x, \mathcal{M}^\star)/\zeta_{\min} \geq 1 - r/\zeta_{\min}$, we obtain

$$d_{\mathcal{M}^\star}^2\big(\operatorname{Proj}_{\mathcal{M}}(x), y\big) \lesssim \frac{2r\,\varepsilon' + (\varepsilon')^2}{1 - r/\zeta_{\min}} \lesssim \varepsilon',$$

and hence

$$d_{\mathcal{M}^\star}\big(\operatorname{Proj}_{\mathcal{M}}(x), y\big) \lesssim \sqrt{\varepsilon'} = \tilde{\mathcal{O}}(\sqrt{\varepsilon}),$$

where we used $\varepsilon' = \tilde{\mathcal{O}}(\varepsilon)$ for $\tau = t_0\varepsilon^3$. Recalling $y = \operatorname{Proj}_{\mathcal{M}}(\operatorname{Proj}_{\widehat{\mathcal{M}}}(x))$ completes the proof.  ∎

# K   Coverage of the population surrogate $\widehat{\mu_{\mathrm{proj}}}$

Throughout, $\mathcal{M}^\star \subset \mathbb{R}^D$ is a closed $C^2$ embedded submanifold. We write $\operatorname{Proj}_{\mathcal{M}} : \operatorname{Tub}_{\zeta_{\min}}(\mathcal{M}^\star) \to \mathcal{M}^\star$ for the nearest-point projection, where

$$\zeta_{\min} := \operatorname{reach}(\mathcal{M}^\star) > 0, \qquad \operatorname{Tub}_r(\mathcal{M}^\star) := \{x \in \mathbb{R}^D : \operatorname{dist}(x, \mathcal{M}^\star) < r\}.$$

Let $d_{\mathcal{M}^\star}$ denote the geodesic distance on $\mathcal{M}^\star$, and $\operatorname{Vol}_{\mathcal{M}^\star}$ its Riemannian volume measure.

For $(\alpha, \delta)$ and $y \in \mathcal{M}^\star$, recall the thickened geodesic ball

$$B_{\delta,\alpha}^{\mathcal{M}^\star}(y) := \Big\{x \in \operatorname{Tub}_R(\mathcal{M}^\star) : \operatorname{dist}(x, \mathcal{M}^\star) \leq \alpha,\ \operatorname{Proj}_{\mathcal{M}}(x) \in B_\delta^{\mathcal{M}^\star}(y)\Big\}, \tag{K.1}$$

where $B_\delta^{\mathcal{M}^\star}(y) := \{z \in \mathcal{M}^\star : d_{\mathcal{M}^\star}(z, y) \leq \delta\}$.

**The surrogate.**   Let $t_0 > 0$ be fixed and define

$$\nu := \mu_{\text{data}} * \mathcal{N}(0, t_0 I_D), \qquad \widehat{\mu_{\text{proj}}} := \text{Proj}_{\widehat{\mathcal{M}}\#} \nu,$$

where $\text{Proj}_{\widehat{\mathcal{M}}} : \mathbb{R}^D \to \mathbb{R}^D$ is the terminal-time probability-flow map (9). We first rewrite Theorem 2.4 in a more modular form:

**Theorem K.1 (Coverage of $\widehat{\mu_{\text{proj}}}$)** *Assume $\mu_{\text{data}}$ has a density $p$ w.r.t. $\text{Vol}_{\mathcal{M}^\star}$ satisfying*

$$0 < p_{\min} \le p \le p_{\max} < \infty \qquad on\ \mathcal{M}^\star.$$

*Fix any tube radius $\rho \in (0, \zeta_{\min})$.*

*Assume the terminal-time analysis provides the following two conclusions for $\text{Proj}_{\widehat{\mathcal{M}}}$:*

(N) *(normal contraction) there exists $\alpha > 0$ such that*

$$\text{dist}\big(\text{Proj}_{\widehat{\mathcal{M}}}(x), \mathcal{M}^\star\big) \le \alpha \qquad for\ \nu\text{-a.e. } x; \tag{K.2}$$

(T) *(restricted tangential drift) there exists $\widetilde{\delta} > 0$ such that*

$$\sup_{x \in \text{Tub}_\rho(\mathcal{M}^\star)} d_{\mathcal{M}^\star}\big(\text{Proj}_{\mathcal{M}}(x), \text{Proj}_{\mathcal{M}}(\text{Proj}_{\widehat{\mathcal{M}}}(x))\big) \le \widetilde{\delta}. \tag{K.3}$$

*Define $\delta := 3\widetilde{\delta}$ and assume $\delta \le \text{inj}(\mathcal{M}^\star)/2$.[7] Then there exists a constant $c_{\min} \in (0, 1)$ depending only on $p_{\min}, p_{\max}, t_0, \rho$ and geometric parameters of $\mathcal{M}^\star$ such that $\widehat{\mu_{\text{proj}}}$ $(\alpha, \delta, c_{\min})$-covers $\mu_{\text{data}}$ in the sense of Theorem 2.3.*

By Theorem J.1 and Theorem I.4, the terminal-time flow satisfies the normal and tangential controls (K.2)–(K.3) with

$$\alpha = \tilde{\mathcal{O}}(\varepsilon) \qquad \text{and} \qquad \widetilde{\delta} = \tilde{\mathcal{O}}(\sqrt{\varepsilon}).$$

Under the statistical rate $\varepsilon = \tilde{\mathcal{O}}\big(N^{-(\beta-1)/(2k)}\big)$, this yields

$$\alpha = \tilde{\mathcal{O}}\big(N^{-(\beta-1)/(2k)}\big), \qquad \delta = 3\widetilde{\delta} = \tilde{\mathcal{O}}\big(N^{-(\beta-1)/(4k)}\big),$$

which completes the proof of Theorem 2.4. Thus, for the rest of this section, we focus on the proof of Theorem K.1.

**Proof.**

**Uniform lower bound on thickened balls.**   Fix $y \in \mathcal{M}^\star$ and define the preimage event

$$E_y := \Big\{x \in \text{Tub}_\rho(\mathcal{M}^\star) : \text{Proj}_{\mathcal{M}}(x) \in B_{\widetilde{\delta}}^{\mathcal{M}^\star}(y)\Big\}.$$

We first show that $E_y$ is mapped by $\text{Proj}_{\widehat{\mathcal{M}}}$ into $B_{\delta,\alpha}^{\mathcal{M}^\star}(y)$, up to a $\nu$-null set. Indeed, if $x \in E_y$, then by (K.3),

$$d_{\mathcal{M}^\star}\big(\text{Proj}_{\mathcal{M}}(\text{Proj}_{\widehat{\mathcal{M}}}(x)), \text{Proj}_{\mathcal{M}}(x)\big) \le \widetilde{\delta}.$$

Since also $d_{\mathcal{M}^\star}(\text{Proj}_{\mathcal{M}}(x), y) \le \widetilde{\delta}$, the triangle inequality on $(\mathcal{M}^\star, d_{\mathcal{M}^\star})$ yields

$$d_{\mathcal{M}^\star}\big(\text{Proj}_{\mathcal{M}}(\text{Proj}_{\widehat{\mathcal{M}}}(x)), y\big) \le d_{\mathcal{M}^\star}\big(\text{Proj}_{\mathcal{M}}(\text{Proj}_{\widehat{\mathcal{M}}}(x)), \text{Proj}_{\mathcal{M}}(x)\big) + d_{\mathcal{M}^\star}\big(\text{Proj}_{\mathcal{M}}(x), y\big) \le 2\widetilde{\delta} \le \delta,$$

so $\text{Proj}_{\mathcal{M}}(\text{Proj}_{\widehat{\mathcal{M}}}(x)) \in B_\delta^{\mathcal{M}^\star}(y)$. Moreover, (K.2) gives $\text{dist}(\text{Proj}_{\widehat{\mathcal{M}}}(x), \mathcal{M}^\star) \le \alpha$ for $\nu$-a.e. $x$. Together these imply $\text{Proj}_{\widehat{\mathcal{M}}}(x) \in B_{\delta,\alpha}^{\mathcal{M}^\star}(y)$ for $\nu$-a.e. $x \in E_y$. Consequently,

$$\widehat{\mu_{\text{proj}}}\big(B_{\delta,\alpha}(y)\big) = \nu\big(\text{Proj}_{\widehat{\mathcal{M}}}^{-1}(B_{\delta,\alpha}^{\mathcal{M}^\star}(y))\big) \ge \nu(E_y). \tag{K.4}$$

It remains to lower bound $\nu(E_y)$ uniformly in $y$. To this end, we employ the standard technique of *local trivialization*.

---

[7] It is well-known that the reach lower bound implies a corresponding lower bound on the injectivity radius; see, e.g., Aamari et al. (2019).

**Local trivialization and a convolved-mass lower bound.** Let

$$U_{\widetilde{\delta}}(y) := \mathrm{Proj}_{\mathcal{M}}^{-1}\big(B_{\widetilde{\delta}}^{\mathcal{M}^\star}(y)\big) \cap \mathrm{Tub}_\rho(\mathcal{M}^\star).$$

By definition, $U_{\widetilde{\delta}}(y) = E_y$. Consider the map

$$\Psi_y : U_{\widetilde{\delta}}(y) \to B_{\widetilde{\delta}}^{\mathcal{M}^\star}(y) \times \{n \in N\mathcal{M}^\star : \|n\| < \rho\}, \qquad \Psi_y(x) = (\mathrm{Proj}_{\mathcal{M}}(x), n_x), \qquad n_x := x - \mathrm{Proj}_{\mathcal{M}}(x). \tag{K.5}$$

On $\mathrm{Tub}_\rho(\mathcal{M}^\star)$, $n_x$ is well-defined and normal to $\mathcal{M}^\star$ at $\mathrm{Proj}_{\mathcal{M}}(x)$. Thus $\Psi_y$ provides the natural "basepoint + normal displacement" coordinate system on the patch $U_{\widetilde{\delta}}(y)$; the next proposition turns this into a quantitative lower bound on $\nu(U_{\widetilde{\delta}}(y))$.

**Proposition K.2 (Lower bound for $\nu(U_{\widetilde{\delta}}(y))$ via local trivialization)** *Assume $\rho \in (0, \zeta_{\min})$ and $\widetilde{\delta} \leq \min\{\mathrm{inj}(\mathcal{M}^\star)/2,\ \zeta_{\min}/4\}$. Let $Y \sim \mu_{\mathrm{data}}$ and $Z \sim \mathcal{N}(0, t_0 I_D)$ be independent and set $X_0 := Y + Z \sim \nu$. Then for every $y \in \mathcal{M}^\star$ and every $\kappa \in (0, \widetilde{\delta})$,*

$$\nu\big(U_{\widetilde{\delta}}(y)\big) = \mathbb{P}\big(X_0 \in U_{\widetilde{\delta}}(y)\big) \ \geq\ p_{\min} \, \mathrm{Vol}_{\mathcal{M}^\star}\big(B_{\widetilde{\delta}-\kappa}^{\mathcal{M}^\star}(y)\big) \, \mathbb{P}(\|G_k\| \leq a) \, \mathbb{P}(\|G_{D-k}\| \leq \rho/2), \tag{K.6}$$

*where $G_m \sim \mathcal{N}(0, t_0 I_m)$ and*

$$a := \min\Big\{\frac{\rho}{2},\ \frac{\kappa}{2L_\rho}\Big\}, \qquad L_\rho := \frac{\zeta_{\min}}{\zeta_{\min} - \rho}. \tag{K.7}$$

**Proof.** Fix $y \in \mathcal{M}^\star$ and $\kappa \in (0, \widetilde{\delta})$. Define the "inner" ball $B_- := B_{\widetilde{\delta}-\kappa}^{\mathcal{M}^\star}(y)$ and the event

$$\mathsf{E} := \{Y \in B_-\} \cap \{\|Z_T(Y)\| \leq a\} \cap \{\|Z_N(Y)\| \leq \rho/2\},$$

where $Z_T(Y) \in T_Y\mathcal{M}^\star$ and $Z_N(Y) \in N_Y\mathcal{M}^\star$ denote the tangent/normal components of $Z$ with respect to an orthonormal frame at $Y$ (defined below), and $a$ is as in (K.7).

**Step 1: orthonormal trivialization of the normal bundle over $B_{\widetilde{\delta}}^{\mathcal{M}^\star}(y)$.** Since $\widetilde{\delta} \leq \mathrm{inj}(\mathcal{M}^\star)/2$, the geodesic ball $B_{\widetilde{\delta}}^{\mathcal{M}^\star}(y)$ is geodesically convex and contractible; in particular, the restricted normal bundle over this ball is trivializable. Thus we can choose smooth orthonormal fields

$$e_1(u), \dots, e_k(u) \in T_u\mathcal{M}^\star, \qquad \nu_1(u), \dots, \nu_{D-k}(u) \in N_u\mathcal{M}^\star, \qquad u \in B_{\widetilde{\delta}}^{\mathcal{M}^\star}(y),$$

forming an orthonormal basis of $\mathbb{R}^D$ at each $u$. Let $U(u) \in O(D)$ be the orthogonal matrix whose columns are $(e_1(u), \dots, e_k(u), \nu_1(u), \dots, \nu_{D-k}(u))$. For $z \in \mathbb{R}^D$, define

$$\begin{pmatrix} z_T(u) \\ z_N(u) \end{pmatrix} := U(u)^\top z \in \mathbb{R}^k \times \mathbb{R}^{D-k}.$$

By rotational invariance of $Z \sim \mathcal{N}(0, t_0 I_D)$, conditionally on $Y = u$ we have

$$Z_T(Y) \sim \mathcal{N}(0, t_0 I_k), \qquad Z_N(Y) \sim \mathcal{N}(0, t_0 I_{D-k}), \qquad Z_T(Y) \perp Z_N(Y),$$

and these conditional laws do not depend on $u$.

**Step 2: deterministic inclusion $\mathsf{E} \subseteq \{X_0 \in U_{\widetilde{\delta}}(y)\}$.** On $\mathsf{E}$, set $x_0 := Y + Z_N(Y)$, so $\|x_0 - Y\| \leq \rho/2 < \zeta_{\min}$ and $x_0 \in \mathrm{Tub}_{\zeta_{\min}}(\mathcal{M}^\star)$. By the normal-fiber property of projections under positive reach (standard for metric projections on tubes),

$$\mathrm{Proj}_{\mathcal{M}}(x_0) = Y. \tag{K.8}$$

Moreover, $X_0 = x_0 + Z_T(Y)$ satisfies $\|X_0 - x_0\| = \|Z_T(Y)\| \leq a \leq \rho/2$, hence

$$\mathrm{dist}(X_0, \mathcal{M}^\star) \leq \|X_0 - Y\| \leq \|Z_T(Y)\| + \|Z_N(Y)\| \leq \rho,$$

so $X_0 \in \mathrm{Tub}_\rho(\mathcal{M}^\star)$ and $\mathrm{Proj}_{\mathcal{M}}(X_0)$ is defined.

We next control the basepoint $\mathrm{Proj}_{\mathcal{M}}(X_0)$. The projection $\mathrm{Proj}_{\mathcal{M}}$ is Lipschitz on $\mathrm{Tub}_\rho(\mathcal{M}^\star)$ with constant $L_\rho = \zeta_{\min}/(\zeta_{\min} - \rho)$: for all $x, x' \in \mathrm{Tub}_\rho(\mathcal{M}^\star)$,

$$\|\mathrm{Proj}_{\mathcal{M}}(x) - \mathrm{Proj}_{\mathcal{M}}(x')\| \leq L_\rho \|x - x'\|. \tag{K.9}$$

Applying (K.9) with $x = X_0$ and $x' = x_0$, and using (K.8),

$$\| \operatorname{Proj}_{\mathcal{M}}(X_0) - Y \| = \| \operatorname{Proj}_{\mathcal{M}}(X_0) - \operatorname{Proj}_{\mathcal{M}}(x_0) \| \leq L_\rho \, \| X_0 - x_0 \| = L_\rho \, \| Z_T(Y) \| \leq L_\rho \, a \leq \kappa/2.$$

To convert this Euclidean bound into a geodesic one, use the local comparison

$$d_{\mathcal{M}^\star}(u, v) \leq 2 \| u - v \| \qquad \text{for all } u, v \in B_{\widetilde{\delta}}^{\mathcal{M}^\star}(y), \tag{K.10}$$

which holds since $\widetilde{\delta} \leq \zeta_{\min}/4$ and $\mathcal{M}^\star$ has reach $\zeta_{\min}$ (this is a standard result; see Theorem I.3). Since $Y \in B_- \subseteq B_{\widetilde{\delta}}^{\mathcal{M}^\star}(y)$ and $\| \operatorname{Proj}_{\mathcal{M}}(X_0) - Y \| \leq \kappa/2 < \widetilde{\delta}$, we also have $\operatorname{Proj}_{\mathcal{M}}(X_0) \in B_{\widetilde{\delta}}^{\mathcal{M}^\star}(y)$, so (K.10) applies and yields

$$d_{\mathcal{M}^\star}\big(\operatorname{Proj}_{\mathcal{M}}(X_0), Y\big) \leq 2 \| \operatorname{Proj}_{\mathcal{M}}(X_0) - Y \| \leq \kappa.$$

Therefore,

$$d_{\mathcal{M}^\star}\big(\operatorname{Proj}_{\mathcal{M}}(X_0), y\big) \leq d_{\mathcal{M}^\star}\big(\operatorname{Proj}_{\mathcal{M}}(X_0), Y\big) + d_{\mathcal{M}^\star}(Y, y) \leq \kappa + (\widetilde{\delta} - \kappa) = \widetilde{\delta},$$

i.e. $\operatorname{Proj}_{\mathcal{M}}(X_0) \in B_{\widetilde{\delta}}^{\mathcal{M}^\star}(y)$. Together with $X_0 \in \operatorname{Tub}_\rho(\mathcal{M}^\star)$, this shows $X_0 \in U_{\widetilde{\delta}}(y)$. Hence $\mathsf{E} \subseteq \{X_0 \in U_{\widetilde{\delta}}(y)\}$.

**Step 3: lower bound $\mathbb{P}(\mathsf{E})$.** Since $\mathsf{E} \subseteq \{X_0 \in U_{\widetilde{\delta}}(y)\}$,

$$\nu\big(U_{\widetilde{\delta}}(y)\big) = \mathbb{P}(X_0 \in U_{\widetilde{\delta}}(y)) \geq \mathbb{P}(\mathsf{E}).$$

By the conditional independence from Step 1,

$$\mathbb{P}(\mathsf{E}) = \mathbb{P}(Y \in B_-) \cdot \mathbb{P}(\|G_k\| \leq a) \cdot \mathbb{P}(\|G_{D-k}\| \leq \rho/2).$$

Finally, using $p \geq p_{\min}$ on $\mathcal{M}^\star$,

$$\mathbb{P}(Y \in B_-) = \mu_{\mathrm{data}}(B_-) \geq p_{\min} \operatorname{Vol}_{\mathcal{M}^\star}(B_-) = p_{\min} \operatorname{Vol}_{\mathcal{M}^\star}\big(B_{\widetilde{\delta}-\kappa}^{\mathcal{M}^\star}(y)\big),$$

which gives (K.6). ∎

Now, back to the proof of Theorem K.1:

**Convert $\nu(E_y)$ into a coverage inequality.** By (K.4) and $E_y = U_{\widetilde{\delta}}(y)$,

$$\widehat{\mu_{\mathrm{proj}}}\big(B_{\delta, \alpha}^{\mathcal{M}^\star}(y)\big) \geq \nu\big(U_{\widetilde{\delta}}(y)\big).$$

Apply Theorem K.2 with $\kappa = \widetilde{\delta}/2$ to obtain

$$\widehat{\mu_{\mathrm{proj}}}\big(B_{\delta, \alpha}^{\mathcal{M}^\star}(y)\big) \geq p_{\min} \operatorname{Vol}_{\mathcal{M}^\star}\big(B_{\widetilde{\delta}/2}^{\mathcal{M}^\star}(y)\big) \mathbb{P}(\|G_k\| \leq a) \mathbb{P}(\|G_{D-k}\| \leq \rho/2), \tag{K.11}$$

with $a = \min\{\rho/2, \ (\widetilde{\delta}/2)/(2L_\rho)\}$.

On the other hand, since $p \leq p_{\max}$,

$$\mu_{\mathrm{data}}\big(B_\delta^{\mathcal{M}^\star}(y)\big) \leq p_{\max} \operatorname{Vol}_{\mathcal{M}^\star}\big(B_\delta^{\mathcal{M}^\star}(y)\big). \tag{K.12}$$

Because $\delta \leq \operatorname{inj}(\mathcal{M}^\star)/2$ and $\mathcal{M}^\star$ is compact, small geodesic balls have uniformly comparable volumes: there exist $0 < c_{\mathrm{vol}} \leq C_{\mathrm{vol}} < \infty$ depending only on $\mathcal{M}^\star$ such that for all $y \in \mathcal{M}^\star$ and all $0 < s \leq \delta$,

$$c_{\mathrm{vol}} \, s^k \ \leq \ \operatorname{Vol}_{\mathcal{M}^\star}\big(B_s^{\mathcal{M}^\star}(y)\big) \ \leq \ C_{\mathrm{vol}} \, s^k. \tag{K.13}$$

Applying (K.13) with $s = \widetilde{\delta}/2$ and $s = \delta = 3\widetilde{\delta}$ yields

$$\frac{\operatorname{Vol}_{\mathcal{M}^\star}(B_{\widetilde{\delta}/2}^{\mathcal{M}^\star}(y))}{\operatorname{Vol}_{\mathcal{M}^\star}(B_\delta^{\mathcal{M}^\star}(y))} \ \geq \ \frac{c_{\mathrm{vol}}(\widetilde{\delta}/2)^k}{C_{\mathrm{vol}} \delta^k} = \frac{c_{\mathrm{vol}}}{C_{\mathrm{vol}}} \cdot \frac{1}{2^k \, 3^k}. \tag{K.14}$$

Combining (K.11), (K.12), and (K.14) gives

$$\widehat{\mu_{\mathrm{proj}}}\big(B_{\delta, \alpha}^{\mathcal{M}^\star}(y)\big) \ \geq \ c_{\min} \, \mu_{\mathrm{data}}\big(B_\delta^{\mathcal{M}^\star}(y)\big) = c_{\min} \, \mu_{\mathrm{data}}\big(B_{\delta, \alpha}^{\mathcal{M}^\star}(y)\big),$$

where the last equality uses that $\mu_{\mathrm{data}}$ is supported on $\mathcal{M}^\star$, and we may take

$$c_{\min} := \frac{p_{\min}}{p_{\max}} \cdot \frac{c_{\mathrm{vol}}}{C_{\mathrm{vol}}} \cdot \frac{1}{2^k \, 3^k} \cdot \mathbb{P}(\|G_k\| \leq a) \cdot \mathbb{P}(\|G_{D-k}\| \leq \rho/2). \tag{K.15}$$

This proves item 2 of Theorem 2.3, uniformly in $y \in \mathcal{M}^\star$, and completes the proof. ∎

**Remark K.3** *It follows directly from the proof that one may take $a = \rho = \mathcal{O}(\zeta_{\min})$ in (K.15).*

**Remark K.4 (Explicit Gaussian factors)** *The Gaussian terms in (K.6) admit closed-form expressions in terms of (incomplete) gamma functions, and can be bounded explicitly using elementary volume arguments. For $G_m \sim \mathcal{N}(0, t_0 I_m)$ and any $t > 0$,*

$$\mathbb{P}(\|G_m\| \leq t) \ \geq \ (2\pi t_0)^{-m/2} \exp\!\left(-\frac{t^2}{2t_0}\right) \omega_m \, t^m, \qquad \omega_m := \frac{\pi^{m/2}}{\Gamma(\frac{m}{2} + 1)}. \tag{K.16}$$

*Indeed, (K.16) follows by lower bounding the Gaussian density on the Euclidean ball $\{z : \|z\| \leq t\}$ by its minimum value and multiplying by the ball volume. Applying (K.16) with $(m, t) = (k, a)$ and $(m, t) = (D - k, \rho/2)$ yields a fully explicit lower bound for the product $\mathbb{P}(\|G_k\| \leq a)\,\mathbb{P}(\|G_{D-k}\| \leq \rho/2)$ appearing in the coverage constants.*

