# OpenReview forum: "Manifold Generalization Provably Proceeds Memorization in Diffusion Models"
_ICLR.cc/2026/Workshop/GRaM — ICLR 2026 Workshop GRaM Poster_

### Official Review · Reviewer_DQ4a · 2026-02-17
**Interesting results on diffusion models poorly presented**

**Rating:** 4
**Confidence:** 3

**Review:**

**Summary:**

The paper studies diffusion models under the manifold hypothesis and argues that learning a coarse score, that is one that takes discrete values in fixed radius balls, first learns the geometry of the data, and then it is refined to learn its distribution. The main result of this work is that, in the small-noise regime, using coarse denoising score matching recovers the projection map onto the implicit manifold (provided by the manifold hypothesis) leads to better coverage rates than full density estimation.

To do so, they formalize a notion of on-manifold coverage and prove minimax optimal rates for recovering the manifold under a eikonal class of functions and a localized score matching. They argue that this is the reason why diffusion models can general truly novel samples.

**Strengths:**
- The theoretical results combining manifold estimation theory, score matching and ODE/SDE analysis are interesting.
- The idea of geometric recovery being statistically easier provides a fresh perspective on memorization vs. generalization in diffusion models.
- The rates for Hausdorff error, projection accuracy, coverage resolution are impressive and tight, also appearing to be minimax-optimal.
- The techniques and concepts used, reach, tubular neighborhoods, principal angles and smoothness classes, are rigorous and well-grounded.

**Weaknesses:**
- Writing and structure are poor. There are no conclusions and extremely strong claims about the practical implications while skimming through some of its key details about practical implications.
- The empirical results are shown with little to no context: what was the data? what was the manifold?, what was the model used to train it?, any hyperparameters? This makes these results feel suspicious, specially since the appendices are otherwise quite complete.
- Assumptions 1 and 2 are extremely strong and not used by previous works such as (Tang & Yang 2024).
- The eikonal class is very idealized and its relationship to how this can be used in practice only briefly mentioned, prior to also adding a number of other constraints.
- Despite the link www.racetothebottom.xyz/posts/PL-smooth-unique/ being valid, clicking on the one in the pdf says it’s invalid, this should be fixed prior to publication.
- Extremely counterintuitive that to answer the research question: how can an inaccurate score still yield non-memorized high quality samples? It is necessary to make assumptions 1 and 2, which consist of supposing a good enough accuracy. I understand that assumption 2 the loss is coarse so the idea is there, writing explicitly how this is not a problem would help the reader understand.

**Overall assessment:**
Despite containing strong and interesting theoretical results, the way they are presented makes it difficult to assess their relevance for the field. I think this paper would benefit from a restructuring and rewriting before being presented to the community so that its results can be properly communicated and thoroughly understood.

**Pmlr Suitability:**

Yes

---

### Official Review · Reviewer_JPQT · 2026-02-20
**Score learns data manifold**

**Rating:** 6
**Confidence:** 2

**Review:**

This paper studies generalization properties of diffusion models from a manifold learning perspective, motivated by the manifold hypothesis. The authors argue that diffusion models learn the underlaying data manifold prior to the data distribution. I believe the idea of shifting the focus from density estimation to manifold recovery is interesting, although it has already been documented in previous works (as the authors also acknowledge). In this paper, the authors provide a more statistical analysis of this phenomenon.

In particular, the theoretical framework about coverage of the  data manifold is well-presented and insightful. The introduction of the Eikonal equations as a function class is, to the extend of my knowledge,  novel.  Nevertheless, it remains unclear to me how realistic this function class assumption is in practice. I am unsure whether common neural network parametrizations would approximately satisfy these geometric constraints, or whether this is a rather strong assumption.  Regarding the technical details of the proofs, the arguments look correct and carefully developed, but I didn't check carefully all the details in the Appendix.

One limitation of this paper is the paper lack of experimental contributions. I think some of the theoretical claims could be analyzed with relatively simple synthetic experiments, to illustrate the coverage phenomenon for example. Additionally, Figure 1 is not sufficiently documented as implementation details are not included, not even in the Appendix. So their claims appear difficult to evaluate.

Overall, the paper presents nice theoretical ideas about generalization capabilities of diffusion models from a geometric perspective. However, I think it would be nice to add some experimental evidence to demonstrate how these ideas manifest in practice. In addition, the manuscript will also benefit from a conclusion to help the reader understand the main points of the paper.

**Pmlr Suitability:**

Yes

---

### Official Review · Reviewer_eCcu · 2026-02-22
**Novel theoretical analysis of diffusion models under manifold hypothesis**

**Rating:** 7
**Confidence:** 4

**Review:**

**Summary**

This paper presents a theoretically meaningful study of generalization in diffusion models under a manifold-supported data distribution. The authors develop a finite-sample analysis that formalizes the intuition that diffusion models can recover the manifold geometry before (and more easily than) they recover fine-scale distributional information, and they introduce a **coverage** notion to capture “non–mode-dropping” behavior at a geometric resolution.  It strengthens recent studies such as Li et al 2025 that shows diffusion models capture the manifold structure effectively.
Overall, I find the work timely and valuable for workshop discussion: it provides a clean conceptual lens (geometry vs. density) and produces nontrivial sample-size–dependent statements for manifold settings.

**Strengths**
1. The paper derives explicit asymptotic dependencies on the training sample size (N). Finite-sample geometric guarantees of this type are novel in the diffusion literature, and the sample-size–dependent coverage bounds are a meaningful contribution.

2. The introduction of a **coverage** criterion provides a geometrically interpretable measure of generalization that is distinct from global divergence metrics (e.g., KL). The distinction between recovering manifold geometry and recovering on-manifold density is conceptually clear and useful.


**Weaknesses**

1. Assumption 2 (local coarse optimality under LDSM) is central to the main theoretical guarantees, but its practical realism is not fully clarified. In particular:

   * The requirement involves a supremum over training reference points, implying uniform local control. It is unclear whether such uniform control is realistic under standard DSM training.
   * It would be helpful to understand under what conditions the assumed (O(1/N))-type scaling (combined with the sup requirement) can reasonably hold.

2. The paper introduces Local Denoising Score Matching (LDSM) as an analytical device, but it is not entirely clear whether:

   * LDSM is intended as a practical training procedure,
   * or primarily as a theoretical construct to formalize “coarse-score” behavior.

   The extent to which standard DSM training is expected to satisfy the LDSM-style coarse optimality assumption is not fully justified. Clarifying this relationship would strengthen the interpretation of the results.

3. Minor points:  Additional closely related works could be discussed for better positioning:

   * Liu et al NeurIPS 2025 discusses a Taylor-expansion on manifold to similarly to Li et al. This can be addressed.
   * From dynamical viewpoint, Chandramoorthy et al NeurIPS 2025 study the stability of the manifold structure using Lyapunov exponents at the terminal behavior.


**Questions**

Is Local Denoising Score Matching (LDSM) meant to be implemented in practice, or is it primarily an analytical tool?
   If LDSM is intended to be implementable, how does one realize it efficiently in a minibatch setting? Does approximating local neighborhoods effectively increase the required batch size or computational cost?



**References**

Z. Liu, W. Zhang, and T. Li. Improving the Euclidean diffusion generation of manifold data by mitigating score function singularity. Advances in Neural Information Processing Systems (NeurIPS), 2025.

N. Chandramoorthy and A. De Clercq. When and how can inexact generative models still sample from the data manifold? Advances in Neural Information Processing Systems (NeurIPS), 2025.

**Pmlr Suitability:**

Yes

---

### Meta-Review · Area_Chair_RUF7 · 2026-02-25

**Decision:**

Accept

**Metareview:**

This paper proves that diffusion models with coarsely learned scores can still generate novel, high-quality samples because the score first encodes the geometry of the data manifold before it captures fine-scale density information. They introduce a coverage criterion in order to prove their results.

The reviewers agree that this is an interesting paper with **novel and rigorous theoretical results**. We recommend this paper for our proceedings track.

While no revisions are required, the camera-ready version could benefit from a few additions in response to reviewers DQ4a and eCcu: a brief discussion of the key assumptions, some clarification of the empirical results, and a small conclusion section, with parts that could be added or moved in the appendix if needed.

**Relevance To Proceedings:**

Yes — suitable for PMLR (long paper)

**Relevance To Workshop:**

Yes — suitable for GRaM

---

### Decision · Program_Chairs · 2026-03-02

Accept (Poster)